# Principled distillation of UK Biobank phenotype data reveals underlying structure in human variation

Caitlin E. Carey [1,2,3] ✉, Rebecca Shafee [1,4,5], Robbee Wedow [1,2,6,7,8,9,10],
Amanda Elliott[1,2,11], Duncan S. Palmer [1,2,12,13,14], John Compitello[1,2,12],
Masahiro Kanai [1,2,12], Liam Abbott[1,2], Patrick Schultz[1,2,12],
Konrad J. Karczewski [1,2], Samuel C. Bryant [1,2], Caroline M. Cusick[1],
Claire Churchhouse [1,2,12], Daniel P. Howrigan [1,2], Daniel King[1,2,12],
George Davey Smith [12,15,16], Benjamin M. Neale [1,2,3,12,17],
Raymond K. Walters [1,2,11,17] ✉ & Elise B. Robinson [1,2,3,17]

Data within biobanks capture broad yet detailed indices of human variation, but biobank-wide insights can be difficult to extract due to complexity and scale. Here, using large-scale factor analysis, we distill hundreds of variables (diagnoses, assessments and survey items) into 35 latent constructs, using data from unrelated individuals with predominantly estimated European genetic ancestry in UK Biobank. These factors recapitulate known disease classifications, disentangle elements of socioeconomic status, highlight the relevance of psychiatric constructs to health and improve measurement of pro-health behaviours. We go on to demonstrate the power of this approach to clarify genetic signal, enhance discovery and identify associations between underlying phenotypic structure and health outcomes. In building a deeper understanding of ways in which constructs such as socioeconomic status, trauma, or physical activity are structured in the dataset, we emphasize the importance of considering the interwoven nature of the human phenome when evaluating public health patterns.

Decades of public and private investment in large-scale data collection and aggregation have in recent years yielded data repositories called biobanks, which link health outcomes to biological samples for hundreds of thousands of individuals (for example, refs. 1–3). Such resources often include thousands of variables drawn from electronic health records (EHR), self-report survey measures, laboratory assays, and physical and cognitive assessments[3], thus enabling research across diverse data types.

Although these massive resources now power discovery in human health and disease (for example, refs. 4,5), the tremendous breadth and depth of data can obfuscate larger patterns present across a biobank. For example, indicators of a particular health construct might be reflected across a range of variables nested within different data modalities in ways that are both anticipated and unexpected. To more completely consider the correlated human health landscape, approaches are needed that can identify underlying structure and reduce thousands of variables into a smaller number of constructs, refining data in a way that is digestible and scalable for human interpretation.

Dimensionality reduction is a common task in many domains, with a recent proliferation of methods having been applied to biobank-scale data. Principal component analysis (PCA), for example, provides a lower-dimensional representation of the strongest axes of variation in a dataset. It has been leveraged to, for instance, identify dimensions of genetic ancestry in genotype data[6], extract features from individual

biobank questionnaires[7], and identify sets of genetic variants with similar patterns of association across thousands of genome-wide association studies (GWAS)[8]. Other data reduction approaches prioritize identifying correlated sets of variables across data types[9], modelling latent classes[10] or creating lower-dimensional representations for visualization purposes[11,12]. Deep learning methods such as autoencoders and transformers have been used to integrate 'omics data across modalities[13,14] and to extract relevant features from electronic health records (EHRs)[15,16], respectively.

Biobank analyses have devoted relatively less attention to factor analysis (FA)[17–19], an approach commonly used in the social sciences that models the observed correlation between variables as arising from one or more shared continuous latent (unobserved) factors. Factor analysis has the benefit of being model based, facilitating more direct statistical inference than descriptive summaries (for example, PCA) or 'black box' algorithmic solutions, and it directly prioritizes extracting factors that have a simple relationship to the observed items, when possible. Conventionally, factor analysis is applied to sets of items within a single questionnaire to identify or confirm underlying structure and estimate scores that more accurately measure the latent constructs captured. This approach has scaled successfully to large cohorts, for example, in modelling measures of cognition[20] or wellbeing[21], or in identifying structure across disease comorbidities[22]. Recently, factor analysis has been adapted to model the structure of genetic, rather than observed phenotypic, correlations across traits[23–25], with further extensions to other types of large-scale 'omics data proposed[26].

In the current study, we modify and extend the factor analytic approach to explore the structure of a much broader set of multimodal biobank phenotypes. We apply this approach to the cross-sectional phenome of UK Biobank (UKB)[3] and evaluate whether the identified structure is (1) informative about relationships that might be unexpected or normally obscured and (2) effectively summarized by factor scores to enable more powerful analyses of linked phenotypic and genetic data. We also consider the limitations and trade-offs involved in applying factor analysis at this scale. While our approach is one of many that may be used to distill biobank data, these analyses reemphasize the value of principled dimensionality reduction, and reveal important insights into human variation across the complex and multifaceted human phenome.

We note that these analyses are restricted to individuals of predominantly European genetic ancestry, which is a key limitation to their generalizability. This analytic decision is driven by the statistical necessity of having a large ancestrally homogeneous group of research participants in one biobank, to limit impacts of population stratification. Since there is a relatively limited number of respondents of predominantly non-European genetic ancestry in UKB, and since results from factor analysis are non-transferable across datasets, we neither have the necessary sample size within UKB to meaningfully compare results across ancestries, nor could we compare our European-ancestry-only factor analysis results from UKB to factor analysis results from a sample in a different, primarily non-European-ancestry biobank. However, we avidly urge current and future biobank efforts to collect large enough samples sizes in diverse genetic ancestries to make these sorts of critical comparisons possible. This step is imperative to not risk exacerbating health disparities[27].

## Results

### Distilling the phenotypic landscape

We use an adapted multistage factor analytic approach to distill UKB's phenotypic data, first considering 2,772 phenotypes and all unrelated individuals with predominantly estimated European genetic ancestry ($N = 361,144$; Extended Data Fig. 1 and Supplementary Text). FA methods treat observed variables (items) as measures of a small number of unobserved (latent) factors $F$, with corresponding effect sizes or

'loadings', $\Lambda$, and item-specific residuals $\epsilon$, allowing the observed covariance between items $\Sigma$ to be fit as $\Sigma = \Lambda\mathrm{Cov}(F)\Lambda' + \mathrm{Cov}(\epsilon)$[19]. We deconvolve the latent factors and loadings by prioritizing sparsity criteria for the loadings while requiring the factors to be uncorrelated (orthogonal). The content of each fitted factor can therefore be interpreted as reflecting the correlation between sets of items conditional on the relationships captured by other factors (Supplementary Text). To reduce the impact of differential availability of certain assessments across participants, we begin by fitting a factor model in a core subsample of 42,325 individuals with high assessment-level completion and 898 phenotypes with low (that is, mean = 9.1%, s.d. = 10.7%) missingness and prevalence above 1% (Supplementary Text and Table 1). Genetic comparison of the fitted factors between the core subsample and the full European-ancestry cohort suggests that the identified architecture generalizes well (Supplementary Fig. 1).

The final model consists of 35 orthogonal latent factors drawn from 505 observed items in UKB (Fig. 1 and Extended Data Fig. 2). Initial model fitting with exploratory FA (Methods, and Supplementary Figs. 2 and 3) selected a model explaining 18.5% of overall variance across input phenotypes that demonstrated good absolute fit in the modelling sample ($N = 33,860$; root mean square error of the approximation (RMSEA) = 0.015; comparative fit index (CFI) = 0.883; see Supplementary Table 2 for additional fit metrics across the modelling and holdout samples; see Supplementary Text for more on interpretation of absolute and relative fit in the context of this analysis). While total variance explained is low for a conventional factor analysis with a much smaller set of individual items taken from a single survey, the aim was to model and extract major axes of variation across a biobank with a broad and diverse set of items. The same number of components extracted from a principal component analysis explains a similar amount of overall variance (21.6%), but the allocation of items and weights across components differs from that of the factors, highlighting important distinctions between these two common dimensionality reduction algorithms (Extended Data Fig. 3, and Supplementary Table 3 and Text). The initial exploratory model was then refined for more robust modelling in confirmatory FA (Methods and Supplementary Text, Figs. 4 and 5, and Table 4). We adopt a final confirmatory FA model that demonstrates acceptable fit based on absolute fit metrics in a holdout sample of 8,465 UKB participants (RMSEA = 0.028; standardized root mean squared residual (SRMR) = 0.076; Supplementary Table 2 and Text).

The loadings of a factor's component items reflect their relative importance within the construct captured[17–19]. On average, each factor influences 32.49 items (s.d. = 20.48; range 3–84; Supplementary Table 5), with items known to be correlates of numerous health outcomes, such as 'Health satisfaction' and 'Overall health rating'[28,29], loading on as many as 10 factors. Because individual assessments in UKB were designed to capture specific phenotypic domains, many factors draw heavily from individual categories within the 72 UKB questionnaires and assessments included in the factor model (Supplementary Table 6), with 9 deriving most of their items from a single source. Most factors, however, draw items from multiple sources (mean = 11.11; s.d. = 6.26; range 2–26), highlighting the utility of a factor analytic approach in identifying phenotypic structure across assessments.

Factors related to medical diseases recapitulate previous clinical, epidemiological and biological knowledge without any expert or manual curation. Factor 12, for example, captures correlates of hypertension across the phenome including diagnostic items (for example, self-reported hypertension and measured blood pressure), risk factors (for example, family history, body mass index (BMI) and waist circumference), comorbidities (for example, self-reported high cholesterol and diabetes) and relevant medications (for example, diuretics and calcium channel blockers). Along similar lines, Factor 16 captures commonly used diagnostic indicators of coronary artery disease, such as self-report and hospital inpatient diagnoses (for example, chronic ischaemic heart disease and myocardial infarction); symptoms (for

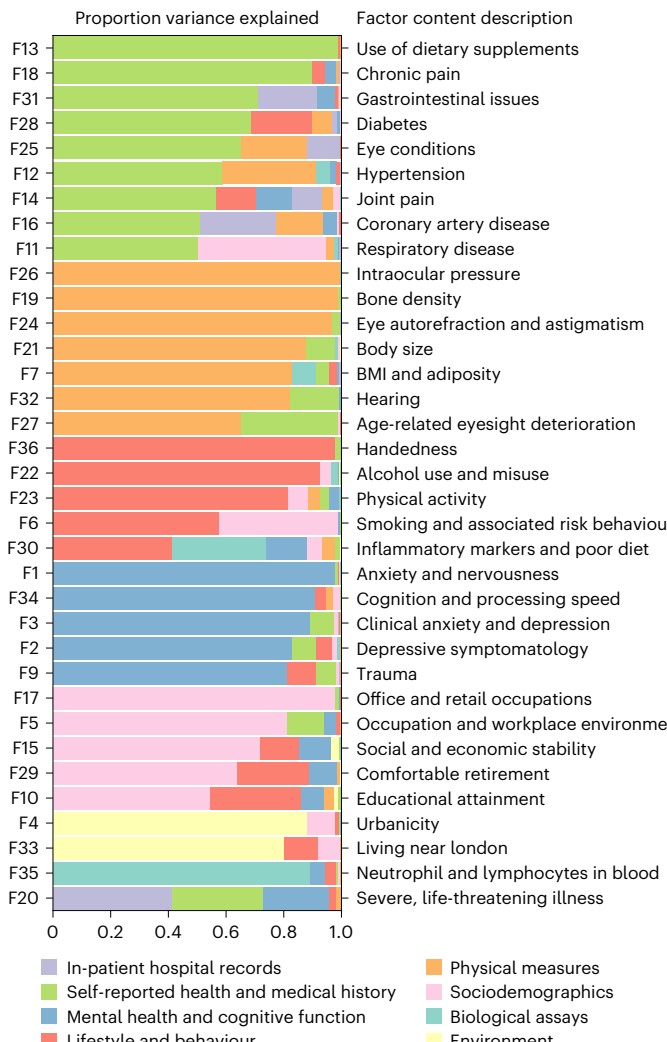

**Fig. 1 | Makeup of factors in the final model.** Horizontal bars represent proportion of variance explained in a given factor score by each of 8 major categories of assessment in UKB, estimated using hierarchical partitioning. To the left, factors are numbered in order of variance extraction in the exploratory factor analysis. To the right, brief descriptions of the items contained within a factor are listed, arrived at by expert consensus of coauthors and colleagues.

example, angina, pain in the chest or throat and shortness of breath); and medications (for example, aspirin, beta-blockers and statins; Supplementary Table 5). Beyond these more expected associations, the factors capture less established links between variables such as BMI and frequency of computer gaming (Factor 7), diet and education (Factor 10), and information technology occupations and performance on computerized cognitive tests (Factor 34).

It is important to note, however, that interpretation of the factors relies on their forced orthogonality. Returning to the medical domain, Factor 28 captures diabetes diagnoses and responsive lifestyle changes such as altering one's diet and reducing sugar intake. Had factors not been modelled as independent, Factor 28 would probably share variance with Factors 12 and 16 above, as well as with Factor 7, which captures BMI and adiposity; each of these factors captures aspects of the well-known 'metabolic syndrome'. In the context of our model, Factor 28, and each factor more generally, is most accurately interpreted as representing the remaining covariance of the items within it (for example, diabetes diagnosis and treatment), once accounting for the variance explained by the other related (for example, cardiometabolic) and unrelated factors.

## Characterizing factors with linked biobank data

To further characterize the 35 factors and their relation to public health outcomes, we generate weighted sum scores indexing individuals' values for each factor in the full sample of individuals with predominately European-estimated genetic ancestry ($N = 361,099$ due to subsequent participant withdrawals from UKB). Scores are computed on the basis of the fitted factor model using weighted linear combinations of observed items while accounting for missingness (Supplementary Text and Figs. 6 and 7). The expected correlation between estimated scores and the corresponding latent factors[30] are generally strong (mean = 0.863, s.d. = 0.073, range: 0.611[Factor 28] to 0.974[Factor 4]), suggesting good reliability. Using these scores, we conduct a series of follow-up analyses including: correlating factors with 403 top-level medical diagnostic codes (or 'phecodes'; Supplementary Table 7 and Extended Data Fig. 4) and 28 biomarkers (Extended Data Fig. 5); predicting all-cause mortality from survey completion to the last date at which participant death records are available (Fig. 2a); and performing a series of genetic analyses including GWAS of the factors (Supplementary Table 8), heritability estimation and enrichment (Fig. 2b and Extended Data Fig. 6), and genetic correlations (Extended Data Fig. 7, and Supplementary Fig. 8 and Table 9). In each case we control for covariates including age, sex, genetic principal components and UK Biobank assessment centre, since we observe that most factors are significantly related to these features (Methods and Supplementary Table 10). We also create polygenic scores (PGS) in the National Longitudinal Study of Adolescent to Adult Health (Add Health[31]) on the basis of factor GWAS summary statistics to supplement and extend our findings in an independent, US-based cohort.

All but three factors are associated with prospective mortality after correction for multiple testing (that is, $P < 0.0014$ for 35 factors), reflecting the significance of these axes of variation across individuals (Fig. 2a). Factor 20, which includes items such as number of surgeries, cancer diagnosis and 'diagnosis with a life-threatening illness' has the highest mortality prediction across all 35 factors (hazard ratio [HR] = 1.62[1.59–1.64]). Other factors among the strongest predictors of mortality capture constructs such as joint pain and disability (Factor 14), trauma (Factor 9), social and economic stability (Factor 15) and physical activity (Factor 23). These results provide a benchmark for assessing which constructs are most central to health outcomes and highlight the utility of using factor scores prospectively as additional longitudinal data are incorporated into a biobank.

Genetic results from GWAS of the factor scores are generally consistent with previous findings for related phenotypes. For example, Factor 11, which captures asthma diagnosis and related medications, comorbidities and lab findings, has a strong genetic correlation with a previous GWAS of asthma[32] ($r_g = 0.89(0.01)$; Intercept$_{Gcov}$ = 0.59[0.02]; UKB-excluded $r_g = 0.82[0.04]$), and its heritability is enriched in regions of the genome associated with blood and immune cell types ($P = 1.61 \times 10^{-6}$; Extended Data Fig. 6), highlighting a known key role for the immune system in the pathogenesis of chronic respiratory disease. Similarly, GWAS of Factor 16 captures known lipid biology, with many of the 33 significant loci mapped to core lipid metabolism genes such as *LPA, LPL, LDLR, SORT1, APOE* and *PCSK9* (Fig. 3d), and exhibits strong genetic correlation with a previous coronary artery disease GWAS[33] ($r_g = 0.87[0.02]$; Intercept$_{Gcov}$ = 0.42[0.01]; UKB-excluded $r_g = 0.81[0.04]$). Comparison of Factor 28 to a previous GWAS of type 2 diabetes[34], however, reveals imperfect capture of the clinical definition ($r_g = 0.68[0.02]$, Intercept$_{Gcov}$ = 0.36[0.01]; UKB-excluded $r_g = 0.70[0.03]$; Extended Data Fig. 8), probably due to its modelled orthogonality from other factors capturing cardiometabolic constructs. More specifically, the factor has higher genetic overlap with cholesterol measures (for example, total cholesterol[35] $r_g = 0.29[0.04]$ vs 0.04[0.03]) but lower overlap with BMI[36] ($r_g = 0.23[0.03]$ vs 0.49[0.03]) and an inverse correlation with blood pressure[37] ($r_g = -0.18[0.02]$ vs 0.20[0.02]), reflective of its inclusion of high cholesterol in its

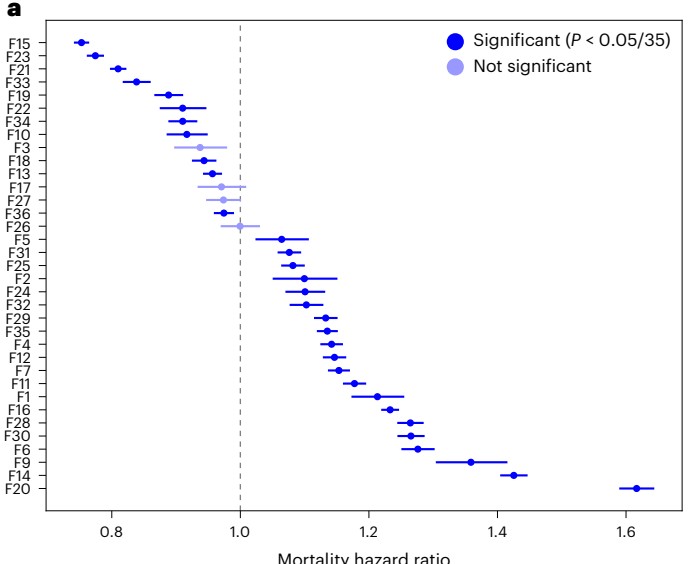

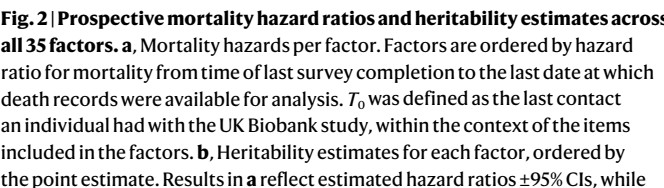

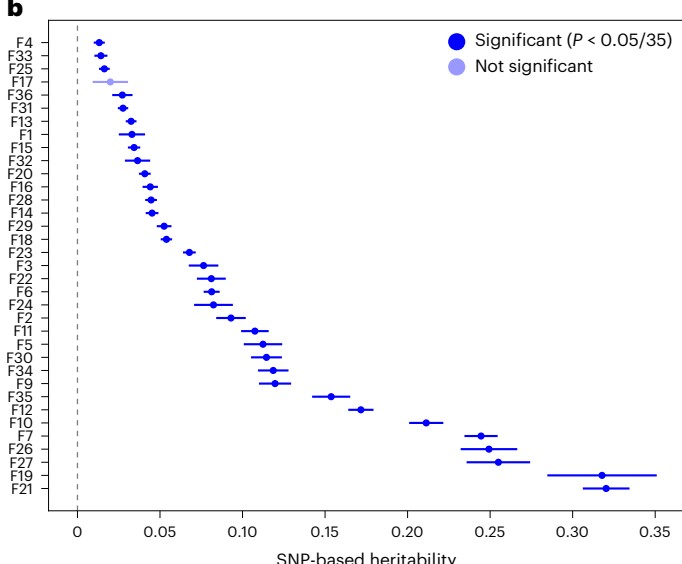

**Fig. 2 | Prospective mortality hazard ratios and heritability estimates across all 35 factors. a**, Mortality hazards per factor. Factors are ordered by hazard ratio for mortality from time of last survey completion to the last date at which death records were available for analysis. $T_0$ was defined as the last contact an individual had with the UK Biobank study, within the context of the items included in the factors. **b**, Heritability estimates for each factor, ordered by the point estimate. Results in **a** reflect estimated hazard ratios ±95% CIs, while

results in **b** reflect estimated SNP heritabilities ±1 standard error. For both panels, darker blue boxes remain significant after adjustment for multiple comparisons. Covariates for both analyses included 20 genetic PCs, age, chromosomal sex, $age^2$, age × chromosomal sex, $age^2$ × chromosomal sex and assessment centre. Mortality analyses additionally included a covariate representing days from baseline assessment to $T_0$.

factor definition and its forced independence of Factors 7 and 12 described above.

Even where genetic results are qualitatively similar, factor analysis aids in genetic discovery by combining shared information across items, which decreases measurement error of the underlying construct. Across all 35 factors, this increase in power from modelling covariance across items yields 548 loci, of 2,329 total, that are not genome-wide significant in GWAS of their top 5 component items (Fig. 3c). More generally, all but one factor are significantly heritable after multiple testing correction (mean factor $h_g^2 = 0.10[0.09]$; Fig. 2b), and the observed-scale single nucleotide polymorphism (SNP) heritability of the 35 factors is on the whole higher than for the 505 component items (mean item observed-scale $h_g^2 = 0.05[0.07]$; 2-sample $t$-test $P = 0.002$; Fig. 3a). Within factors, 20 of 35 have higher SNP heritability point estimates than all 5 of their top-loading items (for example, Fig. 3b). Although factors that capture measured physical characteristics have higher heritabilities on average (mean = 0.22, s.d. = 0.10), the largest benefits are observed for factors containing mostly dichotomous or ordinal self-report items; these items are likely to have higher measurement error than empirically measured continuous items (Supplementary Note of ref. 38; Extended Data Fig. 9).

Furthermore, identifying patterns of shared and non-shared genetic signal across items within a factor can provide insight into how they relate. Factor GWAS are best powered for shared signal, identifying nearly all (that is, 91.5% of) loci significant in at least 3 of 5 top-loading items, but much fewer (that is, 20.7% of) loci significant in only 1 of the top items, which are thus likely to be item-specific (Fig. 3c; for example, Fig. 3d). In other words, loci that are common to multiple component items are more likely to capture shared covariance across these items and thus be picked up in the factor GWAS. Item-specific effects could be hypothesized to reflect qualitatively distinct genetic mechanisms, but attempts to partition non-shared effects between top items and their corresponding factors generally identify similar loci and tissue enrichments as the factor GWAS, implying that 'item-specific' signal residual on the factor may commonly reflect differential relative effects

of shared loci instead (Supplementary Text and Fig. 9). Direct comparisons between GWAS of the top items in a factor have the potential to be more informative by suggesting causal relationships between observed items, such as evidence for a partially causal effect of doctor-diagnosed asthma on self-reported on-the-job breathing problems within Factor 11 (latent causal variable (LCV) genetic causality proportion = 0.80, $P = 3.98 \times 10^{-5}$; Supplementary Fig. 10; see Supplementary Text for further details on LCV analyses); however, power to establish these relationships is limited by the power of the GWAS for each item.

Combining factor definitions with individual-level data and linked biobank phenotypes allows us to further characterize the factors and uncover associations relevant for health. Beyond the general trends described above, we use the remaining results sections to highlight key findings in individual factors across the sociodemographic, psychiatric and behavioural domains.

## Disentangling subdomains of socioeconomic status

Socioeconomic status (SES) is one of largest single predictors of health and mortality[39], and, for research purposes, is traditionally estimated with indicators of education, occupation and income[40]. Three factors predominantly include SES-related variables and reflect correlates of occupation (Factor 5), educational attainment (EA; Factor 10), and social and economic stability (Factor 15), respectively, across the phenome. These factors pull from a range of questionnaires, from diet to employment history to social support, capturing items both traditionally and non-traditionally considered SES indicators (Supplementary Table 5). Factor 10 includes classic SES items such as educational attainment and job codes, as well as apparent cohort-specific correlates such as intake of ground coffee/espresso, muesli and wine. Factor 5 captures jobs such as low-ranking military, physical labour and factory occupations, as well as work environments full of fumes or noise. Finally, Factor 15 reflects social and economic stability, including social support networks, loneliness, home ownership, household income and never having been divorced.

The SES factors are pervasively, yet differentially, associated with health outcomes in UKB (Supplementary Table 7 and Extended Data

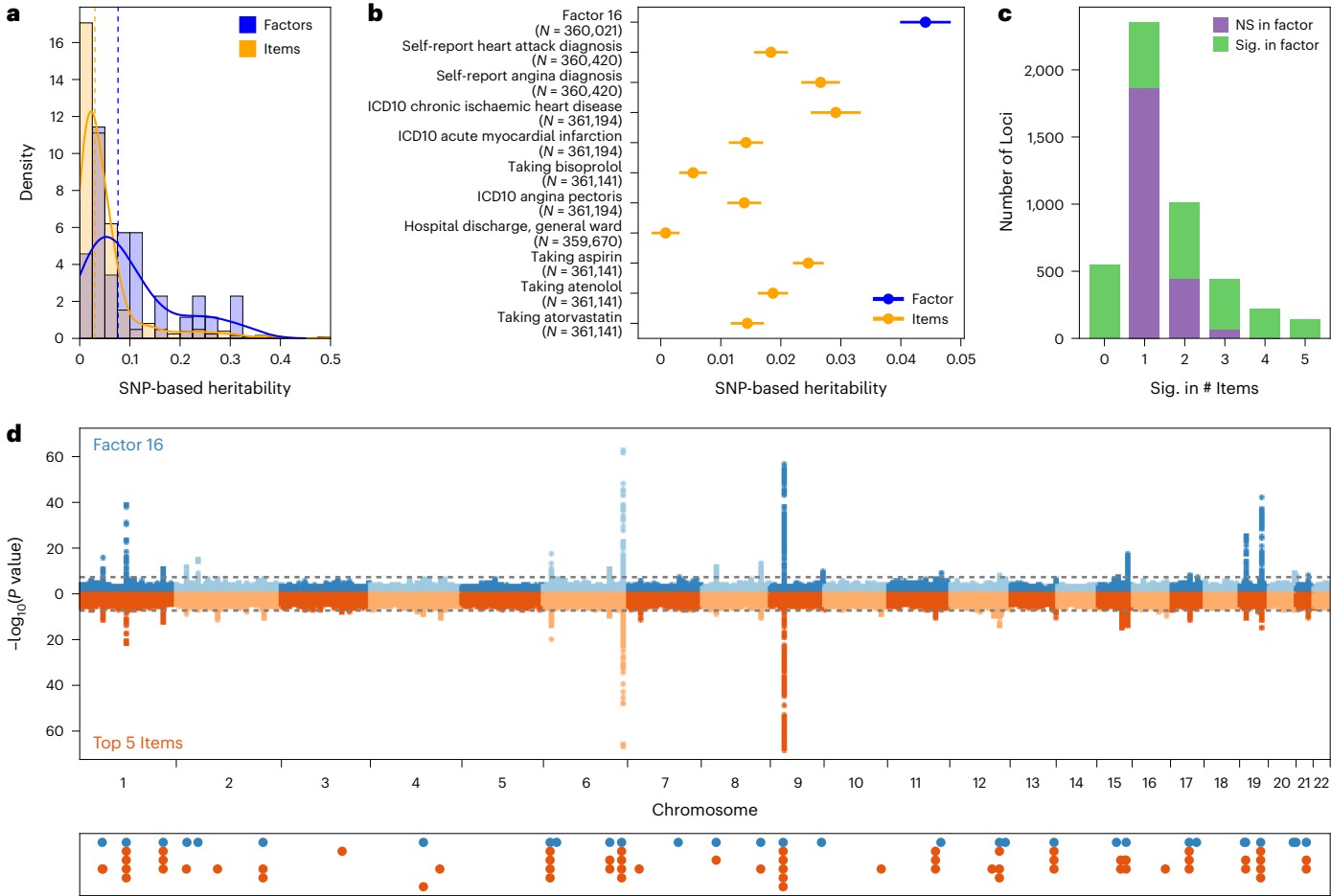

**Fig. 3 | Genetic properties of factors vs items. a**, Distributions of SNP-based heritability point estimates for items and factors, with density curves overlaid. Dashed vertical lines represent the median point estimate for each category. **b**, SNP heritability point estimates with standard error bars shown for an example factor, Factor 16, and its top 10 component items by loading. **c**, Number of GWAS significant loci ($P < 5 × 10^{-8}$ for Bonferroni-adjusted significance within a given phenotype) across all 35 factors and their top 5 component items by loading. Loci shown in purple are significant only in GWAS of one or more top items. Loci shown in green are significant in GWAS of the factors. For example, there are 2,350 loci that are significant in GWAS of only one of a factor's top 5 items (second bar in the graph). Of these loci, 486 are also significant in GWAS of the corresponding factor (shown in green). **d**, Comparison of loci identified in Factor 16 (top of Miami plot) versus its top 5 items by loading (bottom of Miami plot). Below the Miami plot are all loci across the factor (in blue) and top items (in orange), demonstrating the patterns presented in **c** at the single-factor level. All $P$ values for **c** and **d** are from two-sided tests in GWAS using linear regression for each variant and the covariates described in Methods.

Fig. 4). Factors 5 and 10 are much more similar to each other in their patterns of association to linked hospital inpatient phecodes (Pearson correlation of regression betas $|r| = 0.61$) than they are to Factor 15 ($|r| = 0.24$ and $0.33$, respectively). Factors 5 and 10 are most distinguished by differential associations with respiratory and cardiometabolic diseases, with Factor 5 being more associated with respiratory diseases and Factor 10 with cardiometabolic diseases. Factor 15, in contrast, is much more protective against hospitalization for mental health and substance use disorders than the other two factors. Factor 15 therefore suggests a distinct domain of SES that is protective against the 'diseases and deaths of despair' that have been shown to be the most significant drivers of decreasing life expectancy in recent years[41]. In fact, Factor 15 is the most prospectively predictive of survival of all 35 factors (HR = 0.75[0.74–0.76]; Fig. 2a).

Genetic data provide additional insight into how these factors parse multiple axes of SES. Although genetic associations with SES are often correlated with environmental influences and thus cannot be interpreted as purely causal, they remain an informative tool. For example, genetic correlations across the SES factors are low to moderate, suggesting partially overlapping yet distinct domains of SES (Fig. 4a). Factor 10 shares substantial genetic overlap with previous

GWAS of EA[42] ($r_g = 0.93[0.01]$; Intercept$_{Gcov} = 0.29[0.01]$; UKB-excluded $r_g = 0.92[0.02]$) and its correlates, including household income[43] and cognitive performance[44]. Notably, the SNP heritability of Factor 10 ($h_g^2 = 0.21[0.01]$) is greater than that of the EA GWAS[38] ($h_g^2 = 0.12$), reflecting a potential benefit of including correlated measures of traditional and non-traditional SES. Factor 5, meanwhile, reflects strong and roughly equal genetic overlap with EA, region-based social deprivation[45] and household income. Factor 15 is the most distinct SES factor, with moderate genetic correlations with social deprivation and household income, but low and non-significant associations with EA and cognitive performance, respectively. Factors 5 and 15 are associated only with genetic effects on EA that operate through the inherited family environment, while Factor 10 is also associated with direct genetic effects[46]. Associations between previous SES GWAS and other factors within our model reveal that Factors 5, 10 and 15 alone do not fully capture SES, with numerous other residual correlations further reflecting the complex interconnections between SES measures and health (Supplementary Text and Fig. 11).

Polygenic score analyses of the SES factors conducted in the independent National Longitudinal Study of Adolescent to Adult Health (Add Health) dataset further solidify these key distinctions and provide

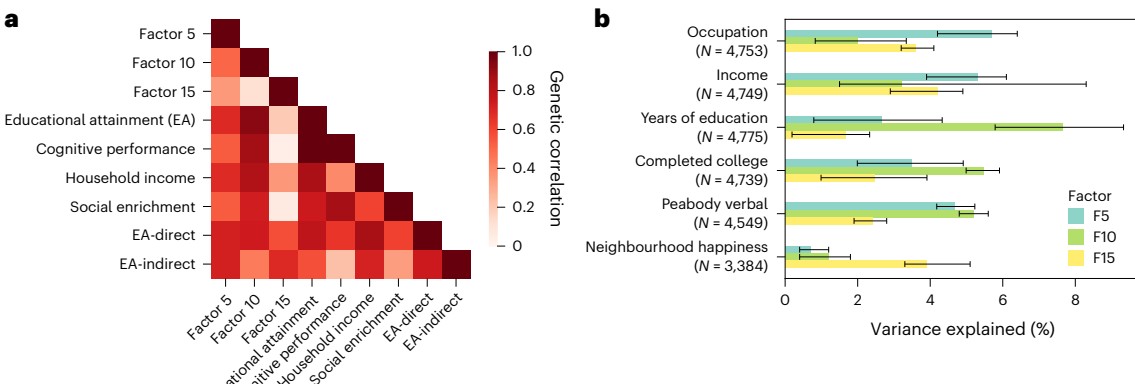

**Fig. 4 | Genetic associations of factors within the SES domain. a**, Genetic correlation across factors in the SES domain and previous GWAS of SES indicators. All genetic associations are flipped to be in the direction reflecting greater SES for consistency (for example, 'Social deprivation' becomes 'Social enrichment'). Colour of each box within the heat map indicates the strength of genetic overlap across the two corresponding phenotypes. **b**, Associations between polygenic scores derived from the SES factors and SES-related items in an outside cohort (Add Health) with corresponding sample sizes (*N*). Barplots show estimated incremental variance explained (change in $R^2$ or Nagelkerke's pseudo-$R^2$ from adding polygenic scores to regression models for continuous or binary outcomes, respectively (Methods)) with error bars representing 95% bootstrapped confidence intervals.

evidence of generalization beyond the UKB sample (Supplementary Table 11 and Fig. 4b). The Factor 10 PGS outperforms those of Factors 5 and 15 in its associations with years of education ($R^2$ = 7.7%), college completion (pseudo-$R^2$ = 5.5%) and verbal cognition ($R^2$ = 4.8%). In contrast, Factor 5 scores are the best predictors of income ($R^2$ = 5.3%) and high-paying vs low-paying jobs (pseudo-$R^2$ = 5.7%), while Factor 15 scores are most associated with neighbourhood satisfaction ($R^2$ = 3.9%). Not only do these findings demonstrate replication of our results in an independent, non-UKB sample, but the level of differentiation in prediction also articulates a clear disentangling of factors related to SES.

## Elucidating associations between trauma and health outcomes

Unlike somatic diseases with well characterized, directly measurable criteria, psychiatric constructs are not biologically defined, and diagnostic boundaries are imprecise. Experiences of trauma, for example, represent one of the most critical but understudied public health concerns globally, with significant associations to downstream chronic health problems and mortality[47]. Factor 9 places trauma within the larger behavioural and medical landscape without the need to decide which trauma-related measures to include or how to combine and weight them. The factor's component items include exposure to traumas such as feeling hated as a child or being physically abused by a family member, as well as related mental health outcomes such as post-traumatic stress disorder (PTSD), major depressive disorder, mania, psychosis, addiction and self-harm (Supplementary Table 5).

Of the 35 factors, Factor 9 has one of the highest prospective all-cause mortality hazards (HR = 1.36[1.31–1.41]; Fig. 2a), reinforcing the critical public health importance of the construct. Its associations with psychiatric and medical diagnoses are uniquely broad (Fig. 5), with strong phenotypic correlations observed across all diagnostic categories, including common circulatory (for example, hypertension, odds ratio (OR) = 1.29[1.27–1.32]), digestive (for example, acid reflux, OR = 1.34[1.30–1.38]), respiratory (for example, asthma, OR = 1.39[1.36–1.43]) and endocrine (for example, type 2 diabetes OR = 1.62[1.57–1.68]) outcomes (Supplementary Table 7). The biomarker most associated with this factor is C-reactive protein ($\beta$ = 0.11[0.004], $z$ = 31.47), a blood-based indicator of inflammation and inflammatory disorders, providing further evidence for its relevance as a clinical flag for suites of trauma-related exposures and outcomes[48].

Factor 9 shows strong genetic correlations with previous GWAS of trauma exposure[49] ($r_g$ = 0.93[0.02]; Intercept$_{Gcov}$ = 0.53[0.01]),

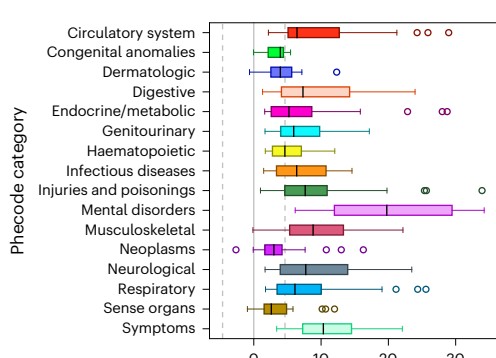

**Fig. 5 | Factor 9 associations across top-level inpatient diagnostic phecodes.** Box-and-whisker plots are shown for associations within UKB with 403 derived medical phecodes grouped by category. These associations are defined as the test statistics (that is, $z$-scores) for the factor score in a logistic regression model including our standard covariates (that is, first 20 genetic PCs, age, chromosomal sex, age², age × chromosomal sex, age² × chromosomal sex and dummy variables representing the assessment centres of origin). Boxes represent the middle quartiles of Factor 9's test statistics across phecodes within a category, with whiskers extending to maximum and minimum observed values, excluding outliers >1.5× the interquartile range away from the middle quartiles which are plotted individually. Median values per category are indicated by individual black lines inside the boxes. The dotted grey lines represent the critical test statistics for significance at two-sided $P < 0.05$ after correcting for multiple comparisons across all 403 phecodes.

childhood maltreatment[50] ($r_g$ = 0.81[0.02]; Intercept$_{Gcov}$ = 0.46[0.01]) and PTSD[51] ($r_g$ = 0.75[0.07]; Intercept$_{Gcov}$ = 0.13[0.01]). It has moderate genetic correlations with external GWAS of psychiatric (for example, schizophrenia[52] $r_g$ = 0.35[0.03]; Intercept$_{Gcov}$ = 0.02[0.01]) and substance use (for example, cannabis use disorder[53] $r_g$ = 0.45[0.05]; Intercept$_{Gcov}$ = 0.01[0.01]) outcomes (Extended Data Fig. 7). Of 8 genome-wide significant factor loci, 4 have been previously identified in GWAS of trauma phenotypes[49,50]. The remaining 4 loci novel to trauma associations have been identified in previous GWAS of psychiatric, behavioural, neural and medical outcomes (for example, refs. 54–57; Supplementary Table 8). Within the Add Health cohort, the Factor 9 PGS outperforms those derived from previous GWAS of trauma-related constructs[49–51] in predicting adulthood psychiatric

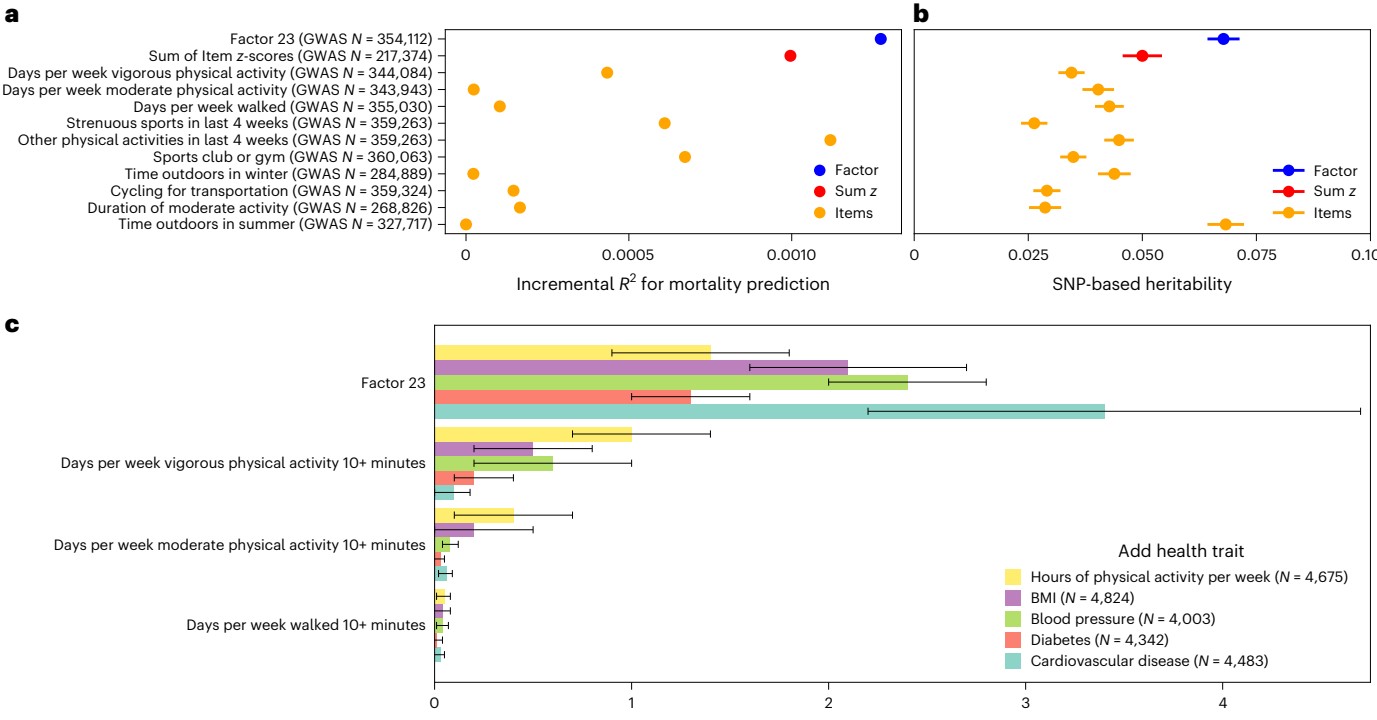

**Fig. 6 | Comparative performance of Factor 23 versus individual items.** The factor score is compared to each of the top 10 items and to an unweighted sum of $z$-scores of those items. **a**, Comparison of incremental $R^2$ for mortality prediction in $N = 217,393$ individuals with complete data for the included items and sufficient accuracy in the independent-variable factor scores for Factor 23 (Methods). The comparative baseline model for each included covariates for the first 20 genetic PCs, age, chromosomal sex, age$^2$, age × chromosomal sex, age$^2$ × chromosomal sex, dummy variables representing the assessment centres of origin and days from baseline assessment to $T_0$. **b**, Comparison of point estimates of heritability.

Results show estimated observed-scale SNP heritability ±1 standard error from GWAS with the listed sample size ($N$). **c**, Comparison of variance explained by polygenic scores for Factor 23 vs its top 3 component items for 5 relevant traits in the external Add Health study. Barplots show estimated variance explained (change in $R^2$ from adding polygenic scores to linear regression models for each outcome (Methods)) with error bars representing 95% bootstrapped confidence intervals, with a lower bound of 0 for visualization purposes. See Supplementary Table 13 for comparison to all top 10 items.

diagnoses (Supplementary Table 12), providing further evidence for an increase in genetic signal when aggregating across multiple facets of trauma-related experiences, as well as for the generalizability of that signal to an independent sample.

### Bundling related health behaviours to boost association power

Health behaviours represent a strong candidate for factor analysis since multiple observed indicators probably reflect a generalized underlying tendency capturing individual differences. Factor 23, for example, includes traditional self-reported measures of exercise frequency and duration, as well as other physical and social activity measures such as spending time outdoors, playing sports and walking for pleasure (Supplementary Table 5). It also incorporates dietary items such as fruit, vegetable and oily fish intake, probably reflecting broader correlations across pro-health behaviours. Modifiable health behaviours such as these have been shown to mitigate adverse medical outcomes, with inactivity accounting for 1% of disability-adjusted life years lost globally[58]. Indeed, Factor 23 has strong correlations across all categories of medical phecodes and a substantial association with prospective survival (HR = 0.77[0.76-0.79]; Fig. 2a).

Analyses linking Factor 23 to other biobank data identify clearer associations than comparable analyses using any individual top item (Fig. 6). Prospective analyses of all-cause mortality, for example, show weaker effects for each of Factor 23's 10 top-loading items (incremental pseudo-$R^2 = 1.27 \times 10^{-3}$ for factor vs $6.75 \times 10^{-10}$ to $1.12 \times 10^{-3}$ for individual items). An unweighted sum of $z$-scores across these top items is also more weakly associated with survival (incremental

pseudo-$R^2 = 1.00 \times 10^{-3}$) than the weighted sum used by the factor scores (Fig. 6a). Even with optimal weighting, at least 7 items are necessary to capture 80% of total factor variance, suggesting that signal in Factor 23 relies on the ability to summarize information across related items.

This stronger signal enables GWAS of Factor 23 to identify the largest number of significant loci so far for a self-reported measure of physical activity. Of 34 loci, 24 are not significant in GWAS of the top 5 items, and 25 have not previously been identified in GWAS of self-reported physical activity, although 14 were significant at a less stringent threshold (that is, $P < 5 \times 10^{-5}$ (ref. 59)). Similarly, consistent with the trend across the factors, the SNP heritability of Factor 23, 0.07(0.003), is higher than most of its component items and sum of top-item $z$-scores ($h_g^2 = 0.05$ [0.004]; Fig. 6b). This increase in power probably comes as a result of leveraging the correlated structure across items and moving towards a more continuous construct measure. Furthermore, heritability for Factor 23 is enriched for regions of the genome associated with central nervous system cell types ($P = 2.52 \times 10^{-9}$; Extended Data Fig. 6), indicating that physical activity is primarily linked to brain and behaviour, rather than traits such as muscle tone or cardiovascular health. Among top items in Factor 23, there is nominal evidence for a partially causal effect of recent participation in strenuous sports on current self-rated health (LCV genetic causality proportion = 0.37, $P = 5 \times 10^{-3}$; Supplementary Text and Fig. 10), additionally supporting a role for motivation and socialization in linking activity to perceived health.

Echoing results within UKB, polygenic score analyses within the independent Add Health dataset reveal that the Factor 23 PGS outperforms polygenic scores formed from each of its top 10

items in predicting medical and health behaviour outcomes traditionally thought of as relevant to physical activity (for example, $R^2 = 1.4\%$ for hours of physical activity per week, $R^2 = 2.1\%$ for BMI and pseudo-$R^2 = 3.4\%$ for cardiovascular disease; Supplementary Table 13 and Fig. 6c). The Factor 23 PGS even outperforms a multivariate regression model including all 10 item-based PGS (Factor 23 $R^2 = 3.4\%$ [2.2%, 4.7%]; combined top item PGSs $R^2 = 2.6\%$ [1.4%, 3.9%]), further emphasizing the power gained from combining measures across the phenome in a way that leverages underlying correlation structure.

## Discussion

The results of this study demonstrate that applying a model-based data reduction technique to hundreds of diverse items across a biobank can distill the phenotypic landscape into tractable latent constructs that differentiate individuals across interpretable axes of variation. Latent factors extracted from UKB link causes, correlates and consequences of health, behaviour and disease at the cohort level, empowering deeper analyses of the identified domains with linked genetic, registry and biomarker data, and revealing patterns that are otherwise obscured by the sheer volume of data.

For researchers interested in the structure of human variation across the phenome, these results provide a critical advance. While approaches such as principal component analysis can reduce dimensionality and maximize variance explained, FA explicitly models relationships between observed variables and the broad constructs that underlie them, with its emphasis on 'simple structure' ensuring that the content of each factor can be inferred from its top-loading items[17–19] (see Supplementary Text and Table 3, and Extended Data Fig. 3 for comparison). Indeed, within the medical domain in particular, our extracted factors capture causes, correlates and consequences of diagnoses such as asthma (Factor 11) and coronary artery disease (Factor 16), in a manner that is hypothesis-free and driven entirely by the correlation structure of the underlying data.

Summarizing items across the phenome into a smaller number of estimated factor scores based on correlation structure can increase heritability and power for downstream analyses (Fig. 3). We show that factors are, on average, more heritable than their component items, which leads to greater power for genetic discovery, particularly for loci that capture shared signal. Notably, these relative gains in power are not uniformly distributed, with factors containing mostly dichotomous or self-report survey items, rather than empirical measurements, experiencing the greatest increases in heritability (Extended Data Fig. 9). Factor 23, for example, leverages the correlation structure between self-reported physical activity, engagement in sports and other pro-health behaviours to yield factor scores that have greater heritability than 9 of the factor's top 10 items, and outperform all top 10 items in prospective survival prediction in UKB and in variance explained in morbidity outcomes by polygenic scores in a separate, US-based sample (Fig. 6 and Supplementary Table 13). Polygenic scores for Factor 9, which combines indicators of trauma and its sequelae in a hypothesis-free, data-driven manner, similarly outperform PGS based on previous binary or summative trauma measures[49–51] (Supplementary Table 12). In each case, these results support considering multiple indicators across the phenome, particularly when studying complex human phenotypes that cannot be empirically assayed.

Beyond yielding increases in power, factors underscore important trends in the social and behavioural sciences, a number of which we have described in vignettes within this manuscript. Factors 5, 10 and 15, for example, successfully deconvolve indicators of socioeconomic status into subdomains, in a way that generalizes to an independent US-based sample (Fig. 4). This data-driven demonstration of the multifaceted nature of SES supports longstanding hypotheses (for example, ref. 40) of separate axes of education, income, occupation and other elements of social standing, and begins to parse their differential relationships to health outcomes (Supplementary Table 7 and Extended Data Fig. 4). Factor 23 links physical activity with self-rated health and significantly predicts mortality despite having been constructed as orthogonal to other factors capturing physical and mental health outcomes (Fig. 2a), reinforcing evidence that physical activity improves quality of life and has numerous health benefits independent of disease status[60,61]. Factor 9, meanwhile, draws on the strength of relationships between multiple trauma indicators across the phenome, combining them in a data-driven way to demonstrate robust and ubiquitous associations with morbidity and mortality (Fig. 5). While such associations with trauma are neither unexpected nor individually novel, no analysis to our knowledge has articulated such associations at this scale across the phenome; this underscores the powerful public health impact of trauma and the value of linking multiple items together when assessing this complex construct[47].

Our approach represents only one possible way of distilling biobank-scale data into informative measures. Although comparison to PCA provides some initial intuition on the benefits of factor analysis (see Supplementary Text and Extended Data Fig. 2), this study is exploratory and not intended to recommend a single best approach. Indeed, we anticipate that comparisons across methods will prove most informative. For instance, our results modelling observed phenotypic correlations may serve as a useful comparator to those of recently developed methods for decomposing genetic correlations[8,25]. Furthermore, within the medical domain, supplementing curated, clinician-driven methods for defining disease phenotypes from registry data (for example, refs. 62,63) with data-driven connections to survey and lab-based measures where available, as suggested by our factors, could prove useful in stratifying individuals for prognosis and treatment.

The results are subject to several important caveats and limitations (see Supplementary Text for additional discussion). Perhaps most importantly, latent constructs identified via factor analysis are not 'real'; they are simply statistical relationships, or the weighted linear combination of items that capture the correlation structure from the overall dataset, regardless of whether there is some true underlying phenomenon driving the measured variables. Factors are therefore non-trivially dependent on the nature of the dataset used, including the variables measured, the characteristics of the participants and the sociodemographic context in which the data were collected[64].

Participants at the core of this study are, for example, unlikely to be representative of the global population. Analyses were restricted to UKB participants of predominately estimated European genetic ancestry, thus limiting generalizability of results, particularly as it relates to genetic inference[27]. We openly recognize this as a key limitation of these analyses, which we highlight in the introduction to this paper to emphasize its critical importance.

As an additional limitation, UK Biobank participants are known to be non-representative of the UK population as a whole, with documented ascertainment and participation biases (for example, refs. 3,65,66). Even among those participating in the study, individuals providing more complete initial responses and contributing to later optional follow-up surveys are known to be, in general, healthier and more educated[67,68]. This trend is reflected in our own core, low-missingness data group being substantially more likely to report having completed college or university (45.7% core group, 30.7% non-core group; $\chi^2 = 3,816.0$, $P < 0.001$). While polygenic score analyses in Add Health suggest that genetic signal captured in the fitted factors at least partially generalizes to individuals of European ancestry in the United States, identifying which constructs elucidated in this study are most robustly maintained across different sociopolitical, cultural and diagnostic contexts requires future work in additional cohorts.

Adapting factor analysis to permit execution at the scale and structure of UK Biobank also forced us to make analytic decisions that introduce a number of methodological limitations. Most prominently, we required orthogonality across the factors to reduce the number of

parameters to be estimated and substantially simplify matrix algebra operations. As such, the variance captured by each factor is forced to be independent, complicating interpretation; specifically, each factor must be viewed as representing the covariance structure of the items within it, once accounting for covariance modelled by the other factors. Caution must therefore be exercised in interpreting relationships within and between factors, since this independence is an artefact of the modelling approach. Relatedly, although full information maximum-likelihood estimation methods remain the 'gold standard' for modelling missingness and categorical items, their computational burden proved intractable at this scale[69]. Instead, we used pairwise deletion and Pearson correlations for the exploratory factor analysis (EFA), and used diagonal weighted least squares (DWLS) to fit the confirmatory factor analysis (CFA), which we expect to have weaker relative performance but to nevertheless be an acceptable alternative (see Supplementary Text for additional details)[70]. Future computational advances, including the use of graphics processing units, along with continued methods development[70], may allow for factor analysis and structural equation modelling to be more widely and more optimally applied at biobank scale.

As the first example to our knowledge of the factor analytic approach applied across multimodal data at biobank scale, these results provide a proof-of-concept that such methods can return both sensible and insightful relationships. This approach provides an important first step towards better embracing the full and complex measured phenome to power discovery for human health and wellbeing.

## Methods

### Ethics approvals

All research was conducted in compliance with all relevant ethics regulations. Use of the UKB data was approved under application 31063. Analysis of the UKB data was reviewed by the Partners HealthCare Institutional Review Board (IRB) (Partners Human Research), which determined in expedited review that the project met the US federal criteria definition of 'not human subjects research'. Analysis of the Add Health data was reviewed by the Office of Research Subject Protection (OSRP) at the Broad Institute of MIT and Harvard, which determined that the project met US federal criteria for exemption from IRB review.

### UK Biobank cohort

**Sample selection and quality control.** The UK Biobank is a longitudinal health study of roughly 500,000 volunteers between the ages of 40–69 at recruitment time, which was between the years of 2006–2010. Starting from the 487,409 genotyped participants in the UK Biobank second round release, we first subset to unrelated individuals with low autosomal missingness rates used for PCA in ref. 3. We adopted this selection to aid consistency with other UK Biobank applications and analyses. We then restricted the cohort to individuals of predominantly estimated European genetic ancestry based on analysis of the top six principal components (PCs). This selection was intended to be more broadly inclusive than the 'white British' criteria used by the UK Biobank genetics team[3] while still restricting to a sufficiently homogeneous and unrelated set of individuals to permit GWAS with conventional linear regression. After ancestry selection, we made final exclusions for individuals who withdrew from UK Biobank participation before the GWAS analysis and individuals who were omitted from imputation phasing (for example, individuals with sex chromosome aneuploidies). After all sample quality control (QC), there were 361,194 QC positive individuals. Between initial QC and the start of analyses for the current study, an additional 50 participants withdrew from UKB, resulting in a final $N$ of 361,144.

**Phenotype curation.** A core challenge to the analysis of such a wide range of phenotypes as those available in the UK Biobank is the curation and harmonization of the large number of variable scalings, categorizations and follow-up responses. To automate this process, we used a modified version of the PHEnome Scan Analysis Tool (PHESANT[71]). Unlike standard PHESANT, the modified version does not perform association analyses but simply generates a collection of recoded phenotypes.

The incorporation of new phenotypes requires careful examination of raw data codings and, in the case of binary phenotypes, consideration of control definition. Recodings of variables and inherent orderings of ordinal categorical variables are defined in the data-coding file, available in the GitHub repository https://github.com/astheeggeggs/PHESANT. We restricted the phenotype data to those that belong to individuals in the unrelated GWAS subset, and ran the modified version of PHESANT on the phenotypes in the UK Biobank application. For continuous phenotypes, we retained the raw version of the continuous phenotype, with no transformation applied to the data. We processed 3,011 unique phenotypes using PHESANT. For all binary phenotypes, we required a minimum case count of 100.

Along with these PHESANT-curated phenotypes, we also processed 633 ICD10 disease codes, treating all individuals with a specific ICD10 code as cases. The remaining UK Biobank samples were treated as controls. Curation of the ICD10 codes was carried out separately for computational efficiency. For the ICD10 phenotypes, individuals were assigned a vector of ICD10 primary diagnoses. We truncated these codes to the three-digit category level and assigned each individual to either case or control status for that ICD10 code in turn by checking whether their vector of primary diagnoses contains that code. Throughout, we assumed that the data contained no missingness, so the sum of cases and controls is the number of individuals in the European-ancestry subset of the UK Biobank data. Consistent with our treatment of binary phenotypes, ICD10 code case/control phenotypes were removed if less than 100 individuals in the European-ancestry subset had a given phenotype as a primary diagnosis.

### Add Health sample

Add Health originated as an in-school survey of a nationally representative sample of US adolescents enrolled in grades 7 through 12 during the 1994–1995 school year[31]. Respondents were born between 1974 and 1983, and a subset of the original Add Health respondents had been followed up with in-home interviews, allowing researchers to assess correlates of outcomes in the transition to early adulthood. In Add Health, the mean birth year of respondents is 1979 (s.d. = 1.8) and the mean age at the time of assessment (Wave 4) is 29.0 years (s.d. = 1.8).

### Factor analysis modelling

The factor analysis model treats observed variables $X$ as measures of a smaller number of unobserved latent factors $F$, with corresponding effect sizes or 'loadings', $\Lambda$, and item-specific residuals $\epsilon$[17–19,72]. Specifically, let

$$X = F\Lambda' + \epsilon \tag{1}$$

where $X$ is the $n \times p$ matrix of $p$ centred and standardized observed variables for $n$ individuals, $F$ is the $n \times t$ matrix of $n$ individuals' values for $t$ latent variables, and $\Lambda$ is the $p \times t$ matrix of the effects of the $t$ latent variables on the $p$ observed variables. This model is easily extendable to the inclusion of 'nuisance' covariates, either by explicitly adding terms for covariates or residualizing them out of $X$. For the purposes of our analyses, we chose the latter approach, which is described in later sections.

We are able to fit this model by considering the observed covariance across items.

$$\Sigma = \Lambda F'F\Lambda' + \epsilon'\epsilon \tag{2}$$

Assuming that the observed covariance between the items is fully explained by the latent factors, we denote $\Psi = E[\epsilon'\epsilon]$ as the $p \times p$

diagonal matrix of residual variances per item (that is, item uniquenesses). We additionally chose to model the latent factors as independent and fixed their scale to have unit variance, allowing us to model the observed covariance between the items with the matrix decomposition

$$\Sigma = \Lambda\Lambda' + \Psi \qquad (3)$$

Multiple methods exist for the extraction of latent factors which generally yield similar results; we considered multiple such methods below. Once factors are extracted, however, there are an infinite number of equivalent $\Lambda$s up to a particular rotation; this is referred to as 'rotational indeterminacy'. To uniquely fit the model, additional optimization criteria must be specified for $\Lambda$. For the current analysis, we relied on the 'varimax' rotation, one of a number of standard rotations that encourages sparsity, or a 'simple structure', on the model by penalizing factors or items with multiple large loadings. This rotational restriction facilitates interpretability, as the majority of the signal associated with a particular factor can be identified on the basis of a limited number of its top-loading items.

In the current study, we undertook a multistage approach to best meet the assumptions of the factor analysis methodology while adapting it for such large-scale data (Extended Data Fig. 1). We first identified a core data group consisting of 42,325 individuals and 898 items from which a stable pairwise correlation matrix could be estimated across a range of questionnaires and assessments. We split this group into modelling ($N = 33,860$) and holdout ($N = 8,465$) subgroups on the basis of an 80:20 split. After systematically removing collinear items from the dataset (730 items remaining), we performed an EFA within the modelling subgroup to determine the factor structure (that is, the number of latent factors $t$ and which elements of $\Lambda$ are non-zero). We then further refined the model suggested by the EFA, utilizing a structural equation model for CFA in the same modelling subgroup. We tested the fit of this final model, with constrained parameters, in the holdout sample. These steps are described in more detail below, as well as in the Supplementary Text.

**Core data group.** To enable the estimation of reasonably unbiased pairwise correlations between variables, we began with all individuals of European ancestry ($N = 361,144$) and all phenotypes analysed in both sexes in the initial release of the Neale Lab UKB Round 2 mega-GWAS (2,772 phenotypes; https://www.nealelab.is/uk-biobank/ukbround2announcement). We first identified a core group of individuals with a high rate of assessment completion ($N = 42,325$; see Supplementary Text: Selection of individuals for core data group). We then identified items with low missingness in this core group, sufficient prevalence (>1%), and non-structured and non-item-dependent missingness, from which pairwise correlations could be successfully estimated (898 items; see Supplementary Text: Item selection for core data group). The overall missingness rate in this final core data group was 9.1%, with missingness on each item of up to 28.6% (s.d. 10.7%), and for each individual up to 33.3% (s.d. 7.9%). See Supplementary Text: Characteristics of core data group, for more information about demographic and item composition.

From this core data group, we systematically removed collinear items to improve the stability of factor analysis estimation. Starting with a Pearson correlation matrix residualized for our chosen 'nuisance' covariates (that is, first 20 genetic PCs, age, chromosomal sex, age$^2$, age × chromosomal sex, and age$^2$ × chromosomal sex; see Supplementary Text: Selection of 'nuisance' covariates), we removed items that were redundant between observed and derived items (67 items), highly correlated with missingness (6 items), had pairwise correlations $r > 0.95$ (43 items), had squared multiple correlation (SMC) > 0.98 (43 items) or had correlation induced by 'None of the above' response categories (6 items). After these exclusions, 730 items remained for factor analysis.

Finally, this core group was further divided into modelling ($N = 33,860$) and holdout ($N = 8,465$) groups on the basis of an 80:20 split to avoid bias from overfitting in the evaluation of model fit.

**Exploratory factor analysis.** We performed an EFA using the 'psych' package in R (v.4.0.2) on a partial pairwise Pearson correlation matrix—residualized for the first 20 genetic PCs, age, chromosomal sex, age$^2$, age × chromosomal sex, and age$^2$ × chromosomal sex—within the modelling subgroup ($N = 33,860$) to determine the factor structure.

Conventional methods for selecting the number of latent factors provided inconsistent results for the data: the scree plot suggested 30–50 factors (Supplementary Fig. 2), parallel analysis suggested 177 factors, and 253 eigenvalues of the correlation matrix were >1. To best model large-scale structure across phenotypic items within UKB, we therefore based our decision for number of factors to extract primarily on the inflection point of the scree plot, in tandem with measures of model stability and factor non-triviality (see Supplementary Text: Exploratory factor analysis, for further discussion).

We explored factor solutions with an increasing number of factors using weighted least square (WLS), generalized weighted least square (GLS), minimum residual (MINRES) and unweighted least square (ULS), all with 'varimax' rotation to extract orthogonal factors. As an upper bound, Heywood and ultra-Heywood cases were observed when fitting more than 169 factors with GLS, 186 factors with WLS or 38 factors with ULS or MINRES. Inspection of these models found that WLS and GLS yielded many factors with strong loadings (>0.3) for at most 1 item, while the ULS or MINRES solutions generally yielded factors incorporating variation from multiple items (Supplementary Fig. 3).

On the basis of stability and non-triviality (that is, lack of factors with only 1–2 items with loadings >0.3), we selected a 36-factor MINRES solution (MINRES-36; root mean square of residuals = 0.02, variance explained = 18.5%; Supplementary Fig. 3d) as our preferred EFA model for subsequent refinement.

**Principal component analysis.** We compared the factor analysis to PCA, another commonly used dimensionality reduction method, using the same partial Pearson correlation matrix that served as input to the EFA. Eigenvalues and eigenvectors of this input matrix were computed using the linalg.eig function within numpy in Python. Eigenvectors corresponding to the 36 greatest eigenvalues, in order, were extracted for comparison to the 36-component model of the EFA.

**Confirmatory factor analysis.** We further refined the model suggested by the EFA, utilizing a structural equation model in the same modelling subgroup. CFA allowed us to test the fit of a more parsimonious model (that is, omitting loadings with small estimates in the EFA). Within the CFA, we also more appropriately modelled the covariance structure of the diverse variable types (that is, binary, ordinal, continuous), and more robustly handled missingness, in contrast to our decision to use a partial pairwise Pearson correlation matrix as input to the EFA.

For the 564 variables from the EFA with loadings >0.1 on at least one factor (see Supplementary Text: Selection of minimum loading for factor inclusion; Supplementary Fig. 4), missing data were imputed using classification and regression trees (CART) within the Multivariate Imputation by Chained Equations (MICE) package[73] in R, with all covariates as well as 20 additional auxiliary variables (for example, previously excluded 'gatekeeper' items and assessment centre) included as predictors. Items whose missingness pattern depended on the target item's missingness were omitted as predictors for that target item. Evaluation of this approach using synthetic missingness at random (MAR) and completely at random (MCAR) showed good convergence and minimal systematic bias. Comparisons of correlation and covariance matrices generated using pairwise deletion versus imputation revealed them to be nearly identical (Supplementary Text: Multiple imputation of core data group; Supplementary Fig. 5); as such, to

conserve computational resources, only a single imputed dataset was used for modelling purposes.

Confirmatory factor analysis models were fit to the imputed data for the modelling group (now $N = 33,854$ due to participant withdrawals during the course of the study) using structural equation modelling with an extensively modified version of the lavaan package[74] (v.0.6-3) in R (see Supplementary Text: Computing aspects of structural equation modelling; adapted code is available at https://github.com/ce-carey/ukb-factor-analysis). Correlations between variables were estimated as appropriate for their measurement scale (for example, polychoric for pairs of ordered variables, Pearson for pairs of continuous variables and polyserial for pairs containing one of each), assuming an underlying normal distribution. Continuous variables were standardized before modelling, and all variables were modelled conditional on exogenous 'nuisance' covariates (that is, first 20 genetic PCs, age, chromosomal sex, $age^2$, age × chromosomal sex, and $age^2$ × chromosomal sex). Model parameters were estimated using DWLS, with final robust standard errors and test statistics calculated using a scale-shifted approach (that is, the WLSMV option in lavaan).

Fitting the EFA-derived model using CFA yielded a number of initial errors due to a lack of estimable pairwise correlation (due to collinearity) and cell sizes of 0 for ordinal variables. After removing 9 items to address these errors, 23 Heywood cases remained, indicative of overfitting and near collinearity. These new instances of collinearity were observed in part due to the lenient pairwise $r > 0.95$ threshold used for the initial EFA and the addition of latent modelling of ordinal and categorical variables in the CFA. Items were iteratively removed until no negative residual variances remained (505 final items). In addition, once these items were removed, one factor (Factor 8) overlapped completely with another (Factor 4) and was removed to facilitate model fitting (see Supplementary Text: Differences between EFA and CFA). Supplementary Table 4 documents the reason for each variable's exclusion from the EFA to the final factor model.

Finally, we noticed that misfit in certain parts of the model was being driven by the presence of extreme outliers (see Supplementary Text: Extreme outliers of continuous variables). Therefore, we removed from analysis all individuals in the core group with values greater than 20 standard deviations from the mean on any continuous variable ($N$ in modelling group = 52; $N$ in holdout group = 13). This resulted in a final $N$ of 33,802 in the modelling subgroup.

To evaluate the applicability of the factor model beyond the modelling subgroup, we obtained fit metrics in the validation holdout subgroup (initially $N = 8,465$; $N = 8,452$ after removing continuous-variable outliers) while constraining the model parameters (that is, factor loadings) to those estimated in the training subgroup.

Traditional assessments of model fit include model chi-square ($P$ value should not be significant), RMSEA (values 0.01, 0.05 and 0.08 indicate excellent, good and acceptable fit, respectively), SRMR (values < 0.08 indicate good fit), CFI (values > 0.90 indicate good fit) and Tucker Lewis index (TLI; values > 0.90 indicate good fit). For the sake of completeness, we report all of the above values for the EFA and for both the modelling and holdout samples for the CFA in Supplementary Table 2. In assessing overall model fit, we relied primarily on the absolute fit measures (for example, RMSEA) since they are better suited to evaluating how well our fitted model approximates the structure of the observed phenotypic data (see Supplementary Text: Assessment of model fit, for more on interpretation of absolute and relative fit in the context of this analysis). In addition, where applicable we report both the uncorrected and 'scaled' versions of these metrics, where the scaled values rely on the adjusted test statistic from fitting the model with DWLS. We caution that neither of these values are fully robust for use with categorical data, but no better alternatives are currently available without computation of the likelihood, which is currently infeasible here[70].

## Factor scores

On the basis of the final factor model, we then generated latent factor scores for each individual from the values of their observed indicator items (see Supplementary Text: Factor score generation, for full details). For each individual $i$, the estimated factor score for factor $t$ is a weighted sum of the items $x_j$

$$\hat{f}_{i,t} = \sum_j a_{j,t} x_{i,j} \tag{4}$$

If we take the factor model as true, then the resulting estimates are of an individual's 'true' score for the underlying latent construct, otherwise they simply estimate the value that best approximates the observed data for each individual with the low rank approximation of the complete data modelled by the CFA. The current analysis used two sets of factor scores, corresponding to two different estimation methods to compute the factor scoring coefficients $a_{j,t}$, following previous recommendations to avoid biased results in factor score regression[75].

**Dependent variable factor scoring coefficients.** Where factor scores were used as the dependent variable in an analysis (for example, GWAS), we calculated $\hat{f}_{i,t}$ using factor scoring coefficients computed with Bartlett's method[76]:

$$A_{\mathrm{B}} = \hat{\Psi}^{-1} \hat{\Lambda} (\hat{\Lambda}' \hat{\Psi}^{-1} \hat{\Lambda})^{-1} \tag{5}$$

where $X$ is the $n \times p$ matrix of $p$ residualized and standardized observed variables for $n$ individuals, $\hat{\Lambda}$ is the $p \times t$ matrix of estimated factor loadings and $\Psi$ is the $p \times p$ diagonal matrix of estimated residual variances (that is, item uniquenesses). Bartlett's estimator is a weighted least squares solution that minimizes the residual variance of the items given the factor scores, weighting by the fitted item uniquenesses from the model.

**Independent variable factor scoring coefficients.** Where factor scores were used as the independent variable in the analysis (for example, mortality), we used factor scoring coefficients computed with the Thomson–Thurstone (regression) method[72,77],

$$A_{\mathrm{TT}} = \hat{\Psi}^{-1} \hat{\Lambda} (I + \hat{\Lambda}' \hat{\Psi}^{-1} \hat{\Lambda})^{-1} \tag{6}$$

where $I$ is an identity matrix. These factor scoring coefficients give the best linear prediction of the factor score, minimizing the sum of the expected mean squared error across factors.

**Adjustments for categorical items.** The above framework for the factor score estimators assumes that all of the item data $X$ are observed and residualized for exogenous 'nuisance' covariates. Although this is true for observed continuous items in the model, it is not true for the CFA model where the observed data are categorical and modelled through a link function.

To address the different measurement scale for categorical variables, we estimated the expected value of each individual's latent continuous variable given the observed categorical item and the fitted probit regression with exogenous 'nuisance' covariates. The residual between this expected latent value and the value predicted by the covariates was then substituted for the observed categorical variable in the factor scoring calculations.

The categorical variables would also have weaker covariances with the factor than the unobserved latent values modelled for the CFA loadings. To account for this attenuation, we transformed the loadings and residual variances corresponding to categorical items for use in factor scoring (see Supplementary Text: Modifications for categorical and missing data). Specifically, we used

$$\hat{\lambda}_{j,t}\sqrt{\mathrm{var}\left(x_{j},|,z_{k}\right)} \times \frac{\sum_{c\in C} h(c)}{\sigma_{x}} \qquad (7)$$

as loadings, where $\hat{\lambda}_{j,t}$ is the loading for item $j$ on trait $t$ estimated in the CFA, $h(c)$ is the density of the standard normal distribution at the fitted probit threshold for each category $c$ of the categorical variable, the variance of the categorical item conditional on the covariates $\mathrm{var}\left(x_{j},|,z_{k}\right)$ was estimated empirically from the residual expected latent values described above, and the standard deviation of the categorical item $\sigma_{x}$ was estimated from the class probabilities. Similarly, we substituted

$$\mathrm{var}\left(x_{j},|,z_{k}\right) \times \left(1 - \left[1 - \psi_{jj}\right] \frac{\left[\sum_{c\in C} h(c)\right]^{2}}{\sigma_{x}^{2}}\right) \qquad (8)$$

as the residual variance for categorical items in the factor scoring equations, where $\psi_{jj}$ is the estimated residual variance in the CFA and the remaining terms are defined as in the transformation of the loadings.

**Adjustments for missingness.** Factor score estimates that sum across all items as described above cannot be computed for individuals with missing data. Instead, for each individual with a set of missing items $M$, we computed factor scoring coefficients optimized for the subset of items that are observed, that is

$$A_{\mathrm{B},-M} = \Psi_{-M}^{-1}\hat{\Lambda}_{-M}\left(\hat{\Lambda}'_{-M}\Psi_{-M}^{-1}\hat{\Lambda}_{-M}\right)^{-1} \qquad (9)$$

for dependent variable (Bartlett) factor scores and

$$A_{\mathrm{TT},-M} = \hat{\Psi}_{-M}^{-1}\hat{\Lambda}_{-M}\left(I + \hat{\Lambda}'_{-M}\hat{\Psi}_{-M}^{-1}\hat{\Lambda}_{-M}\right)^{-1} \qquad (10)$$

for independent variable (Thomson–Thurstone) factor scores.

The resulting factor score estimates maintain their desired relationship to other variables conditional on each missingness pattern, consistent with the bias-avoiding method of factor score regression, but the different amounts of information about the factor available for each individual leads to heteroskedasticity in factor scores across the missingness patterns. For analyses with the factor score as the dependent variable, we addressed this heteroskedasticity using WLS regression with estimated inverse-variance weights

$$w_{i,t} = \frac{1}{A'_{-M,t}S_{-M}A_{-M,t}} \qquad (11)$$

where $S$ is the sample covariance matrix of pairwise complete observations after residualization for exogenous 'nuisance' covariates. For linear regression analyses with factor scores as the independent variable, we used Huber–White sandwich standard errors[78]. Detail on motivation for each of these adjustments is provided in the Supplementary Text.

To further limit potential artefacts from heteroskedasticity related to structured missingness and ensure consistent interpretation of scores across UKB participants, individuals were included in factor score regressions if the factor score from their observed items was expected to correlate ($r^{2} \geq 0.8$) with what their estimated factor score would be with if all items were observed (see Supplementary Text: Minimum correlation with complete data scores; Supplementary Fig. 6). Because we observed this threshold to be more universally liberal in the independent than in dependent factor scores, to allow for better concordance in samples across phenotypic and genetic analyses, we further restricted phenotypic analyses with the independent variable factor score to only those individuals included in the genetic analyses, resulting in sample sizes ranging from 75,226 (Factor 24) to 360,656 (Factor 20) (mean $N = 252,219.571[121,829.646]$).

**Validation of factor scoring adjustments.** To validate our factor-score-generating methodologies, we compared scores from our methods to those generated using a maximum-likelihood (ML)-based method in lavaan[74] for the core data group. Factor scoring was performed using the ML option in lavaan for all 10 multiple imputations of the core dataset. To test for phenotypic concordance across methods, we obtained Pearson correlation coefficients between factor scores generated using our method and the mean of those obtained in lavaan across all 10 imputations. GWAS of the lavaan-generated scores were conducted with WLS using inverse-variance weights estimated on the basis of the observed variance in scores across individuals and across imputation replicates. Heritability and genetic correlation between these GWAS results and the GWAS for our dependent-variable factor scores were compared using linkage disequilibrium (LD) score regression[79] (Supplementary Fig. 7).

### Phenotypic association analyses

To further characterize the latent factors and also reveal potentially interesting associations, we tested the independent-variable factor scores for associations with 403 top-level phecodes and 28 biomarkers in the UK Biobank, as well as with prospective mortality. Covariates for these phenotypic analyses included the first 20 genetic PCs, age, chromosomal sex, $age^{2}$, age × chromosomal sex, $age^{2}$ × chromosomal sex and dummy variables representing the assessment centres of origin, to account for residual correlation between factors and these key variables (Supplementary Table 10).

**Phecodes.** Phecodes (1,685 items) were taken from the Pan-UK Biobank pan-ancestry GWAS project (https://pan.ukbb.broadinstitute.org/)[80] and were derived from ICD-9 and ICD10 codes across a patient's inpatient hospitalization records and, if applicable, death registry data. These diagnostic codes were mapped to descriptive phecodes using scripts from the University of Michigan (available at https://github.com/umich-cphds/createUKBphenome), which derived their mappings[81,82] from those supplied by PheWAS Catalog (https://phewascatalog.org/). Phecodes were filtered to have a minimum case count of 250 in the full EUR sample, with a minimum of 25 cases per chromosomal sex, leaving 940 for analysis. Given the nested nature of the phecodes, such that top-level codes contain all diagnoses listed in subcodes, we restricted analyses to the 403 remaining top-level phecodes only. Associations were performed using a generalized linear model with a binomial link function and Huber–White ('HC0') robust standard errors[78] using the statsmodels package (v.0.13.1) in Python.

**Biomarkers.** Serum biomarkers were obtained from the UK Biobank (28 items, after excluding rheumatoid factor and oestradiol for known QC issues[83]). Associations between each independent-variable factor score and biomarker were performed using ordinary least squares regression and Huber–White ('HC0') robust standard errors[78] using the statsmodels package (v.0.13.1) in Python. Due to known issues with sample dilution[83], an additional covariate was included representing the estimated serum sample dilution factor.

**Mortality analyses.** Given that the factors from our analyses represent major axes of measured phenotypic variation in the UK Biobank, it is plausible that they would be differentially associated with downstream mortality. We therefore performed Cox proportional hazards regression to assess relative risk of mortality across individuals on the basis of independent-variable factor scores.

Since surveys and assessments were administered at different times, to avoid issues of immortal time bias, $T_{0}$ was defined as the last contact an individual had with the UK Biobank study, within the context of the items included in the factors. Of the items included in the final factor model, the differently timed assessments included: baseline, a maximum of five 24-hour diet follow-up questionnaires, a work

environment questionnaire and a mental health questionnaire. For example, if an individual completed the baseline assessment, mental health questionnaire and a 24-hour diet follow-up questionnaire, their $T_0$ would be their most recent questionnaire completion date. We included several 'continuously updating' items within the factors (for example, primary ICD10 codes and items relating to hospital stays, which are updated on the basis of linked inpatient hospital records). Therefore, for the mortality analyses, we recoded each individual's factor scores on factors containing these 'continuously updating' items with their values at $T_0$. If a person's first instance of a primary ICD10 code was dated after their $T_0$, they would be recoded and rescored as 'not' having that code.

Date of death was obtained with linked death registries, and analyses were right censored to the earliest recommended censoring date across the death registries specified by UKB for England, and Wales and Scotland (that is, 30 September 2021 at the time of analyses). In addition to the standard phenotypic analysis covariates (that is, first 20 genetic PCs, age, chromosomal sex, age$^2$, age × chromosomal sex, age$^2$ × chromosomal sex and dummy variables representing the assessment centres of origin), a covariate was added representing days from baseline assessment to $T_0$. Analyses were performed using the lifelines package (v.0.26.4) in Python.

### Genome-wide association analysis

**Variant QC.** Over 92 million imputed autosomal and X chromosome variant dosages were available in the UK Biobank release. The variant QC process focused on using widely adopted GWAS QC parameters to retain high-quality variants. After restricting to the 361,194 QC positive individuals, we retained SNPs with minor allele frequency (MAF) > 0.001, Hardy–Weinberg Equilibrium (HWE) $P$ value > $1 \times 10^{-10}$ and imputation INFO score > 0.8. INFO scores were taken directly from the UK Biobank SNP manifest file. The only exception involved SNPs annotated as having protein-truncating or missense consequences (from Ensembl VEP consequence annotation), where we relaxed the cut-off to MAF > $1 \times 10^{-6}$. After variant QC, 13,364,303 autosomal variants were retained for association analysis.

**GWAS model and implementation.** GWAS of the dependent-variable factor scores were performed in Hail (https://hail.is/) using WLS regression. Variance weights were calculated as described in the 'Factor score generation' section above. To limit the impact of structured missingness and ensure consistent interpretation of scores across individuals, individuals were included only if their score based on their missingness pattern explained 80% of variance in a hypothetical observed factor score for which no items were missing (Supplementary Fig. 6). Covariates included the first 20 genetic PCs, age, chromosomal sex, age$^2$, age × chromosomal sex, age$^2$ × chromosomal sex and dummy variables representing the assessment centres of origin. Post-association test statistics were corrected for LD score regression (LDSC) intercept to reduce potential impacts of stratification. Effective sample size for genetic analyses, taking into account missingness, was calculated as the sum of each person's inverse-variance weight divided by the regression weight assigned to a hypothetical person with 0 missingness, and ranged from 74,782 to 359,419 (mean =236,980.029[112899.969]).

**Identification of significant independent loci.** We used the FUMA[84] pipeline to identify independent genomic loci. We considered an independent locus as the region including all SNPs in pairwise LD ($r^2 > 0.6$), with the lead SNPs in a range of 250 kb from each other and independent from other loci at $r^2 < 0.1$. We used the 1000 Genomes Phase 3 Europeans reference panel to determine LD.

**Comparison of factor vs item GWAS.** To investigate the genetic properties of the factors, we compared factor GWAS to the GWAS of component items. Summary statistics for all 505 items included in

the final factor model are publicly available via the Neale Lab UKB Round 2 mega-GWAS (https://www.nealelab.is/uk-biobank/ukbround2announcement), and were adjusted for LD score intercept to be consistent with the factor summary statistics.

Independent loci for the 5 top-loading items per factor were identified using a local version of FUMA[84] and identical parameters to those used to define the factor GWAS loci. Given that the intention for these analyses was to compare loci across top items and corresponding factors, loci were henceforth defined by their basepair intervals, and loci across GWAS (for example, for a factor and its top item) were combined if their basepair intervals overlapped.

**Comparison of factor vs item SNP heritability.** For each of the 505 items included in the final factor model, observed-scale $h_g^2$ estimates were downloaded from the publicly available Neale Lab UKB Round 2 mega-GWAS Heritability Browser (https://nealelab.github.io/UKBB_ldsc/index.html).

It should be noted that conventionally, to account for differences in heritability estimates by measurement type (that is, continuous vs binary items), estimates are reported on the liability scale. Indeed, this is the default estimate reported in the Heritability Browser. However, for factor scores there is not a clear transformation to the liability scale since they are neither binary (since they are weighted sums over a large number of items) nor fully continuous (as when individual binary/categorical items contribute substantial weight). For this reason, we report all heritabilities (for both factors and constituent items) on the observed scale. Although this will tend to understate the amount of underlying genetics that may exist for binary items, it will equitably indicate the amount of genetic signal captured in the item or factor as it is currently measured.

**LD score regression analyses.** LD score regression analyses of heritability, enrichment[85] and genetic correlation[79] were performed using LDSC (available at https://github.com/bulik/ldsc) with LD scores computed in individuals of European genetic ancestry from the 1000 Genomes Project. All analyses were performed with default settings except where otherwise indicated.

*SNP heritability.* SNP-based heritability was estimated on the observed scale for each factor using stratified LD score regression (S-LDSR)[85] and v.1.1 of the baseline-LD model (available at https://alkesgroup.broadinstitute.org/LDSCORE/). We used this stratified model to more robustly fit variation in genetic signal across the genome, estimating per-SNP heritability conditional on 75 annotations, including functional categories, evolutionary constraint, histone marks, and LD- and allele frequency-related annotations. The default filter in LDSC for maximum chi-square was omitted to avoid truncating top hits at our large sample size.

*Cell-type enrichment analyses.* To gain insights into the underlying biology of the factors, we evaluated heritability enrichment of regions associated with cell-type-specific chromatin marks using S-LDSC[85] and annotations derived from the Roadmap Epigenomics Consortium[86], as described in ref. 85. To reduce multiple testing burden and give a broader summary of systems-level biology, we grouped the updated cell-type–specific annotations described in ref. 87 into 9 tissue groups (adipose, blood/immune, cardiovascular, central nervous system, digestive, liver, musculoskeletal/connective, pancreas, and other) following the same procedure described in ref. 85, taking the union of annotations belonging to each group. Consistent with recommendations in ref. 85, only factors with strongly significant heritability estimates ($z > 7$) were included in these analyses.

*Genetic correlation analyses.* Genetic correlations across factors were computed using LD score regression with LD score estimates derived from European-ancestry individuals in the 1000 Genomes Project. We

additionally ran genetic correlations between our factors and 68 traits selected from previous GWAS, spanning a number of anthropometric, medical, psychological, behavioural and sociodemographic domains (Supplementary Table 9). Traits used for genetic correlation analyses were chosen before conducting the analyses, with the agreement of the coauthors. Note that given the near-ubiquity of the UK Biobank in modern genetics research, a number of these previous GWAS share overlapping samples and, at times, component phenotype definitions, with the current study. As such, we have reported genetic correlations with UKB samples removed within the text as well, when such summary statistics were available. We also report the genetic covariance intercepts, which are a proxy for the amount of sample overlap present across pairs of summary statistics within a genetic correlation analysis[79].

### Polygenic scoring

Polygenic scores in Add Health were constructed with LDpred[88]. LDpred estimates polygenic scores using SNP weights that estimate the conditional association of each SNP accounting for LD and the estimated genetic architecture of the trait, and has been shown to have greater prediction accuracy than conventional LD pruning followed by $P$ value thresholding.

For the Add Health sample, we used the genotyped data from the Add Health prediction cohort to create the LD reference file. After imputing the genetic data to the Haplotype Reference Consortium[89] using the Michigan Imputation Server[90], we used only HapMap3 variants with a call rate >98% and a minor allele frequency >1% to construct the polygenic scores. We limited the analyses to European-ancestry individuals. Polygenic scores were calculated with an expected fraction of causal genetic markers set at 100%. In total, we used 1,168,025 HapMap3 variants to construct the polygenic scores in Add Health. We then used Plink[91] to multiply the genotype probability of each variant by the corresponding LDpred posterior mean over all variants. For all sets of summary statistics, that is, from Factor Score GWAS summary statistics, individual item GWAS summary statistics or outside publicly available GWAS summary statistics, we ensured that Add Health was not included in the discovery GWAS sample.

We then determined the association of a given polygenic score and an outcome of interest in Add Health. Outcomes used in analyses were as follows: (1) years of completed education, (2) income, (3) a binary measure of having ever completed college, (4) a binary indicator of working either a high-paying or low-paying job, (5) a measure of cognition called the Peabody Picture Vocabulary Test, (6) a 1–5 Likert scale measure of neighbourhood satisfaction ('If, for any reason, you had to move from here to some other neighbourhood, how happy or unhappy would you be?'), self-reports of doctor-diagnosed (7) panic or anxiety disorder, (8) bipolar disorder or (9) PTSD, (10) self-report of doctor-diagnosed cardiovascular disease, (11) self-report of doctor-diagnosed diabetes, (12) self-report of number of hours of physical activity per week, (13) measured BMI and (14) measured systolic blood pressure.

Prediction accuracy was based on an ordinary least squares (OLS) or logistic regression (depending on the outcome) of the outcome phenotype on the polygenic score and a set of standard controls, which included birth year, an interaction between birth year and sex, and the first 10 genetic principal components of the variance–covariance matrix of the genetic data. Variance explained by the polygenic scores was calculated in regression analyses as either the $R^2$ change (for continuous or quasi-continuous phenotypes), or the Nagelkerke's pseudo-$R^2$ change (for binary outcomes), that is, the $R^2$ or pseudo-$R^2$ of the model including polygenic scores and covariates minus the $R^2$ or pseudo-$R^2$ of the model including only covariates. The 95% confidence intervals around all pseudo-$R^2$ values were bootstrapped with 1,000 repetitions each.

### Reporting summary

Further information on research design is available in the Nature Portfolio Reporting Summary linked to this article.

## Data availability

GWAS results for our 35 factors are available through GWAS Catalog (https://www.ebi.ac.uk/gwas/studies/) via accession nos. GCST90309336 GCST90309370 (in numerical order of the factors). Access to individual-level data from the UK Biobank can be obtained by bona fide scientists through application with UK Biobank (https://www.ukbiobank.ac.uk/enable-your-research). Individual-level dependent variable (modified Bartlett) and independent variable (Thomson-Thurstone) factor scores have been returned the UK Biobank to be made available for download through their Returned Datasets Catalogue (https://biobank.ndph.ox.ac.uk/ukb/docs.cgi?id=1). Inverse-variance weights (for use in weighted least squares regression) and expected correlations between observed and hypothetical zero-missingness scores (used to exclude individuals with insufficient data for a particular factor) have also been deposited for the modified Bartlett scores. Variables within the UKB Returned Datasets Catalogue are available to researchers registered with the UK Biobank. Summary statistics for item-level GWAS are available as part of the Neale Lab UKB Round 2 Mega-GWAS (http://www.nealelab.is/uk-biobank/ukbround2announcement). For information about access to the data from the Add Health study, contact addhealth@unc.edu.

## Code availability

Analytic code used in these analyses is available through GitHub at https://github.com/ce-carey/ukb-factor-analysis (ref. 92). Documentation and instructions for installation for the lifelines Python package can be found at https://lifelines.readthedocs.io/en/stable/; those for statsmodels may be found at https://www.statsmodels.org/stable/index.html. R packages utilized for our analyses are similarly publicly available (lavaan: https://lavaan.ugent.be/; MICE: https://cran.r-project.org/web/packages/mice/index.html; and psych: https://cran.r-project.org/web/packages/psych/index.html). LD Score regression software is available at https://github.com/bulik/ldsc.

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

## Acknowledgements

C.E.C. was, and E.B.R. is, supported by National Institutes of Health (NIH) grant R01MH124851 and the Stanley Center for Psychiatric Research. R.W.'s work is supported by AnalytiXIN, which is primarily funded through the Lilly Endowment, IU Health and Eli Lilly and Company. B.M.N.'s work is supported by NIH grant 5R37MH107649 and the Novo Nordisk Foundation (NNF21SA0072102). R.K.W. is supported by the Stanley Center for Psychiatric Research and R01 MH101244. G.D.S. works within the MRC Integrative Epidemiology Unit at the University of Bristol, which is supported by the Medical Research Council (MC_UU_00032/01). The National Longitudinal Study of Adolescent to Adult Health (Add Health) is supported by grant P01 HD031921 to Kathleen Mullan Harris from the Eunice Kennedy Shriver National Institute of Child Health and Human Development (NICHD), with cooperative funding from 23 other federal agencies and foundations. Add Health GWAS data were funded by NICHD grants to Harris (R01 HD073342) and to Harris, Boardman and McQueen (R01 HD060726). The funders had no role in study design, data collection and analysis, decision to publish or preparation of the manuscript. We especially acknowledge and thank the participants in these studies for providing biological data and for their responses and non-responses to survey questions that made this study possible.

We thank S. Hyman, R. Hosking, B. Domingue, M. Nivard and T. Ulrich for carefully reading and commenting on the paper; members of the Neale and Robinson Labs, the Stanley Center for Psychiatric Research at the Broad Institute of MIT and Harvard, and the Analytic and Translational Genetics Unit at Massachusetts General Hospital for helpful discussions and feedback. This research was conducted by using the UK Biobank Resource under application 31063. We thank all cohort participants for making this study possible.

## Author contributions

C.E.C., R.S., B.M.N., R.K.W. and E.B.R. conceived the project. C.E.C., R.S., A.E., M.K., D.P.H., D.K., B.M.N., R.K.W. and E.B.R. designed the project. C.E.C., D.S.P., M.K., L.A., K.J.K., S.C.B., C.M.C., C.C., D.P.H., B.M.N., R.K.W. and E.B.R. acquired data. C.E.C., R.S., R.W., D.S.P., M.K. and L.A. analysed data. C.E.C., R.S., R.W., A.E., G.D.S., B.M.N., R.K.W. and E.B.R. interpreted data. C.E.C., D.S.P., L.A., J.C., P.S. and D.K. developed software. C.E.C., R.W., B.M.N., R.K.W. and E.B.R. wrote and edited the paper.

## Competing interests

C.E.C. is currently an employee of Novartis. R.W. is a research fellow at AnalytiXIN, which is a consortium of health-data organizations, industry partners and university partners in Indiana primarily funded through the Lilly Endowment, IU Health and Eli Lilly and Company. B.M.N. is a member of the scientific advisory board at Deep Genomics and Neumora, and a consultant of the scientific advisory board of Camp4 Therapeutics. R.K.W. has received honoraria from the Jackson Laboratory and sponsored travel from the Russell Sage Foundation in the past 36 months. G.D.S. reports Scientific Advisory Board Membership for Relation Therapeutics and Insitro. The remaining authors declare no competing interests.

## Additional information

**Extended data** is available for this paper at https://doi.org/10.1038/s41562-024-01909-5.

**Correspondence and requests for materials** should be addressed to Caitlin E. Carey or Raymond K. Walters.

[1]Stanley Center for Psychiatric Research, Broad Institute of MIT and Harvard, Cambridge, MA, USA. [2]Analytic and Translational Genetics Unit, Massachusetts General Hospital, Boston, MA, USA. [3]Center for Genomic Medicine, Massachusetts General Hospital, Boston, MA, USA. [4]Department of Genetics, Harvard Medical School, Boston, MA, USA. [5]Section on Developmental Neurogenomics, National Institute of Mental Health, Bethesda, MD, USA. [6]Department of Sociology, Purdue University, West Lafayette, IN, USA. [7]Department of Medical and Molecular Genetics, Indiana University School of Medicine, Indianapolis, IN, USA. [8]AnalytiXIN, Indianapolis, IN, USA. [9]Center on Aging and the Life Course, Purdue University, West Lafayette, IN, USA. [10]Department of Statistics, Purdue University, West Lafayette, IN, USA. [11]Department of Medicine, Harvard Medical School, Boston, MA, USA. [12]Program in Medical and Population Genetics, Broad Institute of MIT and Harvard, Cambridge, MA, USA. [13]Nuffield Department of Population Health, Medical Sciences Division University of Oxford, Oxford, UK. [14]Big Data Institute, Li Ka Shing Centre for Health Information and Discovery, University of Oxford, Oxford, UK. [15]MRC Integrative Epidemiology Unit, University of Bristol, Oakfield House, Bristol, UK. [16]Population Health Sciences, Bristol Medical School, University of Bristol, Bristol, UK. [17]These authors contributed equally: Benjamin M. Neale, Raymond K. Walters, Elise B. Robinson. ✉e-mail: cemcarey@gmail.com; rwalters@broadinstitute.org

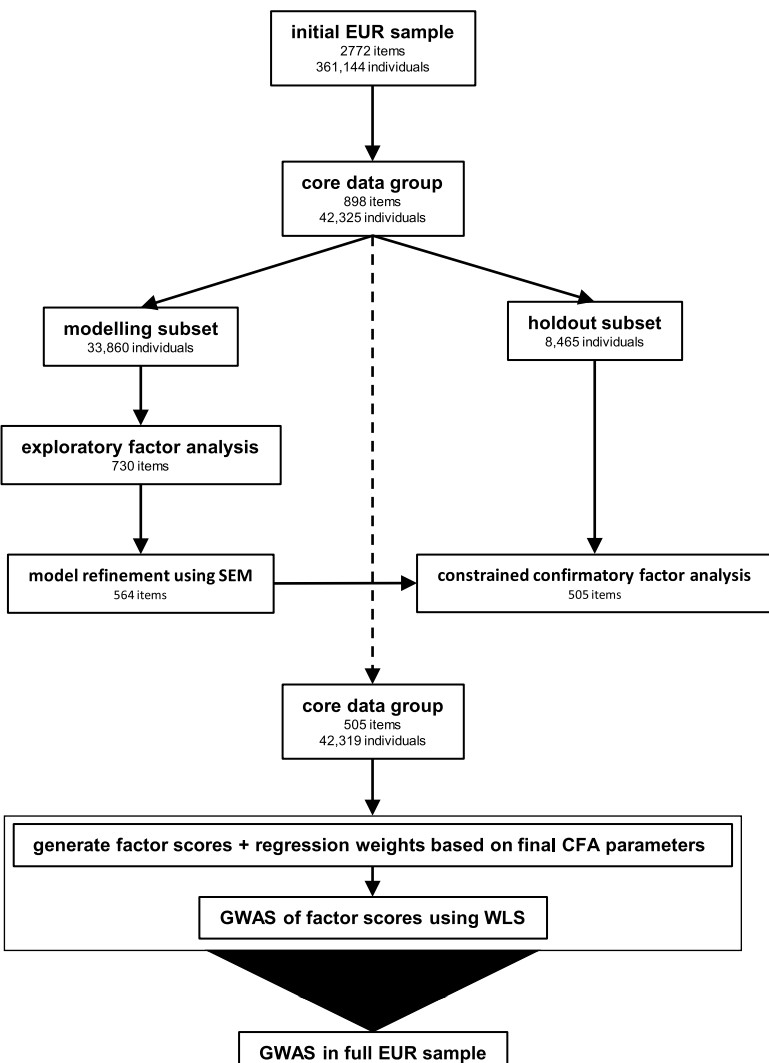

**Extended Data Fig. 1 | Schematic of overall analytic plan.** Displays the outline of analyses performed in the study as well as number of phenotypes and participants at each step.

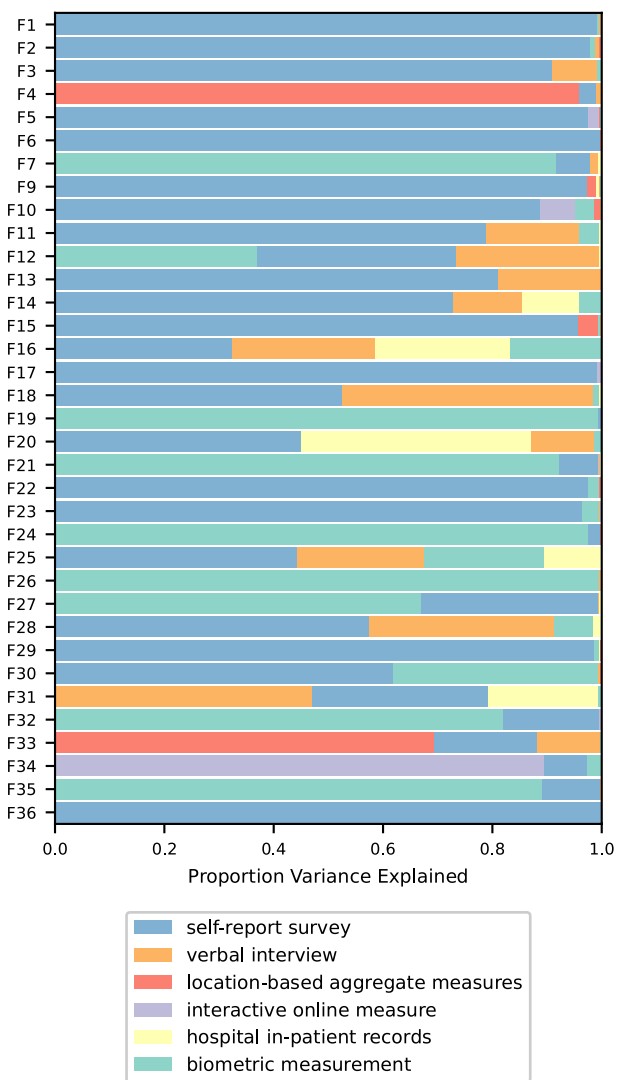

**Extended Data Fig. 2 | Representation of item types across factors.** Horizontal bars represent proportion variance explained in a given factor score by each of 6 major data types in UKB, estimated using hierarchical partitioning. To the left, factors are numbered in order of variance extraction in the exploratory factor analysis.

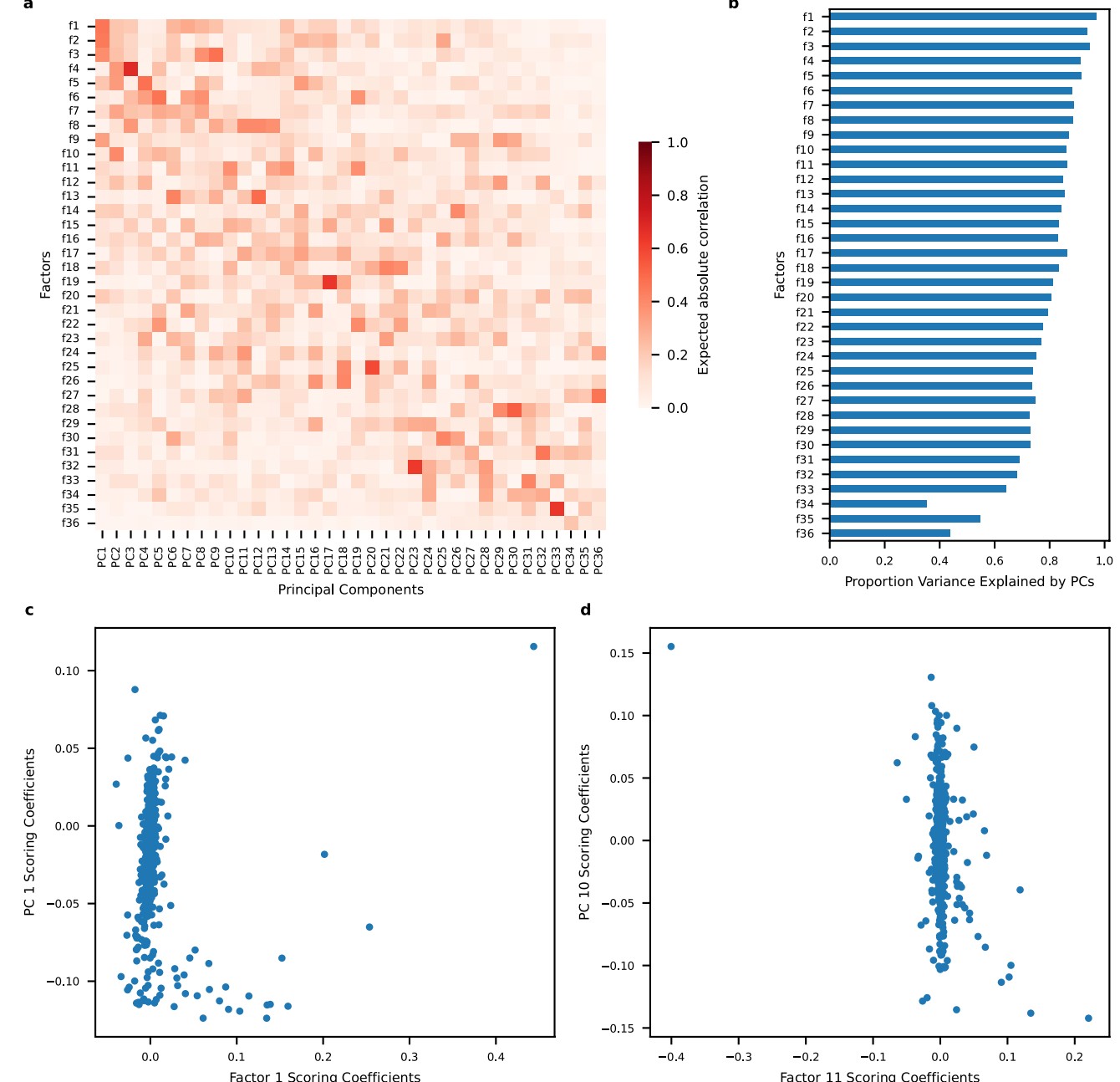

**Extended Data Fig. 3 | Comparison of EFA to PCA. a**) The expected absolute correlations across the 36 EFA factors and principal components. **b**) For each of the 36 EFA factors, the proportion of variance explained by all 36 PCs. **c** and **d**) Per-item scatterplots of scoring coefficients for factors vs. PCs across thematically similar pairs, demonstrating sparser loadings amongst the factor scoring coefficients vs. the PC scoring coefficients.

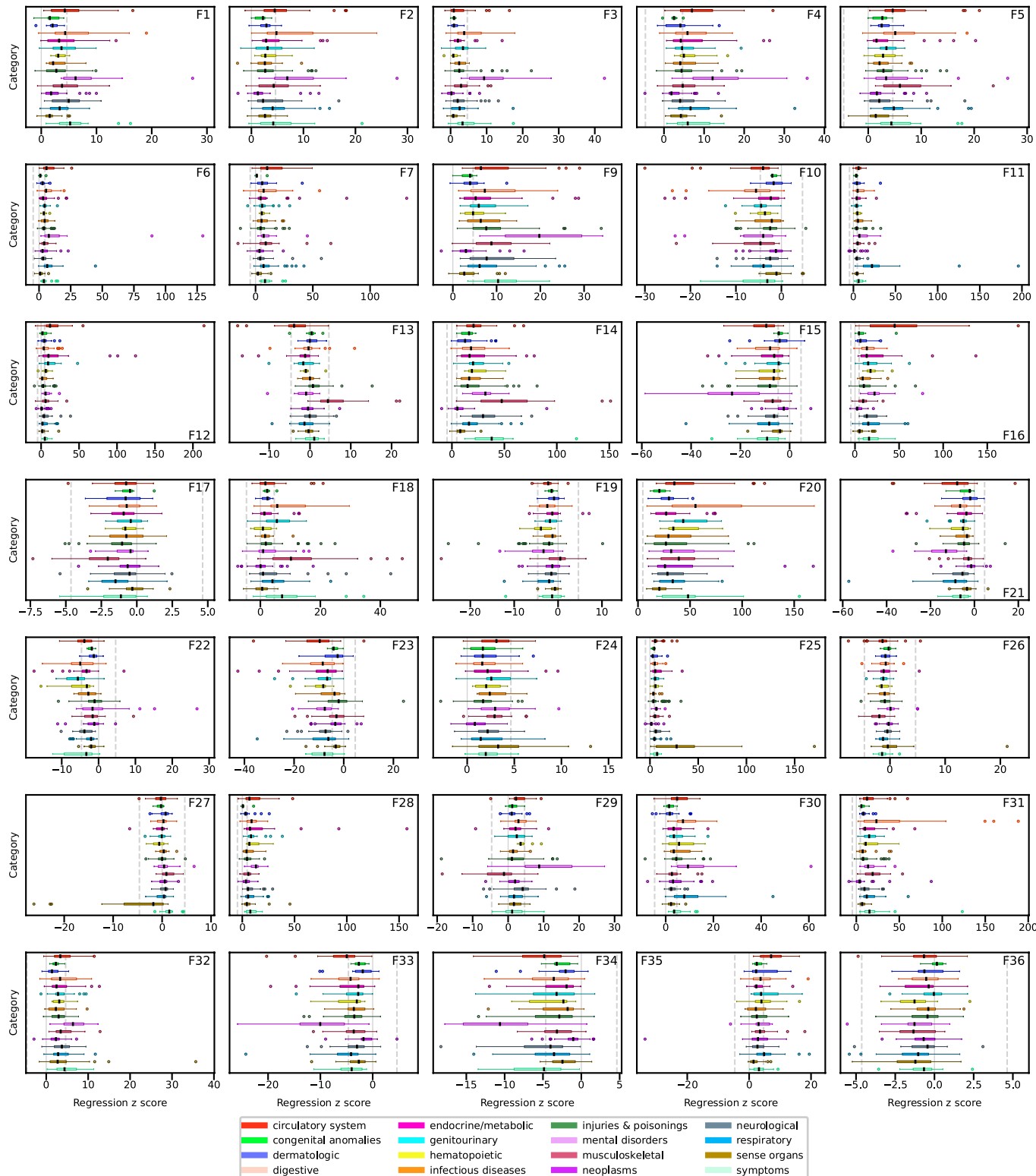

**Extended Data Fig. 4 | Phecode associations by factor.** Box-and-whisker plots are shown for associations with 403 derived medical phecodes grouped by category. These associations are defined as the test statistics (that is, z scores from estimated regression coefficients and Huber-White robust standard errors) for the factor score in a logistic regression model including our standard covariates (that is, first 20 genetic PCs, age, chromosomal sex, age², age-x-chromosomal sex, age²-x-chromosomal sex, and dummy variables representing the assessment centers of origin). Boxes represent the middle quartiles of a factor's test statistics across phecodes within a category, with whiskers extending to 1.5x the interquartile range. Median values per category are indicated by individual black lines inside the boxes. The dotted grey lines represent the critical test statistics for significance at two-sided $p < 0.05$ after correcting for multiple comparisons across all 403 phecodes.

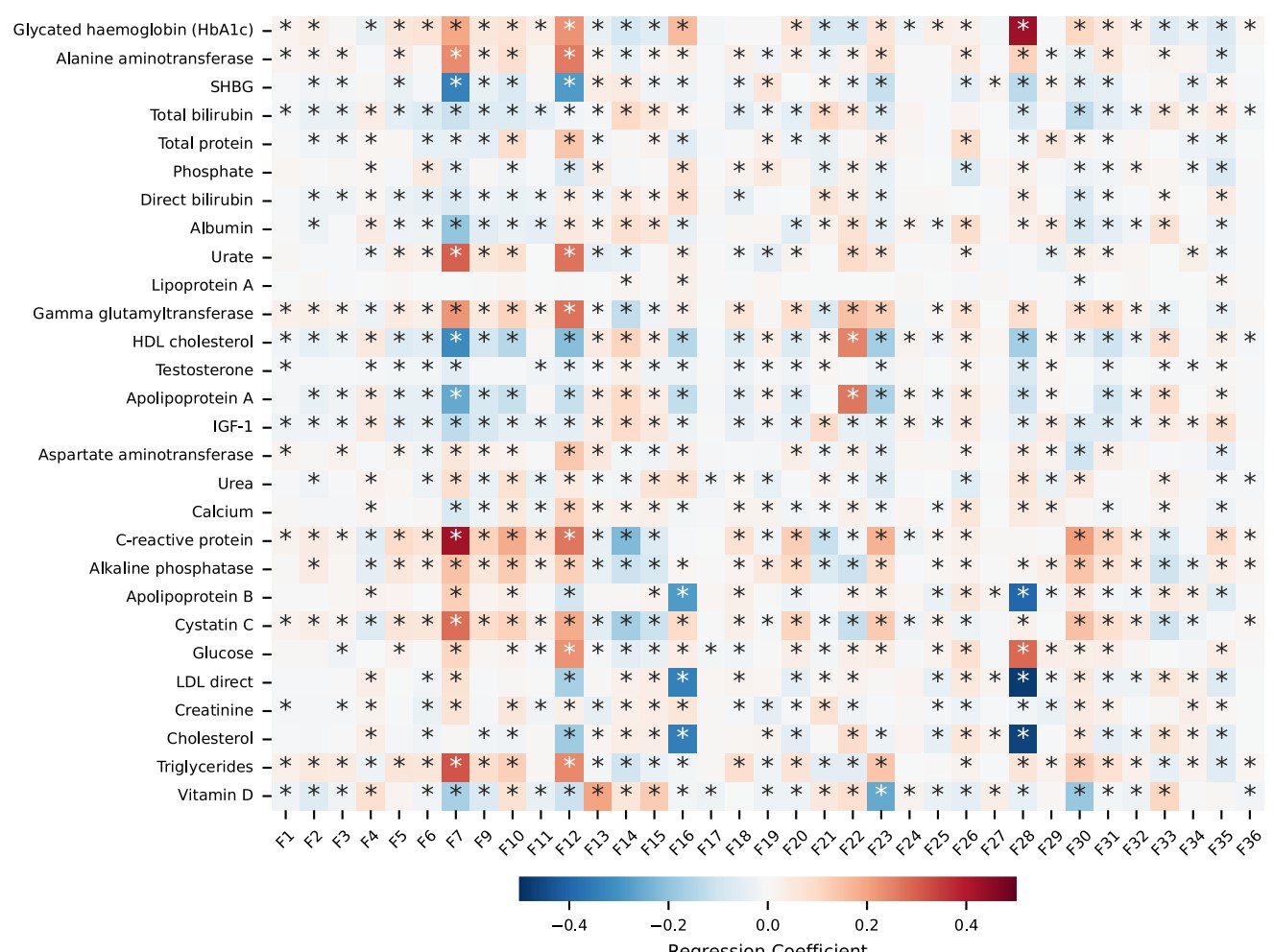

**Extended Data Fig. 5 | Biomarker associations by factor.** Phenotypic associations between factors and 28 biomarkers assayed in UKB. Colors represent the magnitude and direction of correlation, and asterisks (*) indicate which associations remain significant in ordinary least squares regression with Huber-White robust standard errors after correction for multiple testing (that is, two-sided p < 0.05 / (28 biomarkers x 35 factors)).

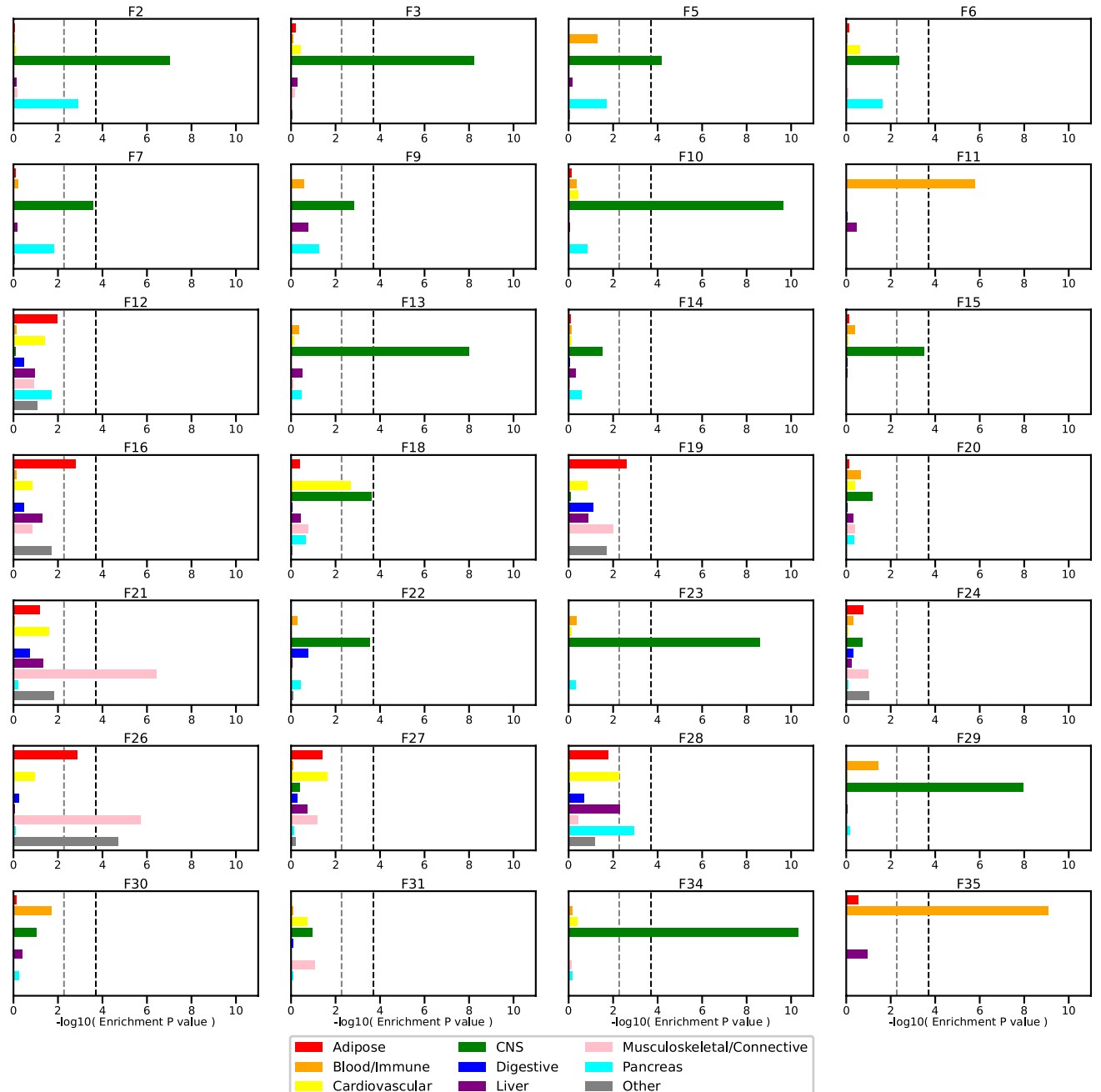

**Extended Data Fig. 6 | Heritability enrichment by cell type group.** Consistent with prior guidelines, only the 28 factors with h2 z > 7 were included in these analyses. Barplots show the -log10(p-value) from the two-sided t-test in partitioned LD score regression testing enrichment of GWAS signal in regions with annotated chromatin marks in cell types from the given group. The light grey dashed line represents the threshold for FDR-corrected significance at 0.05, while the black dashed line represents Bonferroni corrected threshold for 0.05 / (28 factors x 9 cell type groups).

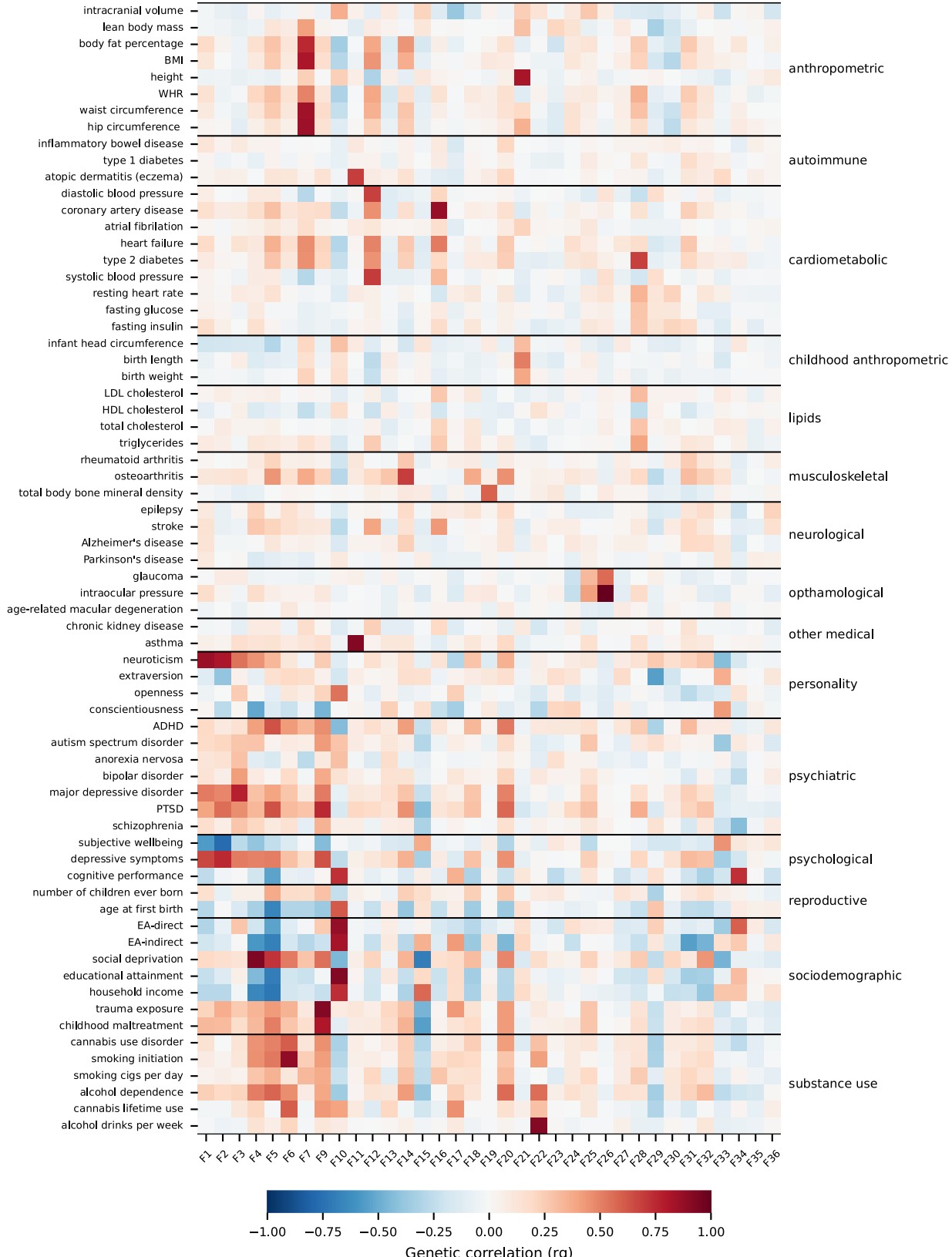

**Extended Data Fig. 7 | Genetic correlations of factors with outside traits.** The heatmap shows the estimated $r_g$ between our 35 factors and 68 selected outside summary statistics. Outside traits are grouped by general category. Color represents the magnitude and direction of genetic correlation.

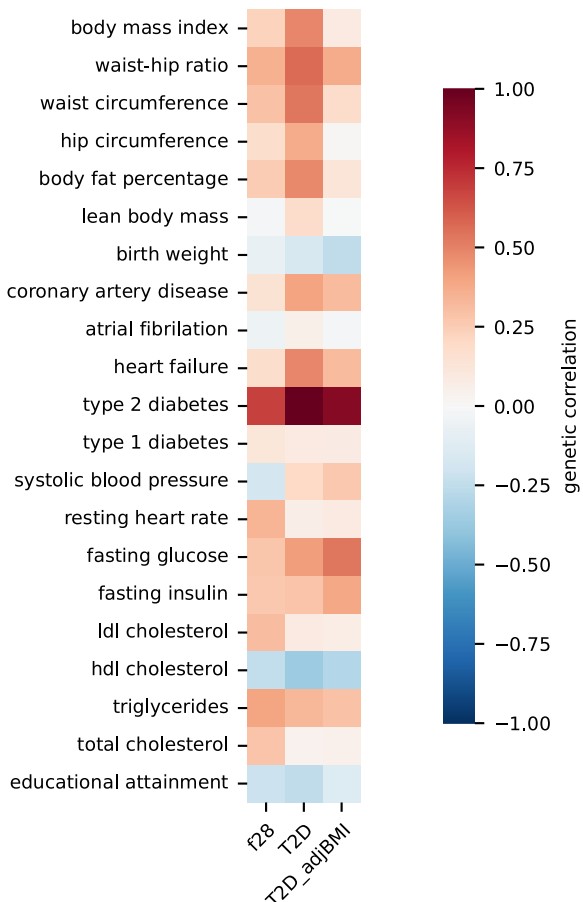

**Extended Data Fig. 8 | Demonstration of the impact of orthogonalization on genetic architecture of Factor 28 versus an outside GWAS of Type 2 Diabetes.** The heatmap shows the estimated $r_g$ between 20 selected outside cardiometabolic summary statistics and 1) our Factor 28, 2) an outside GWAS of Type 2 Diabetes, and 3) an outside GWAS of Type 2 Diabetes adjusted for BMI. Color represents the magnitude and direction of genetic correlation.

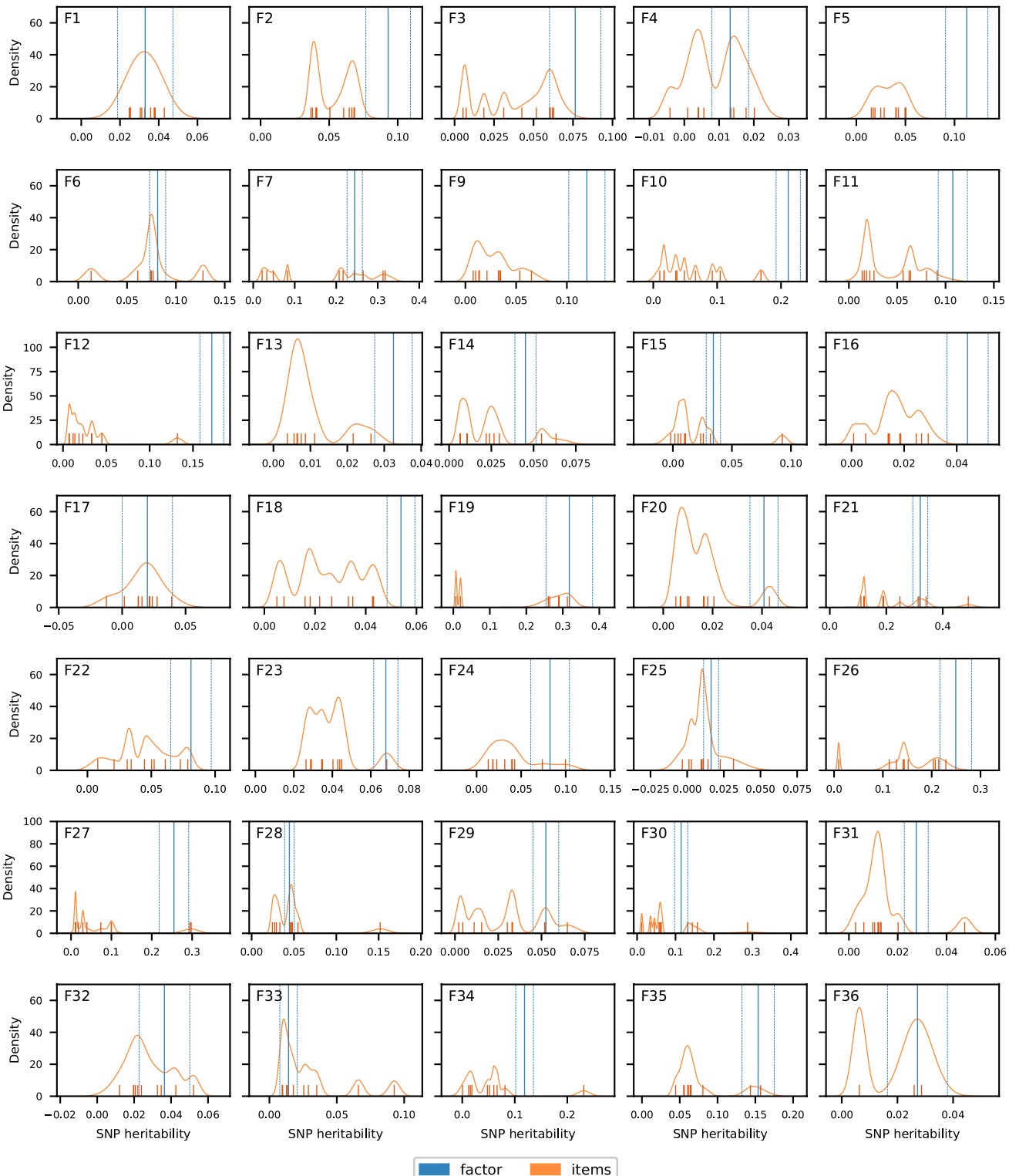

**Extended Data Fig. 9 | Heritability for factors versus top items.** For each factor, density plots showing the distribution of estimated observed-scale SNP-heritability for the top items (orange) compared to the point estimate and 95% confidence interval for the SNP-heritability of the factor (blue). Orange tick marks indicate the point estimates of SNP-heritability for the items.

# Reporting Summary

## Statistics

For all statistical analyses, confirm that the following items are present in the figure legend, table legend, main text, or Methods section.

| n/a | Confirmed | |
|---|---|---|
| ☐ | ☒ | The exact sample size (*n*) for each experimental group/condition, given as a discrete number and unit of measurement |
| ☐ | ☒ | A statement on whether measurements were taken from distinct samples or whether the same sample was measured repeatedly |
| ☐ | ☒ | The statistical test(s) used AND whether they are one- or two-sided<br>*Only common tests should be described solely by name; describe more complex techniques in the Methods section.* |
| ☐ | ☒ | A description of all covariates tested |
| ☐ | ☒ | A description of any assumptions or corrections, such as tests of normality and adjustment for multiple comparisons |
| ☐ | ☒ | A full description of the statistical parameters including central tendency (e.g. means) or other basic estimates (e.g. regression coefficient) AND variation (e.g. standard deviation) or associated estimates of uncertainty (e.g. confidence intervals) |
| ☐ | ☒ | For null hypothesis testing, the test statistic (e.g. *F*, *t*, *r*) with confidence intervals, effect sizes, degrees of freedom and *P* value noted<br>*Give P values as exact values whenever suitable.* |
| ☐ | ☒ | For Bayesian analysis, information on the choice of priors and Markov chain Monte Carlo settings |
| ☒ | ☐ | For hierarchical and complex designs, identification of the appropriate level for tests and full reporting of outcomes |
| ☐ | ☒ | Estimates of effect sizes (e.g. Cohen's *d*, Pearson's *r*), indicating how they were calculated |

*Our web collection on statistics for biologists contains articles on many of the points above.*

## Software and code

Policy information about availability of computer code

| | |
|---|---|
| Data collection | Data consisted of outcomes that were the result of factor analyzing thousands of phenotypes in biobank-scale data, long with genetic data in the form of genotypes. Details on how these factors were extracted and analyzed are available in the manuscript and Methods. Details on how genotypes were obtained in each dataset are available in the Methods section of the manuscript |
| Data analysis | Data analysis was conducted as specified in the Methods section of the manuscript. Hail Version 2.0 was used for GWAS, polygenic scores were generated with PLINK Version 1.9 and LDpred2. Multiple imputation of core data was performed with MICE (Version 3.16.0), factor analyses and other analyses were performed with relevant packages in R Version 4.2.3, including psych (Version 2.4.1) and lavaan (Version 0.6-3) and relevant packages in Python Version 3.13, including statsmodels (Version 0.13.1), numpy (Version 1.26.3), and lifelines (Version 0.26.4), details of which are provided in the text of the Methods and Supplementary Information. All code has been uploaded and is available at https://github.com/ce-carey/ukb-factor-analysis. |

For manuscripts utilizing custom algorithms or software that are central to the research but not yet described in published literature, software must be made available to editors and reviewers. We strongly encourage code deposition in a community repository (e.g. GitHub). See the Nature Portfolio guidelines for submitting code & software for further information.

# Data

Policy information about availability of data

All manuscripts must include a data availability statement. This statement should provide the following information, where applicable:
- Accession codes, unique identifiers, or web links for publicly available datasets
- A description of any restrictions on data availability
- For clinical datasets or third party data, please ensure that the statement adheres to our policy

The GWAS results for our 35 factors are available through the GWAS Catalog accession nos. GCST90309336-GCST90309370 (in chronological order of the factors).

# Research involving human participants, their data, or biological material

Policy information about studies with human participants or human data. See also policy information about sex, gender (identity/presentation), and sexual orientation and race, ethnicity and racism.

**Reporting on sex and gender**

Throughout the analyses, we use only sex (i.e., not gender) and are clear throughout that sex was defined based on chromosomes (i.e., XX is "female", XY is "male"). We use chromosomal sex as a covariate in all analyses because we were interested in population-level trends irrespective of sex.

**Reporting on race, ethnicity, or other socially relevant groupings**

Genetic ancestry was determined using genetic data (using Principal Components analysis). Here we limited our sample to only individuals of European ancestry due to the statistical confounds presented by population stratification, as is standard in the literature. All GWAS and polygenic prediction exercises also controlled for genetic ancestry (the top 20 principal components of the genetic variance-covariance matrix of the genetic data for GWAS analyses and the top 10 for polygenic prediction exercises).

**Population characteristics**

Those included in this study were participants in the UK Biobank, a longitudinal health study of ~500K volunteers between the ages of 40-69 at recruitment between the years of 2006-2010. See the UK Biobank website (https://www.ukbiobank.ac.uk/), Bycroft et al., 2008 and the sections below for more information.

In Add Health, the mean birth year of respondents is 1979 (SD = 1.8), and the mean age at the time of assessment (Wave 4) is 29.0 years (SD = 1.8). All phenotypes included in this study come from Wave 4, the latest wave of Add Health data collection (2007-2009).

**Recruitment**

The UK Biobank (UKB) is a health resource which has the purpose of improving the prevention, diagnosis, and treatment of human disease75. It consists of a prospective cohort of 502,620 men and women aged 40-69 recruited in the years 2006-2010 throughout the United Kingdom. The touchscreen questionnaire is a collection of self-reported information regarding general health, dietary habits, physical activity, psychological and cognitive states, sociodemographic factors, etc. We began with 361,194 unrelated individuals of European genetic ancestry who passed quality control measures (https://www.nealelab.is/uk-biobank/ukbround2announcement).

Add Health originated as an in-school survey of a nationally representative sample of US adolescents enrolled in grades 7 through 12 during the 1994-1995 school year. Respondents were born between 1974 and 1983, and a subset of the original Add Health respondents has been followed up with in-home interviews, which allows researchers to assess correlates of outcomes in the transition to early adulthood.

No

**Ethics oversight**

Use of the UK Biobank data was approved under application 31063. Analysis of the UK Biobank data was reviewed by the Partners HealthCare IRB (Partners Human Research), which determined in expedited review that the project met the US federal criteria definition of "not human subjects research."

Analysis of the Add Health data was reviewed by the Office of Research Subject Protection (OSRP) at the Broad Institute of MIT and Harvard, which determined that the project met US federal criteria for exemption from IRB review (Project #0001 titled "Genetic and environmental factors influencing complex social behavior").

Informed consent was obtained by participants who chose to participate in both the UK Biobank and Add Health studies, and this consent was handled by the teams curating those datasets.

Note that full information on the approval of the study protocol must also be provided in the manuscript.

# Field-specific reporting

Please select the one below that is the best fit for your research. If you are not sure, read the appropriate sections before making your selection.

☐ Life sciences   ☒ Behavioural & social sciences   ☐ Ecological, evolutionary & environmental sciences

For a reference copy of the document with all sections, see nature.com/documents/nr-reporting-summary-flat.pdf

# Behavioural & social sciences study design

All studies must disclose on these points even when the disclosure is negative.

| | |
|---|---|
| Study description | This study used data from the UK Biobank and attempted to better characterize the measured data using dimensionality reduction methods, most specifically factor analysis. Briefly, we performed factor analyses on the phenotypes in the UK Biobank, extracted 35 orthogonal factors using these methods, and then ran follow-up correlational analyses, including genome-wide association studies and polygenic scoring, of our factors. All data are quantitative. |
| Research sample | Using the UK Biobank, we examined 2,772 phenotypes in unrelated individuals with predominantly estimated European genetic ancestry (N=361,144). We performed replication exercises using polygenic scores for 3,414 individuals in the Add Health study. We chose these samples because of the large and required sample size for genetic analyses (UK Biobank) and for existence of well-phenotyped replication data (Add Health). the UK Biobank is not a nationally representative study, while the Add Health study is a US-based nationally representative study. |
| Sampling strategy | Analytic samples were decided by using the samples of subsamples that had the largest N for a given outcome under study. This strategy was chosen, because large samples are prioritized to have enough statistical power to isolate small genetic associations. This is common practice in the field. |
| Data collection | The UK Biobank (UKB) is a health resource which has the purpose of improving the prevention, diagnosis, and treatment of human disease75. It consists of a prospective cohort of 502,620 men and women aged 40-69 recruited in the years 2006-2010 throughout the United Kingdom. The touchscreen questionnaire is a collection of self-reported information regarding general health, dietary habits, physical activity, psychological and cognitive states, sociodemographic factors, etc. We began with 361,194 unrelated individuals of European genetic ancestry who passed quality control measures (https://www.nealelab.is/uk-biobank/ ukbround2announcement).<br><br>Add Health originated as an in-school survey of a nationally representative sample of US adolescents enrolled in grades 7 through 12 during the 1994-1995 school year. Respondents were born between 1974 and 1983, and a subset of the original Add Health respondents has been followed up with in-home interviews, which allows researchers to assess correlates of outcomes in the transition to early adulthood. In Add Health, the mean birth year of respondents is 1979 (SD = 1.8), and the mean age at the time of assessment (Wave 4) is 29.0 years (SD = 1.8). All phenotypes included in this study come from Wave 4, the latest wave of Add Health data collection (2007-2009). |
| Timing | The UK Biobank is a prospective cohort of 502,620 men and women aged 40-69 recruited in the years 2006-2010 throughout the United Kingdom.<br><br>Add Health originated as an in-school survey of a nationally representative sample of US adolescents enrolled in grades 7 through 12 during the 1994-1995 school year. Respondents were born between 1974 and 1983, and a subset of the original Add Health respondents has been followed up with in-home interviews, which allows researchers to assess correlates of outcomes in the transition to early adulthood. |
| Data exclusions | We use only individuals of European ancestry due to the statistical confounds presented by population stratification, as is standard in the literature. Individuals were differentially included/excluded at various steps of the analysis due to data availability; this is extensively documents in the Methods and Supplementary Information. |
| Non-participation | Participants were allowed to drop out of the study, or to not respond to any individual survey items, at any point, for both the UK Biobank and Add Health samples. We considered missingness extensively in the development of our analytic methods and provide details with the manuscript. However, we can of course account for all bias due to non-participation and/or ascertainment. |
| Randomization | No randomization was performed; the study was exploratory/cross-sectional/observational. |

# Reporting for specific materials, systems and methods

We require information from authors about some types of materials, experimental systems and methods used in many studies. Here, indicate whether each material, system or method listed is relevant to your study. If you are not sure if a list item applies to your research, read the appropriate section before selecting a response.

## Materials & experimental systems

| n/a | Involved in the study |
|---|---|
| ☒ | ☐ Antibodies |
| ☒ | ☐ Eukaryotic cell lines |
| ☒ | ☐ Palaeontology and archaeology |
| ☒ | ☐ Animals and other organisms |
| ☒ | ☐ Clinical data |
| ☒ | ☐ Dual use research of concern |
| ☒ | ☐ Plants |

## Methods

| n/a | Involved in the study |
|---|---|
| ☒ | ☐ ChIP-seq |
| ☒ | ☐ Flow cytometry |
| ☒ | ☐ MRI-based neuroimaging |

## Plants

Seed stocks

n/a

Novel plant genotypes

n/a

Authentication

n/a

