## [Peer Review File · Nature Human Behaviour]

Peer Review Information

Journal: Nature Human Behaviour

Manuscript Title: Principled distillation of UK Biobank phenotype data reveals underlying structure in human variation

Corresponding author name(s): Caitlin E. Carey, Robbee Wedow, and Raymond K. Walters

Reviewer Comments & Decisions:

Decision Letter, initial version:

18th January 2023

Dear Dr Carey,

Thank you once again for your manuscript, entitled "Principled distillation of multidimensional UK Biobank data reveals insights into the correlated human phenome", and for your patience during the peer review process.

Your Article has now been evaluated by 3 referees. You will see from their comments copied below that, although they find your work of potential interest, they have raised quite substantial concerns. In light of these comments, we cannot accept the manuscript for publication currently, but we would be interested in considering a revised version if you are willing and able to fully address reviewer and editorial concerns.

We hope you will find the referees' comments useful as you decide how to proceed. If you wish to submit a substantially revised manuscript, please bear in mind that we will be reluctant to approach the referees again in the absence of major revisions.

In particular, we feel that the following points are especially important:

1. Reviewer #2 questions the justification for applying FA rather than PCA to these data, while Reviewer #3 points out that several different dimensionality reduction methods have been used in the genetics literature. In revision, please ensure that robust justification for the application of FA is provided, as well as an explanation of what insights FA can deliver that would not be possible with other dimensionality reduction techniques.
2. Reviewer #3 questions the depth of insight that your existing exploratory analyses provide, especially as your results are not compared with those of other dimensionality reduction techniques. In revision, we ask that you also demonstrate the value of your approach in providing mechanistic or

genetic insight, for example by Reviewer #3's suggestion of "putative hypotheses dug into, tested and validated using external data sets."

3. Reviewer #2 raises several technical concerns about the specific application of EFA and CFA that should be carefully addressed.

4. Reviewer #3 points to the fact that your sample is limited to European ancestry. Please acknowledge this limitation more prominently, e.g. in the abstract.

If you wish to submit a suitably revised manuscript we would hope to receive it within 4 months. I would be grateful if you could contact us as soon as possible if you foresee difficulties with meeting this target resubmission date.

- Include a "Response to the editors and reviewers" document detailing, point-by-point, how you addressed each editor and referee comment. If no action was taken to address a point, you must provide a compelling argument. When formatting this document, please respond to each reviewer comment individually, including the full text of the reviewer comment verbatim followed by your response to the individual point. This response will be used by the editors to evaluate your revision and sent back to the reviewers along with the revised manuscript.
- Highlight all changes made to your manuscript or provide us with a version that tracks changes.

[REDACTED]

Thank you for the opportunity to review your work. Please do not hesitate to contact me if you have any questions or would like to discuss the required revisions further.

Sincerely,

[REDACTED]

Senior Editor
Nature Human Behaviour

REVIEWER COMMENTS:

Reviewer #1:

Remarks to the Author:

In this manuscript, Carey and colleagues conducted a large factor analysis, resulting in 35 latent factors. This work is impressive in scope and provides important insights into the interwoven nature of the human phenome.

I have the following comments on the manuscript:

- 1) The UKB is not representative of its underlying sampling population. The resulting volunteer bias is a known source of confounding. I imagine that correcting for this bias would change the factor structure considerably. How should we interpret the results in this light?
- 2) The orthogonal nature of the factors makes them quite complex to grasp. I think the main text could use some clarification on how to interpret, and not to interpret, the factors. Some trait-groups are known to be related, how should readers interpret the factors underlying those groups, given the orthogonal nature? One or two simple examples would help.
- 3) The authors aim to disentangle SES, comparing their 3 SES factors with proxies of prior GWAS. Given the known genetic overlap of SES with most human traits (e.g., psychiatric disorders), I think it would be interesting to extend Figure 5 such that all factors can be compared to SES.

Reviewer #2:

Remarks to the Author:

Comments to the Authors

This study reports a unique application of factor analysis (FA) to the UK biobank data. As mentioned by the authors, this is the "largest factor analysis to date" (in terms of the number of observed variables). This study "provides an important first step toward better embracing the full and complex measured phenome to power discovery for human health and well-being". While interesting, I have some concerns and comments regarding the FA method in the current application.

1. Factor Analysis (FA) vs Principal Component Analysis (PCA)

The authors need to provide a strong justification for using FA in this application. As pointed out by the authors (page 3), "a host of data reduction methods have been well established and used at scale". It is not clear to me why FA is "optimized" for the purpose of the current study. Given that the goal is not "measuring" latent construct of interests- instead, the purpose is to reduce dimensionality and to get a composite (i.e., the weighted linear combination of items); therefore, it seems to me PCA

is a more proper technique to use (which is more commonly used in bioinformatics anyway).

2. EFA

First, it is not clear to me why WLS/GLS/MINRES/ULS estimators are used, instead of maximum likelihood (ML). ML is the default estimation method in many SEM software and can handle missing data.

Second, related to the above point, the authors used pairwise deletion for handling missing data, which is a suboptimal approach, compared to methods like full information maximum likelihood (FIML) or multiple imputation (MI).

Third, if I understand correctly, the authors used the Pearson correlation matrix as input for EFA, assuming all the variables are continuous. However, when conducting CFA, the authors considered different types of variables. As mentioned on page 35, "this [CFA] allows us to ...more appropriately [model] the covariance structure of the diverse variable types (i.e., binary, ordinal, continuous) ... with more robust handling of missingness". I do not think this is a "difference" between EFA and CFA- a mixed type of variables and modern methods for handling missing data (see point #2) are also available under EFA.

Forth, determining the number of factors and the interpretation of the latent factors are critical steps in EFA. I do not think the authors provide enough information or reasonable justifications for both steps. 1)The authors kept increasing the number of factors until "mathematical error"- this is not a common way of determining the number of factors- at least not the way EFA is designed for. Again, if the goal is simply reducing dimensionality and explaining maximum variances, PCA is a more proper approach. 2)It is also not clear to me how the authors created the "labels" for the 35 factors in the final model. From a psychometric perspective, this step needs to be carefully guided by item analysis and substantive theories. It is not clear why the final 35 constructs are "interpretable constructs" under the labels provided by the authors.

Finally, as recognized by the authors, the factors in the final model should not be orthogonal (e.g., between F2- depressive symptomatology and F3- clinical anxiety and depression). It is not clear why not use oblique rotation for EFA and allow inter-factor correlations for CFA.

3. CFA

First, the authors need to be more specific regarding the estimation method used for CFA. When using diagonal weighted least squares (DWLS), lavaan also provides 'robust' variants (e.g., WLSM and WLSMV) to better correct the standard errors and test statistics. It is not clear if the authors used any robust version of DWLS- if not, why.

Second, when applying FA (EFA and CFA), it is standard to report all the test statistics (chi-square test with df and p-value) and fit indices (RMSEA, CFI, TLI, SRMR, etc.). The authors also mentioned using fit indices such as CFI, SRMR, etc., however, the only fit information reported was RMSEA.

Third, related to the above point, when using DWLS estimators (WLSMV), the methods for computing fit indices are yet to be developed. For example, lavaan typically reports RMSEA, RMSEA.scaled, or RMSEA.robust, it is not clear which RMSEA the authors reported. Also, since multiple imputation was used, it is not clear which procedure the authors used to aggregate the RMSEA across imputed

datasets.

4. Factor scores as measurements

As a goal for the current study, the latent factors (factor scores) were used to “improve measurement of health-related behavior”. From a psychometric perspective, I recommend the authors compute and report reliability measures for the scores.

Reviewer #3:

Remarks to the Author:

Background

This manuscript presents the results of applying factor analysis (FA) to a subset of phenotypes captured in UK Biobank. There is a mixture of phenomenological analyses (what do the derived factors capture and how do they relate to other measurements? - e.g. mortality) and genetic analyses (what is the value of carrying out genetic analysis of factors versus individual measurements? - e.g. numbers of variants discovered associated with factors compared to component phenotypes). Many of the analyses are exploratory - and the conclusions therefore either vague (e.g. “This approach provides an important first step toward better embracing the full and complex measured phenome to power discovery for human health and wellbeing” or “...SES is a multidimensional construct”), and/or come with strong caveats, though there are several interesting observations within the paper.

Advances

Although I have several issues that I would like to see addressed (see below), I’d like to start with highlighting what I see as the advances of the paper. I see the novel insights as:

1. A proof-of-concept that high-dimensional biobank-style data can be represented into lower-dimensional summaries in such a way that generates novel insights (e.g. genetic loci not previously identified). This is not the first paper using dimension-reduction techniques on biobank-style data (e.g. Holden et al. 2011 - DOI: 10.1186/1478-7954-9-15, Ronaldsen et al. 2022 - doi:10.1080/13607863.2021.2017847), but it is uniquely ambitious in scope, combining both medical and non-medical data types.
2. A demonstration that - despite this - more genetic loci are associated with either constituent phenotypes than factors per se (which is intriguing).
3. An observation that genetic risk for factors can be strongly correlated, even when - at a phenotypic level - the factors are defined as being independent (orthogonal).

There are, of course, many specific observations about specific factors and their associations (e.g. around physical activity or trauma). My high-level sense is that - for all of these - the analyses are too superficial to really add to the literature on the specific domains (for example, there is quite a literature on genetic risk factors for PTSD and other traumas - and while the work here acknowledges this, it does not explain what the added insights are).

Major points

1. The highest level issue I have is that the work is - fundamentally - exploratory. Not asking a specific question, but rather applying a single, specific technique (factor analysis) to a complex data set and exploring what is found. I don't find this uninteresting, but it's not really question-driven research. There are interesting research questions within the domain (for example, what's the best way to represent biobank data to generate genetic insights or to enable prediction), but this is not the approach taken. Although detailed notes are provided about exactly how the factor analysis was applied, it is just one of many possible ways of representing such data and comes with specific assumptions (e.g. orthogonality) that may not be appropriate for the problem. My (relatively quick) reading of the literature, suggests that others have tried different approaches to the problem (e.g. multiview learning, LDA-derived approaches, PCA) and I would have enjoyed reading more about whether different approaches come to the same or different conclusions.

2. A related issue is that where findings are consistent with the literature, this is stated as confirmation that the approach works well, but where there are novel findings (e.g. BMI/gaming), these are not followed up. I find this irritating as a reader - if you want me to believe that this approach is going to generate insights that I wasn't able to get from reading the literature, or simple analyses like correlation, then I would like to see putative hypotheses dug into, tested and validated using external data sets. The work on Factor 23 (physical activity) perhaps comes closest to this, but I would, for example, be interested in a much more detailed analysis of this (pulling apart its causes and consequences, potentially using MR to look at causality, asking directly what the best predictor of reduced mortality is, etc.). And also a much deeper analysis of the contributions here compared to other work.

3. As noted above, I do think there are two findings related to genetic factors that are potentially interesting. The first is the finding of variants that affect factors - but constituent phenotypes to a lesser extent. It would be interesting to dig into whether there are differential enrichments for such variants within domain (e.g. wrt Factors 11 or 16), or whether they are just smaller effects of a similar nature? Conversely, what are the enrichments for variants most strongly associated with single phenotypes than factors? Such a differential analysis could provide insights into the genetic architecture of complex traits that are only possible through biobank-style data.

4. The other aspect of interest - though it might be more of a statistical artefact than biology - is the presence of strong genetic correlation between factors, even when they are phenotypically orthogonal (e.g. factors 5, 10 and 15 in Fig 5). I'm not really sure what this tells me about genetic architecture, but it deserves some thought.

Detailed issues

Please note that some of these are potentially quite important - but they are more detailed.

5. There is little discussion of previous attempts to use factor analysis and other dimensionality-reduction techniques when analysing biobank-style data (or at least multimorbidity data).

6. While the impact of factor loadings on mortality is assessed, I expect that many of these may vary in terms of the age and sex distribution. Given that age and sex impact mortality, I would expect to see this analysis removing the effects of such covariates.

7. It's not clear to me that heritability estimates for factors and individual items are directly comparable given that they are measured on different scales. Please consider whether this needs to be addressed.

8. When there is comparison of genetic correlation to phenotypes from external GWAS, I note that these often contain UK Biobank as a constituent cohort - and often the dominant one. I think these should be compared only to cohorts within UK Biobank - for example, when a factor has a strong overlap with EA, it is not surprising that it has a strong genetic correlation with EA measured in the same cohort.

9. I don't think it's justifiable really to only focus on European ancestry individuals. I understand the desire for reducing heterogeneity, but kind of the point of the paper is to embrace the full breadth of human biology and I don't see how you can exclude people based on their race, ethnicity or genetic ancestry on that basis. It feels inappropriate to me.

10. I couldn't work out what the units of Extended Data Fig. 3 are - please label.

Author Rebuttal to Initial comments

REVIEWER COMMENTS AND RESPONSES:

Reviewer #1:

Remarks to the Author:

In this manuscript, Carey and colleagues conducted a large factor analysis, resulting in 35 latent factors. This work is impressive in scope and provides important insights into the interwoven nature of the human phenome.

I have the following comments on the manuscript:

Comment 1: The UKB is not representative of its underlying sampling population. The resulting volunteer bias is a known source of confounding. I imagine that correcting for this bias would change the factor structure considerably. How should we interpret the results in this light?

Response: We thank the reviewer for this important comment. We agree that the identified factor structure is only representative of the study population, which in the case of UKB does not match the

general UK population (or, of course, the global population). Recent work on ascertainment and nonresponse biases in UKB (including by this group, ex: PMIDs 33888908 and 37386106) should enable future research to evaluate the impact of these study biases more directly. We agree wholeheartedly that such considerations need to be discussed within the main text. In the Discussion, we have now added the following text:

Participants at the core of this study are, for example, unlikely to be representative of the global population. Analyses were restricted to UKB participants of predominately estimated European genetic ancestry, thus limiting generalizability of results, particularly as it relates to genetic inference (97). In addition, UK Biobank participants are known to be nonrepresentative of the UK population as a whole, with documented ascertainment and participation biases (4,98–100). Even among those participating in the study, individuals providing more complete initial responses and contributing to later optional follow-up surveys are known to be, in general, healthier and more educated (101,102). This trend is reflected in our own core, low-missingness data group being substantially more likely to report having completed college or university (45.7% core group, 30.7% non-core group; $\chi^2=3816.0$, $p<0.001$). While polygenic score analyses in Add Health suggest that genetic signal captured in the fitted factors at least partially generalizes to individuals of European ancestry in the United States, identifying which constructs elucidated in this study are most robustly maintained across different sociopolitical, cultural, and diagnostic contexts requires future work in additional cohorts.

We have also added polygenic score analyses to evaluate how our results generalize to another, US-based cohort, the National Longitudinal Study of Adolescent to Adult Health (Add Health). These analyses are included throughout the manuscript, interwoven into our vignettes.

Comment 2: The orthogonal nature of the factors makes them quite complex to grasp. I think the main text could use some clarification on how to interpret, and not to interpret, the factors. Some trait-groups are known to be related, how should readers interpret the factors underlying those groups, given the orthogonal nature? One or two simple examples would help.

Response: We fully agree that the orthogonalization makes the interpretation more nuanced. To make this distinction clearer within the main text, we have now explicitly addressed this comment within the Results and Discussion sections.

Within the Results section, we now include the following example text:

It is important to note however that interpretation of the factors relies on their forced orthogonality. Returning to the medical domain, Factor 28 captures diabetes diagnoses and responsive lifestyle changes such as altering one's diet and reducing sugar intake. Had factors not been modeled as independent, Factor 28 would likely share variance with Factors 12 and 16 above, as well as with Factor 7, which captures BMI and adiposity; each of these factors captures aspects of the well-known "metabolic syndrome" (42). In the context of our model, Factor 28, and each factor more generally, is most accurately interpreted as representing the remaining covariance of the items within it (e.g., diabetes diagnosis and treatment), once accounting for the variance explained by the other related (e.g., cardiometabolic), and unrelated, factors.

...

Comparison of Factor 28 to a prior GWAS of type 2 diabetes (51), however, reveals imperfect capture of the clinical definition ($r_g=0.68[0.02]$, $\text{Intercept}_{G_{\text{cov}}}=0.36[0.01]$; UKB-excluded $r_g=0.70[0.03]$; **Extended Data Figure 9**), likely due to its modelled orthogonality from other factors capturing cardiometabolic constructs. More specifically, the factor has higher genetic overlap with cholesterol measures (e.g., total cholesterol [52] $r_g=0.29[0.04]$ vs. $0.04[0.03]$) but lower overlap with BMI (53) ($r_g=0.23[0.03]$ vs. $0.49[0.03]$) and an inverse correlation with blood pressure (54) ($r_g=-0.18[0.02]$ vs. $0.20[0.02]$), reflective of its inclusion of high cholesterol in its factor definition and its forced independence of Factors 7 and 12 described above.

Further, we include the corresponding Extended Data Figure (elevated to an Extended Data Figure, but previously in the Supplement):

Extended Data Figure 9. Demonstration of the impact of orthogonalization on genetic architecture of Factor 28 versus and outside GWAS of Type 2 Diabetes. The heatmap shows the estimated r_g between 20 selected outside cardiometabolic summary statistics and 1) our Factor 28, 2) an outside GWAS of Type 2 Diabetes, and 3) and outside GWAS of Type 2 Diabetes adjusted for BMI. Color represents the magnitude and direction of genetic correlation.

We then further highlight this limitation within the Discussion:

Adapting factor analysis to permit execution at the scale and structure of UK Biobank also forced us to make analytic decisions that introduce a number of limitations. Most prominently, we required orthogonality across the factors to reduce the number of parameters to be estimated and substantially simplify matrix algebra operations. As such, the variance captured by each factor is forced to be independent, complicating interpretation; specifically, each factor must be viewed as representing the covariance structure of the items within it, once accounting for covariance modeled by the other factors. Caution must therefore be exercised in interpreting relationships within and between factors, since this independence is an artifact of the modeling approach.

Comment 3: The authors aim to disentangle SES, comparing their 3 SES factors with proxies of prior GWAS. Given the known genetic overlap of SES with most human traits (e.g., psychiatric disorders), I think it would be interesting to extend Figure 5 such that all factors can be compared to SES.

Response: We agree with the reviewer that additional genetic relationships with SES would be of interest to NHB readers and now reference results of these analyses within the Main Text, with more detail provided within the Supplement. We have taken the 6 external SES GWAS summary statistics and looked at genetic correlations with the rest of our factors.

As the reviewer suspected, they revealed a wide range of associations across the phenome, which we now report in a Supplementary Figure:

Supplementary Figure 11. Genetic overlap between prior GWAS of SES indicators and all factors. All genetic associations are flipped to be in the direction reflecting greater SES for consistency (e.g., “Social Deprivation” becomes “Social Enrichment”). Color of each box within the heatmap indicates the strength and directionality of genetic overlap across the two corresponding phenotypes. EA: Educational Attainment.

Given space limitations and the addition of polygenic score analyses (see below) to the SES vignette within our Results section, we mention these results briefly in the Main Text:

Associations between prior SES GWAS and other factors within our model reveal that Factors 5, 10, and 15 alone do not fully capture SES, with numerous other residual correlations further reflecting the complex interconnections between SES measures and health (**Supplementary Text; Supplementary Figure 11**).

And provide more detailed examples within the Supplementary Text:

Broader comparisons with the prior SES GWAS indicate that our reduced-rank representation of the phenome captured by UKB does not fully capture genetic associations with SES in only these 3 factors (i.e., Factors 5, 10, and 15; **Supplementary Figure 11**). We observe, for example, strong genetic correlations of Factor 4 and Factor 33, containing items related to urbanicity and living near London, respectively, with GWAS of regional social deprivation⁶⁷ (Factor 4 $r_g=0.93[0.08]$; Factor 33 $r_g=-0.47[0.09]$). The results also reflect how our model of uncorrelated factors attempts to partition signal for different aspects of conventional SES measures; for example, non-parental effects on educational attainment⁶⁸ are genetically correlated not only with Factors 5 and 10, but also with Factor 34, which contains items related to cognition and processing speed ($r_g=0.60[0.17]$). Residual genetic correlation is also observed between parental effects on educational attainment and Factors 20 (e.g., severe, life-threatening illness, $r_g=-0.45[0.10]$), 31 (e.g., gastrointestinal issues, $r_g=-0.55[0.11]$), and 32 (e.g., hearing difficulties, $r_g=-0.47[0.14]$), and between social deprivation and Factors 6 (e.g., smoking and associated risk behaviors, $r_g=0.52[0.05]$) and 9 (e.g., trauma, $r_g=0.55[0.06]$), further reflecting the complex interconnections between varied SES measures and health.

We also now, in response to additional reviewer comments, have extended our SES analyses via polygenic scores in the external Add Health dataset, in a way we think could further contribute to understanding of the differences between these factors.

We have added the following text to the SES vignette regarding the polygenic scores analyses:

Polygenic score analyses of the SES factors conducted in the independent Add Health dataset further solidify these key distinctions and provide evidence of generalization beyond the UKB sample (**Supplementary Table 7; Figure 4C**). The Factor 10 PGS outperforms those of Factors 5 and 15 in its associations with years of education ($R^2=7.7\%$), college completion (pseudo- $R^2=5.5\%$), and verbal cognition ($R^2=4.8\%$). In contrast, Factor 5 scores are the best predictors of income ($R^2=5.3\%$) and high-paying vs. low-paying jobs (pseudo- $R^2=5.7\%$), while Factor 15 scores are most associated with neighborhood satisfaction ($R^2=3.9\%$). Not only do these findings demonstrate replication of our results in an independent, non-UKB sample, but the level of differentiation in prediction articulates a clear disentangling of factors related to SES.

And the corresponding figure panel:

Figure 4B. Associations between polygenic scores derived from the SES factors and SES-related items in an outside cohort (AddHealth). Error bars represent 95% bootstrapped confidence intervals. EA: Educational Attainment.

More generally, this excellent comment from Reviewer 1 forced us to reexamine the ordering of our vignettes. While we originally elevated the vignette on medical phenotypes to the first position, we realized that our paper perhaps contributes more interesting general insights in the domains of social science and behavioral outcomes, and that these findings would be of particular interest to the readership of NHB. We now elevate the SES vignette to the first position, and have instead used pieces of the former medical domain vignette to highlight more expected recapitulation of known nosology throughout.

Reviewer #2:

Remarks to the Author:

Comments to the Authors

This study reports a unique application of factor analysis (FA) to the UK biobank data. As mentioned by the authors, this is the “largest factor analysis to date” (in terms of the number of observed variables). This study “provides an important first step toward better embracing the full and complex measured phenome to power discovery for human health and well-being”. While interesting, I have some concerns and comments regarding the FA method in the current application.

Comment 1: Factor Analysis (FA) vs Principal Component Analysis (PCA)

The authors need to provide a strong justification for using FA in this application. As pointed out by the authors (page 3), “a host of data reduction methods have been well established and used at scale”. It is not clear to me why FA is “optimized” for the purpose of the current study. Given that the goal is not “measuring” latent construct of interests- instead, the purpose is to reduce dimensionality and to get a composite (i.e., the weighted linear combination of items); therefore, it seems to me PCA is a more proper technique to use (which is more commonly used in bioinformatics anyway).

Response: The reviewer’s point is very well taken, and we have now attempted to be more explicit in the main text about our choice of FA vs. other dimensionality reduction techniques, especially PCA.

More generally, we have taken this point to heart *throughout* the paper. Much of the text in our initial submission was oriented around defending FA as the “best” choice of dimensionality reduction technique. However, this was a misrepresentation of the original analyses, which instead sought to adapt factor analytic methodology to biobank scale, and detail findings of the underlying structure captured in the model and additional potential insights gained from combining resulting factor scores with additional registry and genetic data. We are therefore very grateful for this reviewer’s comment, which has allowed us to relax our language to more appropriately reflect what our goal was as a scientific team.

More specifically, we have removed language around FA being more “optimized” for our use case and instead have clarified the ways in which FA is particularly *suited* to our purposes. Below we detail a number of changes we have made to the paper to provide further clarity:

In the Introduction, we now include a number of references to prior publications that have used dimensionality reduction techniques in the context of large-scale biobanks, both on phenotypic data and genetic results:

Dimensionality reduction is a common task in many domains, with a recent proliferation of methods having been applied to biobank-scale data. Principal component analysis (PCA), for example, provides a lower-dimensional representation of the strongest axes of variation in a dataset. It has been leveraged to, for instance, identify dimensions of genetic ancestry in genotype data (8), extract features from individual biobank questionnaires (9), and identify sets of genetic variants with similar patterns of association across thousands of genome-wide association studies (GWAS)(10). Other data reduction approaches prioritize identifying correlated sets of variables across data types (11), modeling latent classes (12), or creating lower-dimensional representations for visualization purposes (13,14). Deep learning methods such as autoencoders and transformers have been used to integrate 'omics data across modalities (15,16), and to extract relevant features from electronic health records (EHRs) (17,18), respectively.

Biobank analyses have devoted relatively less attention to factor analysis, an approach commonly used in the social sciences that models the observed correlation between variables as arising from one or more shared continuous latent (unobserved) factors (19). Factor analysis has the benefit of being model-based, facilitating more direct statistical inference than descriptive summaries (e.g., PCA) or "black box" algorithmic solutions, and it directly prioritizes extracting factors that have a simple relationship to the observed items, when possible (20,21). Conventionally, factor analysis is applied to sets of items within a single questionnaire to identify or confirm underlying structure and estimate scores that more accurately measure the latent constructs captured. This approach has scaled successfully to large cohorts, for example in modeling measures of cognition²² or well-being (23), or in identifying structure across disease comorbidities²⁴. Recently, factor analysis has been adapted to model the structure of genetic, rather than observed phenotypic, correlations across traits (25–27), with further extensions to other types of large-scale 'omics data proposed (28).

As mentioned above, we have now also softened language in our Introduction around FA being "optimized" for our use case:

We also consider the limitations and tradeoffs involved in applying factor analysis at this scale. While our approach is one of many that may be used to distill biobank data, these analyses reemphasize the value of principled dimensionality reduction, and reveal important insights into human variation across the complex and multifaceted human phenome.

To demonstrate the ways in which PCA and FA might differ, we further conducted a PCA on the partial Pearson correlation matrix that was originally input to the EFA. We now reference this analysis within the Results:

An initial exploratory model explains 18.5% of overall variance across input phenotypes and demonstrates good absolute fit in the modeling sample (N=33,860; RMSEA=0.015; CFI=0.883; see **Supplementary Table 1** for additional fit metrics across the modelling and holdout samples; see **Supplementary Text** for more on interpretation of absolute and relative fit in the context of this analysis). ... The same number of components extracted from a principal component analysis explains a similar amount of overall variance (i.e., 21.6%), but the allocation of items and weights across components differs from that of the factors, highlighting important distinctions between these two common dimensionality reduction algorithms (**Extended Data Figure 3; Supplementary Table 2; Supplementary Text**).

And include a corresponding Extended Data Figure:

Extended Data Figure 3. Comparison of EFA to PCA. A) The score correlations across the 36 factors and principal components. **B)** The proportion of variance explained in each of the 36 factors by all 36 PCs. **C and D)** Per-item scatterplots of scoring coefficients for factors vs. PCs across thematically similar pairs, demonstrating sparser loadings amongst the factor scoring coefficients vs. the PC scoring coefficients.

We describe our comparison process within the Methods section:

Principal component analysis

We compare the factor analysis to principal component analysis (PCA), another commonly used dimensionality reduction method, using the same partial Pearson correlation matrix that served as input to the EFA. Eigenvalues and eigenvectors of this input matrix were computed using the `linalg.eig` function within `numpy` in Python. Eigenvectors corresponding to the 36 greatest eigenvalues, in order, were extracted for comparison to the 36-component model of the EFA.

And we go into additional details and interpretations within the Supplement, including discussion of why factor analysis has better theoretical grounding than PCA for this type of analysis:

PCA vs. EFA comparison

Principal components analysis (PCA) and exploratory factor analysis (EFA) are both commonly used dimensionality reduction techniques. Much has been written about the relationship between these methods (e.g. 11–13). We summarize some key features here, before demonstrating the relationship of their results when applied to the current dataset.

Methodological comparison

First, EFA and PCA have different goals. EFA states a model that the observed correlation between items can be explained by the contribution of a set of unobserved, latent factors and residual measurement error, and then attempts to find loadings that best fit this model to the data, generally according to some fit statistic (e.g. “minres”; 14). PCA aims to find linear combinations of items that can explain the most variance in the dataset and are uncorrelated from each other, which corresponds to eigenvectors of the correlation matrix (assuming standardization of the variables prior to analysis).

The distinction between these approaches is evident in the output. PCA produces loadings that are weights for linear combination of the items that compute PCs. The comparable values from EFA are factor scoring coefficients which are used to estimate factor scores (with error) from a linear combination of the items, but these coefficients are a post hoc transformation of the factor loadings fitted by the EFA model. This transformation exists because the loadings estimate the effect of factors on the items, while PCA calculates a contribution of the item to the PC, i.e. in the opposite direction. Both methods ultimately capitalize on the same correlation structure within the data, e.g. since summing correlated variables will yield higher variance for PCA, but these different starting points lead to somewhat different solutions.

The clearest connection between these two methods exists in “principal axes” factor analysis. In that approach, the correlation matrix is modified to replace the diagonal elements with the squared multiple correlation of prediction of the given item by all other items, and then PCA is performed on the modified matrix to estimate factor loadings. The diagonal elements of the correlations matrix are then updated to the “communalities” based on the initial fit, i.e. the proportion of the item explained by the fitted factor structure, and PCA is refit. This process is repeated iteratively until convergence. This approach highlights the impact of the factor analysis model differentiating between the structure explained by factors and residual error as compared to PCA run on a standard correlation matrix. In short, PCA, does not distinguish between unique, or error, variance, and common, shared variance.

Second, the model-based approach of EFA enables inference on the fitted model, both for overall model fit and for individual parameters. Inferential procedures for PCA are much less common. Adding distributional assumptions to PCA permits testing the number of components with Tracy-Widom statistics (15) or Bayesian models (16), and a number of related rules of thumb have also been proposed (13), but inference on overall model fit or individual loading is generally not applicable. Being able to assess model fit in factor analysis is useful both for testing our model construction and for evaluating its generalizability in a holdout sample.

Lastly, factor analysis not only extracts a set of factors but also rotates their axes to aid in interpretability. These rotations enforce a “simple structure” on the factor solution by adding additional optimization criteria that resolve the rotational indeterminacy of the multidimensional set of factors. In general these optimization criteria (e.g. “varimax”; 17) prioritize solutions where each factor has relatively few items with strong loadings, and each item loads strongly on few factors. This ensures a relatively simple structure along each latent axis, generally allowing the nature of each latent factor to be inferred from its top items. In theory this kind of rotation could also be applied to PCA results, but this is not conventionally part of PCA analysis and lacks the depth of previous literature evaluating the use of rotation in factor analysis.

In sum, while both factor analysis and PCA provide dimensionality reduction that captures the correlation structure observed in phenotypic data, factor analysis has the benefit of providing a testable model and adds a conventional step of factor rotation to identify simpler axes for understanding the identified structure.

Results

To demonstrate how our choice of factor analysis compares to PCA for the purpose of our analyses, we additionally perform PCA on the same partial Pearson correlation matrix that served as input to the EFA.

The first 36 principal components explained 21.6% of the variance in the input phenotypes. This is slightly higher than the variance explained by the EFA solution (18.5%), which is perhaps unsurprising given that the PCA algorithm is “greedy” and seeks to maximize explained variance in the input matrix. By comparison, the EFA algorithm optimizes for the accuracy of approximating the observed correlation between variables rather than directly optimizing the variance explained.

Comparison of the identified factors and PCs shows broad differences. Estimated correlations between factor scores and principal components reveal only sporadic overlap between the contents of the factors and components (**Extended Data Figure 3A**). Inspection of the top items (i.e., by loading for EFA, by scoring coefficient for PCA) reveals that the content of each PC is less clearly differentiated than in the factor solution. A number of components, but not factors, share the same top item; for example, PCs 29 and 30 both have “Diabetes diagnosed by doctor” as their top item. As a result fewer items are strongly represented by the PCs: 175 items (or 24.0%) fall within the top 5 item in one or more EFA factors, but in the PCA this number falls to 105 (14.4%). The top items observed for each PC are also often harder to interpret (**Supplementary Table 2**).

Factor or Component	Item Number	EFA	PCA
1	1	Recent inability to stop or control worrying	Recent feelings of depression
1	2	Recent worrying too much about different things	Recent feelings of tiredness or low energy
1	3	Recent feelings or nervousness or anxiety	Recent lack of interest or pleasure in doing things
1	4	Recent trouble relaxing	Recent feelings of inadequacy
1	5	Recent feelings of depression	General happiness with own health
2	1	Happiness	Body mass index (BMI)
2	2	Fed-up feelings	Nitrogen dioxide air pollution; 2005
2	3	Mood swings	Year ended full time education
2	4	Frequency of depressed mood in last 2 weeks	Particulate matter air pollution (pm10); 2007
2	5	Miserableness	Waist circumference

Top few rows of Supplementary Table 2

The lack of alignment between the PCs and the EFA factors is consistent with the axes of variation identified in the EFA being rotated based on the “varimax” criterion for simple structure while the PCA axes are unrotated. The majority of the variance in the factor scores can be jointly explained by the PCs (**Extended Data Figure 3B**), suggesting that the overall space extracted by the two methods is similar. The EFA however yields factors that generally have sparser loadings, with loadings from the PCA generally involving moderate contributions from a much longer list of variables (**Extended Data Figure 3C&D**).

Still, for some factors the correspondence to the PCs is sufficiently diffuse that the content of the factor would likely be overlooked by inspection of the PCs alone. For example, while 76.3% of variance in Factor 23 can be explained by the 36 PCs, no single PC explains more than 11.3% of the factor’s variance. Furthermore, the most predictive PC, PC21, include variables reflecting OTC pain medication use and household size, rather than the variables related to physical activity and health behaviors captured by Factor 23. Similarly, only two PCs individually explain >10% of variance in Factor 9, which in the FA contains items related to trauma and its sequelae: PC1 (explaining 11.0%) is composed primarily of depression-related items, while PC29 (explaining 11.7%) is led by items related to hearing.

Overall, EFA and PCA are both common dimensionality reduction methods with valid potential applications to large-scale phenotypic data, but represent different priorities. However, when the desire exists to evaluate underlying structure across a dataset in addition to simply reducing data complexity, factor analysis is better aligned with those goals.

Given that the primary focus of this paper is FA, but we understand the reviewer's desire for comparison to PCA (as a very commonly used dimensionality reduction method), we felt that placing these details within the Supplement for those interested readers was more appropriate than in the Main Text.

Within our Discussion, we now further emphasize that our approach is simply one of many, and that future combinations and comparisons across methods may be particularly fruitful:

Our approach represents only one possible way of distilling biobank-scale data into informative measures. Though comparison to PCA provides some initial intuition on the benefits of factor analysis (see **Supplementary Text** and **Extended Data Figure 2**), this study is exploratory and not intended to recommend a single best approach. Indeed, we anticipate that comparisons across methods will prove most informative. For instance, our results modelling observed phenotypic correlations may serve as a useful comparator to those of recently developed methods for decomposing genetic correlations (10,27). Furthermore, within the medical domain, supplementing curated, clinician-driven methods for defining disease phenotypes from registry data (e.g., 106,107) with data-driven connections to survey and lab-based measures where available, as suggested by our factors, could prove useful in stratifying individuals for prognosis and treatment.

Finally, throughout, we make sure to mention that our FA methods are adapted or modified, since, as the reviewer points out later in their review, we depart from more traditional FA methods as we scale our analysis for use in a full biobank.

Comment 2: EFA

First, it is not clear to me why WLS/GLS/MINRES/ULS estimators are used, instead of maximum likelihood (ML). ML is the default estimation method in many SEM software and can handle missing data.

Response: We agree with the reviewer that ML would be the ideal method of choice. However, the computational burden of ML is too great when dealing with a biobank-scale dataset. In particular, FIML requires high dimensional integrals that scale exponentially with the number of factors, and above linearly with the number of items and samples (PMID: 26751181). Other workarounds to the poor scaling of FIML have been proposed, but are generally still computationally intensive and not widely available (PMID: 17402812), and thus lack thorough evaluation of their performance. We therefore went with least-squares-based estimators in the EFA and DWLS specifically in the CFA to reduce computational burden, as is suggested as a reasonable, if imperfect, alternative by a number of SEM texts, especially for application at large sample sizes (ex: Brown 2015, Wang & Wang 2012). We very much understand the reviewer's concern here, and indeed grappled with this decision ourselves in our analytic choices.

We have now expanded upon our Supplemental section in which we previously discussed computational limitations of SEM to address this further:

Computational implementation of structural equation modeling

Structural equation modeling was carried out using lavaan (18; version 0.6-3) as a template, with significant modifications made to achieve computational efficiency. It is unsurprising that modifications were necessary for our application, given the conventionally a factor analysis with 120 items (vs. 564 here), 3 factors (vs. 35 here), and 2,400 people (vs. 33,854 in our core data modelling group) would be considered "large" (20). Full-information maximum likelihood, or "FIML" methods, often viewed as the "gold-standard" for SEM in the context of missingness, proved infeasible for our use case due to the computational burden of requiring high-dimensional numerical integration (21). DWLS is often suggested as a reasonable, if imperfect, alternative to FIML that is able to appropriately handle missingness and categorical data (22–24). Nonetheless, DWLS requires the creation of a weight matrix W which contains the asymptotic variance-covariance of each element of the observed covariance matrix (25). As such, W scales quartically with increasing numbers of observed items (i.e., in our case, 564), at a heavy computational cost.

Leveraging matrix sparsity (e.g., using R's Matrix package [26], and R's bigstatsr package [27]), an optimized BLAS (i.e., Intel MKL BLAS), explicit parallelization (e.g., using R's dparallel package [28]), and reduction of linear algebra computations based on our specific use case, we were able to greatly reduce the computational and time complexity of the analysis. Adapted code is available via github [LINK TO BE ADDED UPON ACCEPTANCE]. Nonetheless, parts of the

computation—notably, the computation of W —required the use of a Google Cloud Compute virtual machine with 80 vCPUs and 1.9TB of RAM (i.e., m1-ultramem-80). Future advances in computation, including the use of GPUs, along with continued methods development (29,30), may allow for FA and SEM to be more widely applied at the biobank scale in analyses such as this one.

We also now include this as a limitation within the Discussion section:

Adapting factor analysis to permit execution at the scale and structure of UK Biobank also forced us to make analytic decisions that introduce a number of limitations. ... although full information maximum likelihood estimation methods remain the “gold standard” for modelling missingness and categorical items, their computational burden proved intractable at this scale (103). Instead, we used pairwise deletion and Pearson correlations for the EFA, and used DWLS to fit the CFA, which we expect to have weaker relative performance but to nevertheless be an acceptable alternative (see **Supplementary Text** for additional details)(104,105). Future computational advances, including the use of GPUs, along with continued methods development (104,105), may allow for factor analysis and structural equation modelling to be more widely—and more optimally—applied at biobank scale.

Comment 3: Second, related to the above point, the authors used pairwise deletion for handling missing data, which is a suboptimal approach, compared to methods like full information maximum likelihood (FIML) or multiple imputation (MI).

Response: In our response to Comment 2 above, we elaborate on the prohibitive computational constraints of FIML at biobank scale, and these limitations also apply to this Comment as well. While we agree that this is a suboptimal approach, we conducted sensitivity analyses within the context of CFA modeling which showed that correlation (and covariance) matrices generated based on pairwise deletion within the low-missingness core sample did not differ substantially from those generated using MI. This is perhaps unsurprising given the relatively low level of missingness in the core sample, as well as the fact that imputation itself utilizes correlation across nonmissing variables to infer missing data. We thus concluded that though the approach isn't *optimal*, it was unlikely to substantially alter results.

We have now added the following text to address this within the Supplement:

We generated 10 imputations of the core data group, with 10 iterations per imputation. Visual inspection of mean values across imputations revealed good convergence for all variables with any missingness. In the context of the CFA, we conducted sensitivity analyses to determine the degree to which imputation versus pairwise deletion may affect our results. Specifically, we compared the correlation matrices within the core, low-missingness sample resulting from pairwise deletion versus imputation and found little difference (e.g., for one imputation vs. the pairwise matrix of correlations: mean absolute difference = 0.001[0.003]; **Supplementary Figure 5**). We viewed this as evidence that our decision to use pairwise deletion in the EFA, though suboptimal, was unlikely to affect results. Similarly, going forward, given the strong concordance across pairwise nonmissing data and the individual imputations, as well as across the individual imputations themselves (mean of mean differences across all pairwise combinations of imputations: 0.002[1.38e-5]), we decided to proceed with modelling using only a single imputation for relative computational ease, and instead used the multiple imputations to validate our factor scoring algorithm later on.

And we additionally include the following Supplementary Figure:

Supplementary Figure 5. Comparison of correlation matrices generated using pairwise-complete versus imputed data in the core sample. The lower triangle correlations were generated using pairwise deletion for missingness, while the upper triangle correlations were generated using complete data for a single imputation. Comparisons are shown across all items on the left, as well as within two representative factors (Factor 1 and Factor 7) on the right. Correlation magnitude and direction is indicated by color.

Comment 4: Third, if I understand correctly, the authors used the Pearson correlation matrix as input for EFA, assuming all the variables are continuous. However, when conducting CFA, the authors considered different types of variables. As mentioned on page 35, “this [CFA] allows us to ...more appropriately [model] the covariance structure of the diverse variable types (i.e., binary, ordinal, continuous) ... with more robust handling of missingness”. I do not think this is a “difference” between EFA and CFA- a mixed type of variables and modern methods for handling missing data (see point #2) are also available under EFA.

Response: We agree with the reviewer that ideally this would be addressed in the EFA, and that this is not *technically* a difference between EFA and CFA. However, in our search of freely available packages in both R (e.g., psych, EFAtools, and lavaan) and python (e.g., statsmodels) when we began this project, we were unable to find a function that could handle the following constraints of our dataset in the context of an EFA without using ML (see responses to Comments 2 and 3 above): 1) missingness, 2) different variable types (i.e., binary, ordinal, and continuous), and 3) partial correlation (i.e., through residualization for our chosen covariates; in particular when those covariates were not of the same data type as the target variable). Even more recent searches on our part failed to reveal any clear out-of-the-box advancements in this area. As such, we made the decision to derive the initial model [admittedly suboptimally] based on the partial Pearson correlation matrix, and then to appropriately adjust for these known limitations of the initial EFA downstream in a fuller SEM framework.

We have now updated the sentence highlighted by the reviewer to hopefully clarify that this was not a limitation of EFA but, rather, an analytic decision:

This confirmatory factor analysis (CFA) allows us to test the fit of a more parsimonious model (i.e., omitting loadings with small estimates in the EFA). Within the CFA, we also more appropriately model the covariance structure of the diverse variable types (i.e., binary, ordinal, continuous) and

more robustly handle missingness, in contrast to our decision to use a partial pairwise Pearson correlation matrix as input to the EFA.

As mentioned above, we have also done our best now to point out that our methods are an adapted or modified set of factor analytic methods, in order to scale to a full biobank dataset.

Comment 5: Forth, determining the number of factors and the interpretation of the latent factors are critical steps in EFA. I do not think the authors provide enough information or reasonable justifications for both steps. 1)The authors kept increasing the number of factors until “mathematical error”- this is not a common way of determining the number of factors- at least not the way EFA is designed for. Again, if the goal is simply reducing dimensionality and explaining maximum variances, PCA is a more proper approach.

Response: We agree with the reviewer that our approach to selecting the number of factors is atypical, and is one of a number of modifications made to the “traditional” factor analytic approach for our particular use case (many such modifications are addressed in response to other comments by this reviewer). We had previously loosely outlined our approach within the Methods section, with slightly greater detail and justification (as well as a number of corresponding plots) within the Supplement.

We have since expanded the discussion within the Supplement, to more fully address traditional methods of factor number determination, and why we chose our unusual approach:

A number of guidelines exist for the determination of the number of factors to extract within an EFA, though no hard-and-fast rules exist (6,7). “Stopping criteria” may be influenced, for example, by raw eigenvalues (e.g., Kaiser’s stopping rule, which suggests extracting the number of factors with eigenvalues >1 ; 8), by the number of observed eigenvalues significantly larger than those calculated based on simulated datasets (e.g., as in parallel analysis; 9), by visual inspection of the scree plot (e.g., selecting a number of factors around the “elbow” or inflection point, of the line plot of eigenvalues; 10), by setting a threshold for cumulative variance explained by the factors, or by examining the structural characteristics of the resulting factor solution (e.g., including only the number of “non-trivial” factors).

Eigenvalue-based approaches within our analysis yielded inconsistent results. The scree plot, for example, suggested 30 – 50 factors (**Supplementary Figure 2**). Parallel analysis suggested 177 factors and 253 eigenvalues of the correlation matrix were >1 . The latter results are perhaps unsurprising, given the “long tail” of correlations likely to exist across scattered pairs of items across the 730 input items. Given that one of our main goals was to understand larger-scale structure across the phenotypes assessed within UKB, we therefore placed greater emphasis on the inflection point of the scree plot when deciding on the number of factors to extract.

Supplementary Figure 2. Scree plot of eigenvalues of the correlation matrix used for exploratory factor analysis. The red dots show the 730 eigenvalues, and the horizontal dashed line corresponds to a value of 1.

To further refine the number of factors to extract, we devised a custom approach, with the goal to find a stable structural solution with no non-trivial (e.g., <3 items with loadings >0.3) factors. We explored factor solutions using the following factor extraction methods: WLS (weighted least square), GLS (generalized weighted least square), MINRES (minimum residual) and ULS

(unweighted least square). In all steps the “varimax” rotation was used to find solutions with orthogonal factors. For each method the number of factors extracted was increased until the occurrence of Heywood and ultra-Heywood cases. Heywood (and ultra-Heywood) cases indicate that common factors explain 100% (or more) of an individual item’s total variance, suggesting that the number of extracted factors is too large. This provided an upper limit for the maximum number of factors for each method: 169 for GLS, 186 for WLS, 38 for ULS and 38 for MINRES. The MINRES and ULS methods produced almost identical results and so we only used GLS, WLS and MINRES for further analyses.

For each factor extraction method, increasing the number of factors increased the variance explained by the model but at the same time produced “trivial” factors with one or no items having significant loading (magnitude > 0.3). **Supplementary Figure 3** shows the distribution of the number of significant items in the factors for the different models. Because the 169-factor GLS solution and the 186-factor WLS solution produced many factors with only one significant item, we decided to pursue the 38-factor MINRES model. We were also encouraged by the fact that the number of non-empty factors extracted within this model was within the 30-50 factors suggested by visual inspection of the scree plot.

Supplementary Figure 3. Distribution of number of items in each factor for different factor models. A-C: GLS, WLS and MINRES methods with maximum number of factors (no ultra-Heywood cases). D: final EFA solution of 36 factors using the MINRES method.

The 38-factor MINRES solution contained one “empty” factor, i.e., one factor with no item loading with magnitude > 0.3 . A 37-factor MINRES solution also contained an empty factor. Reducing the number of factors again resulted in a 36-factor MINRES solution (MINRES-36, RMSR = 0.02, variance explained = 18.5%) with no empty nor trivial factors (**Supplementary Figure 3D**).

Within the Methods section, we now refer readers to the Supplementary Text above when discussing our unorthodox method of factor selection (and correct an issue with language surrounding “interpretability” vs “non-triviality” in response to this reviewer’s next comment):

Conventional methods for selecting the number of latent factors provided inconsistent results for the data: the scree plot suggested 30 – 50 factors (**Supplementary Figure 2**), parallel analysis suggested 177 factors, and 253 eigenvalues of the correlation matrix were >1 . To best model large-scale structure across phenotypic items within UKB, we therefore based our decision for number of factors to extract primarily on the inflection point of the scree plot, in tandem with measures of model stability and factor non-triviality (see **Supplementary Text: Exploratory factor analysis** for further discussion).

We explored factor solutions with an increasing number of factors using WLS (weighted least square), GLS (generalized weighted least square), MINRES (minimum residual) and ULS (unweighted least square), all with “varimax” rotation to extract orthogonal factors. As an upper bound, Heywood and ultra-Heywood cases were observed when fitting more than 169 factors with GLS, 186 factors with WLS, or 38 factors with ULS or MINRES. Inspection of these models found that WLS and GLS yielded many factors with strong loadings (>0.3) for at most 1 item, while the ULS or MINRES solutions generally yielded factors incorporating variation from multiple items (**Supplementary Figure 3**).

Based on stability and non-triviality (i.e., lack of factors with only 1-2 items with loadings >0.3), we selected a 36-factor MINRES solution (MINRES-36; RMSR = 0.02, variance explained = 18.5%; **Supplementary Figure 3D**) as our preferred EFA model for subsequent refinement.

Regarding the reviewer’s assertion that PCA may be a more proper approach for our use case, please see our response to the reviewer’s Comment 1, in which we have hopefully sufficiently now updated our paper to justify our choice of FA vs. PCA.

Comment 6: 2) It is also not clear to me how the authors created the “labels” for the 35 factors in the final model. From a psychometric perspective, this step needs to be carefully guided by item analysis and substantive theories. It is not clear why the final 35 constructs are “interpretable constructs” under the labels provided by the authors.

Response: We recognize the careful process that usually precedes naming of factors. For the current analysis, while there are substantive theories and/or expected structure for some of the items (e.g., based on the scales/measures used by UKB), these theories do not extend to cover the full phenome. Therefore,

we treat this analysis as fully exploratory and limit our use of factor labels to the minimum amount necessary to clearly describe the results. We had initially not included labels for our factors at all, but found when presenting the results both internally and at conferences that other researchers wanted at the very least brief descriptions for convenience's sake. In the end, we found that without this brief description at the outset of introducing the factors, we could not maintain the engagement of our readers.

Within the main text, we take care to always refer to each factor by number first, then to give examples of the items they contain. For example(s):

Factor 12, for example, captures correlates of hypertension across the phenome including diagnostic items (e.g., self-reported hypertension and measured blood pressure), risk factors (e.g., family history, BMI, and waist circumference; 39), comorbidities (e.g., self-reported high cholesterol and diabetes; 40,41), and relevant medications (e.g., diuretics and calcium channel blockers).

Factor 33, which captures geographic and cultural indicators of living near London...

Factor 23 ... includes traditional self-reported measures of exercise frequency and duration, as well as other physical and social activity measures like spending time outdoors, playing sports, and walking for pleasure (**Supplementary Table 3**). It also incorporates dietary items such as fruit, vegetable, and oily fish intake, likely reflecting broader correlations across pro-health behaviors.

We now clarify in the legend of Figure 1 that these labels were arrived at by expert consensus (i.e., in this case, an extended meeting between a number of PhDs and MDs within the Analytic and Translational Genetics Unit at MGH during which factor contents were discussed and subsequently described in brief phrases):

Figure 1. Makeup of factors in the final model. Horizontal bars represent proportion variance explained in a given factor score by each of 8 major categories of assessment in UKB, estimated using hierarchical partitioning. To the left, factors are numbered in order of variance extraction in the exploratory factor analysis. To the right, brief descriptions of the items contained within a factor are listed, arrived at by expert consensus of coauthors and colleagues.

We have also altered our language regarding “interpretability” with regard to our EFA solution in the Methods section and Supplement, both in response to this comment and to the previous comment. We now clarify that we were basing our choice not on “interpretability”, but on “non-triviality”.

Methods section:

We explored factor solutions with an increasing number of factors using WLS (weighted least square), GLS (generalized weighted least square), MINRES (minimum residual) and ULS (unweighted least square), all with “varimax” rotation to extract orthogonal factors. As an upper bound, Heywood and ultra-Heywood cases were observed when fitting more than 169 factors with GLS, 186 factors with WLS, or 38 factors with ULS or MINRES. Inspection of these models found that WLS and GLS yielded many factors with strong loadings (>0.3) for at most 1 item, while the ULS or MINRES solutions ~~provided more interpretable results with generally yielded~~ factors incorporating variation from multiple items (**Supplementary Figure 3**).

Based on stability and ~~interpretability~~non-triviality (i.e., lack of factors with only 1-2 items with loadings >0.3), we selected a 36-factor MINRES solution (MINRES-36; RMSR = 0.02, variance explained = 18.5%; **Supplementary Figure 3D**) as our preferred EFA model for subsequent refinement.

Supplement:

For each factor extraction method, increasing the number of factors increased the variance explained by the model but at the same time produced “~~small~~trivial” factors with one or no items having significant loading (magnitude > 0.3). **Supplementary Figure 3** shows the distribution of the number of significant items in the factors for the different models. Because the 169-factor GLS solution and the 186-factor WLS solution produced many factors with only one significant item, we decided to pursue the ~~more interpretable~~38-factor MINRES model. We were also encouraged by the fact that the number of non-empty factors extracted within this model was within the 30-50 factors suggested by visual inspection of the scree plot.

We very much appreciate the concerns of the reviewer here; this was a key focus of our attention during analyses, and we appreciate the opportunity to expand on this decision further.

Comment 7: Finally, as recognized by the authors, the factors in the final model should not be orthogonal (e.g., between F2- depressive symptomatology and F3- clinical anxiety and depression). It is not clear why not use oblique rotation for EFA and allow inter-factor correlations for CFA.

Response: We agree with the reviewer that the ideal way to perform these analyses would have been to use an oblique rotation and allow for inter-factor correlations within the EFA. However, again due to computational complexity, particularly with an eye towards the CFA, we decided to enforce orthogonality, which allowed us to reduce memory and runtime requirements by reducing the number of parameters and substantially simplifying some matrix algebra operations. We do recognize that this is a significant limitation of our approach, and hope that future work will be able to relax this restriction.

We now include an explicit example of how this orthogonality assumption might affect factor interpretation within the Results section of the main text:

It is important to note however that interpretation of the factors relies on their forced orthogonality. Returning to the medical domain, Factor 28 captures diabetes diagnoses and responsive lifestyle changes such as altering one's diet and reducing sugar intake. Had factors not been modeled as independent, Factor 28 would likely share variance with Factors 12 and 16 above, as well as with Factor 7, which captures BMI and adiposity; each of these factors capture aspects of the well-known "metabolic syndrome" (42). In the context of our model, Factor 28, and each factor more generally, is most accurately interpreted as representing the remaining covariance of the items within it (e.g., diabetes diagnosis and treatment), once accounting for the variance explained by the other related (e.g., cardiometabolic), and unrelated, factors.

...

Comparison of Factor 28 to a prior GWAS of type 2 diabetes (51), however, reveals imperfect capture of the clinical definition ($rg=0.68[0.02]$, InterceptGcov=0.36[0.01]; UKB-excluded $rg=0.70[0.03]$; Extended Data Figure 9), likely due to its modelled orthogonality from other factors capturing cardiometabolic constructs. More specifically, the factor has higher genetic overlap with cholesterol measures (e.g., total cholesterol [52] $rg=0.29[0.04]$ vs. $0.04[0.03]$) but lower overlap with BMI (53) ($rg=0.23[0.03]$ vs. $0.49[0.03]$) and an inverse correlation with blood

pressure (54) ($r_g = -0.18[0.02]$ vs. $0.20[0.02]$), reflective of its inclusion of high cholesterol in its factor definition and its forced independence of Factors 7 and 12 described above.

And include a corresponding Extended Data Figure (previously part of the Supplement):

Extended Data Figure 9. Demonstration of the impact of orthogonalization on genetic architecture of Factor 28 versus and outside GWAS of Type 2 Diabetes. The heatmap shows the estimated r_g between 20 selected outside cardiometabolic summary statistics and 1) our Factor 28, 2) an outside GWAS of Type 2 Diabetes, and 3) and outside GWAS of Type 2 Diabetes adjusted for BMI. Color represents the magnitude and direction of genetic correlation.

We then further highlight this limitation within the Discussion:

Adapting factor analysis to permit execution at the scale and structure of UK Biobank also forced us to make analytic decisions that introduce a number of limitations. Most prominently, we

required orthogonality across the factors to reduce the number of parameters to be estimated and substantially simplify matrix algebra operations. As such, the variance captured by each factor is forced to be independent, complicating interpretation; specifically, each factor must be viewed as representing the covariance structure of the items within it, once accounting for covariance modeled by the other factors. Caution must therefore be exercised in interpreting relationships within and between factors, since this independence is an artifact of the modeling approach.

Comment 8: 3. CFA

First, the authors need to be more specific regarding the estimation method used for CFA. When using diagonal weighted least squares (DWLS), lavaan also provides ‘robust’ variants (e.g., WLSM and WLSMV) to better correct the standard errors and test statistics. It is not clear if the authors used any robust version of DWLS- if not, why.

Response: We thank the reviewer for pointing out the confusion in how we reported this originally. Since lavaan uses only the diagonal for estimation (even when “robust” variants are specified), we had previously reported *estimating parameters* using DWLS. However, once we achieved the final model, we utilized WLSMV to calculate robust standard errors and test statistics (using a scale-shifted approach).

We have now clarified this within the Methods section and Supplement:

Model parameters were estimated using diagonal weighted least squares (DWLS), with final robust standard errors and test statistics calculated using a scale-shifted approach (i.e., the WLSMV option in lavaan).

Comment 9: Second, when applying FA (EFA and CFA), it is standard to report all the test statistics (chi-square test with df and p-value) and fit indices (RMSEA, CFI, TLI, SRMR, etc.). The authors also mentioned using fit indices such as CFI, SRMR, etc., however, the only fit information reported was RMSEA.

Response: We thank the reviewer for drawing our attention to this omission, which we have now explained more fully. We had been referencing “Supplementary Text” when reporting the fit metrics, which was mistakenly missing.

In short, we argue that for the purposes of our analyses, absolute fit indices (e.g., RMSEA) are more informative than relative fit indices (e.g., CFI), for the following reasons: 1) our desire to summarize the structure of phenotypes within UKB rather than to test and infer the truth of that structure; 2) the unique challenge of modeling hundreds of phenotypes across assessments and surveys, with low cross-item correlations across domains; and 3) known issues with concordance across relative and absolute fit metrics. We have made changes to the manuscript to reflect and explain this.

We now provide more complete justification and discussion of our assessment of model fit, including our primary reliance on absolute rather than relative fit metrics, within the Supplementary Text:

Assessment of model fit

Traditional assessments of model fit include model chi-square (p-value should not be significant), root mean square error of the approximation (RMSEA; values 0.01, 0.05, and 0.08 indicate excellent, good, and acceptable fit, respectively), standardized root mean squared residual (SRMR; values <0.08 indicate good fit), comparative fit index (CFI; values >0.90 indicate good fit), and Tucker Lewis index (TLI; values >0.90 indicate good fit). We report each of these metrics for the modelling and holdout samples in **Supplementary Table 1** for the sake of completeness. In addition, where applicable we report both the uncorrected and “scaled” versions of these metrics, where the scaled values rely on the adjusted test statistic from fitting the model with DWLS. We caution that neither of these values are fully robust for use with categorical data, but no better alternatives are currently available without computation of the likelihood, which is currently infeasible here (31,32).

However, as our approach differed from more typical applications of factor analysis—chiefly, in that we were analyzing items across the phenome rather than those originating within a single scale—not all fit metrics and corresponding cutoffs were deemed equally applicable. We argue that in the context of our analyses, the relative fit indices (i.e., CFI and TLI) are less likely to be informative.

First, our goal is to find a useful summary of the structure of the phenotypic data across UKB, rather than to infer the truth of that structure. Therefore, it is less important how our chosen model fits compared to a null baseline (e.g., relative fit) versus how well the model approximates the data (i.e., absolute fit). This also means we’re willing to accept some levels of model misfit in the interest of model parsimony. For instance, we choose to have the CFA model include EFA

loadings > 0.1, balancing between a more saturated model with all loadings that appeared to potentially be greater than chance (e.g. > 0.03; see Supplementary Text section Selection of minimum loading for factor inclusion) or a sparser model using the conventional rule of thumb (loading > 0.3). We could certainly improve model fit by carrying forward weaker loadings, as evident in the better fit of the EFA (CFI=.883, RMSEA=.015) than the CFA (CFI=.818, RMSEA=.027), but that would come at the expense of having a less parsimonious summary. This same motivation supports focusing on approximate fit (e.g., RMSEA) over exact fit (e.g. chi square).

Second, expectations for what constitutes “good” comparative fit values may be particularly ill-suited to our data. We are modeling an unusually large number of items for factor analysis, and it has been shown that for a given level of misspecification, CFI will report worse fit as the number of items increases (33). We are also modeling a large number of domains, the correlations across items are, on average, quite low. As a result, the baseline null model for the comparative indices has better fit than would be normal for conventional factor analysis data, leaving less room for our factor model to improve on the baseline. Specifically, our results suggest the null model for comparative fit would be sufficient to achieve RMSEA=0.063 in the CFA holdout data (i.e., based on rearranging the equation $TLI = (1 - RMSEA^2) / RMSEAnull^2$ to solve for $RMSEAnull$), which is far less than the RMSEA=0.153 level of misfit that has been recommended as necessary for comparative fit indices to be informative (<https://davidakenny.net/cm/fit.htm>).

Lastly, there is room for concern about overreliance on fit statistics of any form and their respective common cutoffs. Lai and Green (2016) provide an excellent overview of features that can cause disagreement between relative and absolute fit and the many pitfalls in trying to draw inferences from their discordance. As such, in addition to favoring RMSEA/absolute fit over CFI in our model evaluation, we directly inspected the model-implied vs. actual correlation matrices to identify the sources of remaining residual misfit (and note that the most prominent areas of misfit are due to “categorical-single” items and medication codes—see Supplement section **Impact of questionnaire structure**).

For completeness, we also report CFI, TLI, SRMR, and the goodness-of-fit chi square, in addition to RMSEA, for the EFA, the CFA on modeling set, and the CFA on the holdout set in **Supplementary Table 1**. (Note that we now include, though do not advise the interpretation of, the “scaled” fit indices, in response to the reviewer’s subsequent comment):

	EFA: MODELLING	CFA: MODELLING		CFA: HOLDOUT	
	raw	raw	scaled	raw	scaled
χ^2	2113236, df=240435, p=0	3229339.489, df=126123, p=0	1134113.682, df=126123, p=0	956588.912, df=127260, p=0	285614.737, df=127260, p=0
RMSEA	0.015[0.015-0.015]	0.027[0.027-0.027]	0.015[0.015-0.015]	0.028[0.028-0.028]	0.012[0.012-0.012]
SRMR	0.015	0.058	N/A	0.076	N/A
CFI	0.883	0.818	0.805	0.802	0.802
TLI	0.871	0.817	0.803	0.802	0.802

Within the Main Text of the manuscript, we've added CFI values to the description of the Results, with a reference to the aforementioned Supplemental Text:

An initial exploratory model explains 18.5% of overall variance across input phenotypes and demonstrates good absolute fit in the modeling sample (N=33,860; RMSEA=0.015; CFI=0.883; see **Supplementary Table 1** for additional fit metrics across the modelling and holdout samples; see **Supplementary Text** for more on interpretation of absolute and relative fit in the context of this analysis).

...

A reduced and refined confirmatory FA demonstrates acceptable fit based on absolute fit metrics in a holdout sample of 8465 UKB participants (RMSEA=0.028; SRMR=0.076; **Supplementary Table 1; Supplementary Text**).

Finally, we clarify our approach to assessing fit more clearly within the Methods section and refer back to the Supplement section above:

Traditional assessments of model fit include model chi-square (p-value should not be significant), root mean square error of the approximation (RMSEA; values 0.01, 0.05, and 0.08 indicate excellent, good, and acceptable fit, respectively), standardized root mean squared residual (SRMR; values <0.08 indicate good fit), comparative fit index (CFI; values >0.90 indicate good fit), and Tucker Lewis index (TLI; values >0.90 indicate good fit). For the sake of completeness, we

report all of the above values for the EFA and for both the modelling and holdout samples for the CFA in **Supplementary Table 1**. In assessing overall model fit, we rely primarily on the absolute fit measures (e.g., RMSEA) since they are better suited to evaluating how well our fitted model approximates the structure of the observed phenotypic data (see **Supplementary Text: Assessment of model fit** for more on interpretation of absolute and relative fit in the context of this analysis).

Comment 10: Third, related to the above point, when using DWLS estimators (WLSMV), the methods for computing fit indices are yet to be developed. For example, lavaan typically reports RMSEA, RMSEA.scaled, or RMSEA.robust, it is not clear which RMSEA the authors reported. Also, since multiple imputation was used, it is not clear which procedure the authors used to aggregate the RMSEA across imputed datasets.

Response: We appreciate the reviewer's attention to detail and intimate knowledge of the lavaan program! The reviewer is correct in their assumption that though we used WLSMV to compute SEs and test statistics, we reported only uncorrected fit statistics due to robust methods still being in development.

The "scaled" fit metrics output by lavaan are naive estimates based on inserting the model test statistic (as appropriately adjusted from WLSMV) into the standard formulas for computing RMSEA, CFI, etc. Although this approach is intuitive, the resulting scaled scores have poor performance (PMID 27014948). Work has been done to re-derive these fit measures for the MLMV estimator (PMID 29624085), but currently approximations to get accurate "robust" fit metrics when using DWLS for categorical data still require evaluation of the full likelihood (PMID 32054327), which is intractable in the current data.

Still, we recognize the interest in seeing the scaled metrics for reference. They are now reported in **Supplementary Table 1**, pasted for reference below:

	EFA: MODELLING	CFA: MODELLING		CFA: HOLDOUT	
	raw	raw	scaled	raw	scaled
χ^2	2113236, df=240435, p=0	3229339.489, df=126123, p=0	1134113.682, df=126123, p=0	956588.912, df=127260, p=0	285614.737, df=127260, p=0
RMSEA	0.015[0.015-0.015]	0.027[0.027-0.027]	0.015[0.015-0.015]	0.028[0.028-0.028]	0.012[0.012-0.012]
SRMR	0.015	0.058	N/A	0.076	N/A
CFI	0.883	0.818	0.805	0.802	0.802
TLI	0.871	0.817	0.803	0.802	0.802

We also now note our reasoning in our Methods section and Supplement (adding to edits made in response to the previous comment):

Traditional assessments of model fit include model chi-square (p-value should not be significant), root mean square error of the approximation (RMSEA ; values 0.01, 0.05, and 0.08 indicate excellent, good, and acceptable fit, respectively), standardized root mean squared residual (SRMR; values <0.08 indicate good fit), comparative fit index (CFI; values >0.90 indicate good fit), and Tucker Lewis index (TLI; values >0.90 indicate good fit). For the sake of completeness, we report all of the above values for the EFA and for both the modelling and holdout samples for the CFA in **Supplementary Table 1**. ... In addition, where applicable we report both the uncorrected and “scaled” versions of these metrics, where the scaled values rely on the adjusted test statistic from fitting the model with DWLS. We caution that neither of these values are fully robust for use with categorical data, but no better alternatives are currently available without computation of the likelihood, which is currently infeasible here (105, 106).

And we now clarify in the Methods section that the model was fit on a single imputation only:

Evaluation of this approach using synthetic missingness at random (MAR) and completely at random (MCAR) showed good convergence and minimal systematic bias. Comparisons of correlation and covariance matrices generated using pairwise deletion versus imputation revealed them to be nearly identical (**Supplementary Text: Multiple imputation of core data group; Supplementary Figure 5**); as such, to conserve computational resources, only a single imputed dataset was used for modelling purposes.

Confirmatory factor analysis models were fit to the imputed data for the modelling group...

Please see also our response to this reviewer's Comment 3 above, in which we added the following text to the Supplement:

We generated 10 imputations of the core data group, with 10 iterations per imputation. Visual inspection of mean values across imputations revealed good convergence for all variables with any missingness. In the context of the CFA, we conducted sensitivity analyses to determine the degree to which imputation versus pairwise deletion may affect our results. Specifically, we compared the correlation matrices within the core, low-missingness sample resulting from pairwise deletion versus imputation and found little difference (e.g., for one imputation vs. the pairwise matrix of correlations: mean absolute difference = 0.001[0.003]; **Supplementary Figure 5**). We viewed this as evidence that our decision to use pairwise deletion in the EFA, though suboptimal, was unlikely to affect results. Similarly, going forward, given the strong concordance across pairwise nonmissing data and the individual imputations, as well as across the individual imputations themselves (mean of mean differences across all pairwise combinations of imputations: 0.002[1.38e-5]), we decided to proceed with modelling using only a single imputation for relative computational ease, and instead used the multiple imputations to validate our factor scoring algorithm later on.

Our main use of the multiply imputed datasets was to validate our factor scoring algorithm in comparison to lavaan's FIML approach, as detailed in the Supplementary section "Validation of factor scoring methods in the core data group", the beginning text of which is pasted below for reference:

To validate our factor-score-generating methodologies, we compared scores from our methods to those generated using a maximum-likelihood-based (ML) method in lavaan for the core data group. Factor scoring was performed using the ML option in lavaan for all 10 multiple imputations of the core dataset . Though we chose to compare our scoring methods to the ML method in lavaan, the latter cannot be considered the "gold standard," as numerous factor-scoring methods exist, with each simply relying on and/or prioritizing different assumptions. However, this comparison performed to provide more confidence in our chosen method, which sought to mimic ML estimation in spirit.

To test for phenotypic concordance across methods, we obtained Pearson correlation coefficients between factor scores generated using our method and the mean of those obtained in lavaan across all 10 imputations. These correlation estimates did not weight individuals based on the expected precision of their factor scores, but were restricted to only individuals meeting our analytic inclusion thresholds. Phenotypic correlations were moderate to very high for the dependent-variable (range: 0.531-0.994, mean=0.842(0.137)) and independent-variable (range: 0.540-0.993, mean=0.843[0.132]) formulations. Concordance across dependent- and independent-variable formulations was excellent (range: 0.967-1.000, mean=0.992[0.009]).

Comment 11: 4. Factor scores as measurements

As a goal for the current study, the latent factors (factor scores) were used to “improve measurement of health-related behavior”. From a psychometric perspective, I recommend the authors compute and report reliability measures for the scores.

Response: We have now calculated the expected correlation of each factor score to the underlying factor score (i.e. $\sqrt{\text{reliability}}$) and report descriptive statistics within the text:

The expected correlation between estimated scores and the corresponding latent factors (43) are generally strong (mean=0.863, s.d.=0.073, range: 0.611[Factor 28] to 0.974[Factor 4]), suggesting good reliability.

We would like to end our response to Reviewer #2 by again thanking the reviewer for pushing us to carefully consider each of our analytic choices, as well as our language surrounding them. Combing back through each detail, softening language to be more appropriate for the analysis we conducted, and performing robustness checks and comparisons to other methods really strengthened this paper and, perhaps more importantly, heightened the transparency, assumptions, strengths, and weaknesses of our choices; this made for a much better set of analyses and a better overall paper.

Reviewer #3:

Remarks to the Author:

Background

This manuscript presents the results of applying factor analysis (FA) to a subset of phenotypes captured in UK Biobank. There is a mixture of phenomenological analyses (what do the derived factors capture and how do they relate to other measurements? - e.g. mortality) and genetic analyses (what is the value of carrying out genetic analysis of factors versus individual measurements? - e.g. numbers of variants discovered associated with factors compared to component phenotypes). Many of the analyses are exploratory - and the conclusions therefore either vague (e.g. "This approach provides an important first step toward better embracing the full and complex measured phenome to power discovery for human health and wellbeing" or "...SES is a multidimensional construct"), and/or come with strong caveats, though there are several interesting observations within the paper.

Advances

Although I have several issues that I would like to see addressed (see below), I'd like to start with highlighting what I see as the advances of the paper. I see the novel insights as:

1. A proof-of-concept that high-dimensional biobank-style data can be represented into lower-dimensional summaries in such a way that generates novel insights (e.g. genetic loci not previously identified). This is not the first paper using dimension-reduction techniques on biobank-style data (e.g. Holden et al. 2011 - DOI: 10.1186/1478-7954-9-15, Ronaldsen et al. 2022 - doi:10.1080/13607863.2021.2017847), but it is uniquely ambitious in scope, combining both medical and non-medical data types.
2. A demonstration that - despite this - more genetic loci are associated with either constituent phenotypes than factors per se (which is intriguing).
3. An observation that genetic risk for factors can be strongly correlated, even when - at a phenotypic level - the factors are defined as being independent (orthogonal).

There are, of course, many specific observations about specific factors and their associations (e.g. around physical activity or trauma). My high-level sense is that - for all of these - the analyses are too superficial to really add to the literature on the specific domains (for example, there is quite a literature on genetic risk factors for PTSD and other traumas - and while the work here acknowledges this, it does not explain what the added insights are).

Major points

Comment 1: The highest level issue I have is that the work is - fundamentally - exploratory. Not asking a specific question, but rather applying a single, specific technique (factor analysis) to a complex data set and exploring what is found. I don't find this uninteresting, but it's not really question-driven research. There are interesting research questions within the domain (for example, what's the best way to represent biobank data to generate genetic insights or to enable prediction), but this is not the approach taken. Although detailed notes are provided about exactly how the factor analysis was applied, it is just one of many possible ways of representing such data and comes with specific assumptions (e.g. orthogonality) that may not be appropriate for the problem. My (relatively quick) reading of the literature, suggests that others have tried different approaches to the problem (e.g. multiview learning, LDA-derived approaches, PCA) and I would have enjoyed reading more about whether different approaches come to the same or different conclusions.

Response: We agree that our analysis is highly exploratory. Because most work on phenotypic structure focuses on relationships within a domain (or a couple related domains) rather than considering the full phenome, there is less of a foundation for pursuing specific question-driven research. Instead, we suggest that this kind of initial characterization of broad phenotypic structure may serve as a useful reference point for building hypotheses. We have made this motivation more explicit in the Introduction.

In the current study, we modify and extend the factor analytic approach to explore the structure of a much broader set of multimodal biobank phenotypes. We apply this approach to the cross-sectional phenome of UK Biobank (UKB)(4) and evaluate whether the identified structure is (a) informative about relationships that might be unexpected or normally obscured, and (b) effectively summarized by factor scores to enable more powerful analyses of linked phenotypic and genetic data. We also consider the limitations and tradeoffs involved in applying factor analysis at this scale. While our approach is one of many that may be used to distill biobank data,

these analyses reemphasize the value of principled dimensionality reduction, and reveal important insights into human variation across the complex and multifaceted human phenome.

Along those lines, we also agree with the reviewer that FA is indeed one of many possible ways to represent biobank-scale data, and we have worked to clarify in the main text that it is not objectively *better* than other approaches, each of which relies on a number of different assumptions. Though a full comparison of techniques to reduce the dimensionality of UKB is out of scope of this paper, we have now also performed a PCA on the partial Pearson correlation matrix used as input to our EFA, to permit basic comparisons. In order to not detract from the main thrust of the paper--our FA analyses--we reference this comparison briefly in the Results section:

An initial exploratory model explains 18.5% of overall variance across input phenotypes and demonstrates good absolute fit in the modeling sample (N=33,860; RMSEA=0.015; CFI=0.883; see **Supplementary Table 1** for additional fit metrics across the modelling and holdout samples; see **Supplementary Text** for more on interpretation of absolute and relative fit in the context of this analysis). ... The same number of components extracted from a principal component analysis explains a similar amount of overall variance (i.e., 21.6%), but the allocation of items and weights across components differs from that of the factors, highlighting important distinctions between these two common dimensionality reduction algorithms (**Extended Data Figure 3; Supplementary Table 2; Supplementary Text**).

With the corresponding Extended Data Figure:

Extended Data Figure 3. Comparison of EFA to PCA. A) The score correlations across the 36 factors and principal components. **B)** The proportion of variance explained in each of the 36 factors by all 36 PCs. **C and D)** Per-item scatterplots of scoring coefficients for factors vs. PCs across thematically similar pairs, demonstrating sparser loadings amongst the factor scoring coefficients vs. the PC scoring coefficients.

We add the following description in the Methods section:

Principal component analysis

We compare the factor analysis to principal component analysis (PCA), another commonly used dimensionality reduction method, using the same partial Pearson correlation matrix that served

as input to the EFA. Eigenvalues and eigenvectors of this input matrix were computed using the `linalg.eig` function within `numpy` in Python. Eigenvectors corresponding to the 36 greatest eigenvalues, in order, were extracted for comparison to the 36-component model of the EFA.

And we provide additional details and interpretations within the Supplement:

PCA vs. EFA comparison

Principal components analysis (PCA) and exploratory factor analysis (EFA) are both commonly used dimensionality reduction techniques. Much has been written about the relationship between these methods (e.g. 11–13). We summarize some key features here, before demonstrating the relationship of their results when applied to the current dataset.

Methodological comparison

First, EFA and PCA have different goals. EFA states a model that the observed correlation between items can be explained by the contribution of a set of unobserved, latent factors and residual measurement error, and then attempts to find loadings that best fit this model to the data, generally according to some fit statistic (e.g. “minres”; 14). PCA aims to find linear combinations of items that can explain the most variance in the dataset and are uncorrelated from each other, which corresponds to eigenvectors of the correlation matrix (assuming standardization of the variables prior to analysis).

The distinction between these approaches is evident in the output. PCA produces loadings that are weights for linear combination of the items that compute PCs. The comparable values from EFA are factor scoring coefficients which are used to estimate factor scores (with error) from a linear combination of the items, but these coefficients are a post hoc transformation of the factor loadings fitted by the EFA model. This transformation exists because the loadings estimate the effect of factors on the items, while PCA calculates a contribution of the item to the PC, i.e. in the opposite direction. Both methods ultimately capitalize on the same correlation structure within the data, e.g. since summing correlated variables will yield higher variance for PCA, but these different starting points lead to somewhat different solutions.

The clearest connection between these two methods exists in “principal axes” factor analysis. In that approach, the correlation matrix is modified to replace the diagonal elements with the squared multiple correlation of prediction of the given item by all other items, and then PCA is performed on the modified matrix to estimate factor loadings. The diagonal elements of the correlations matrix are then updated to the “communalities” based on the initial fit, i.e. the proportion of the item explained by the fitted factor structure, and PCA is refit. This process is repeated iteratively until convergence. This approach highlights the impact of the factor analysis model differentiating between the structure explained by factors and residual error as compared to PCA run on a standard correlation matrix. In short, PCA, does not distinguish between unique, or error, variance, and common, shared variance.

Second, the model-based approach of EFA enables inference on the fitted model, both for overall model fit and for individual parameters. Inferential procedures for PCA are much less common. Adding distributional assumptions to PCA permits testing the number of components with Tracy-Widom statistics (15) or Bayesian models (16), and a number of related rules of thumb have also been proposed (13), but inference on overall model fit or individual loading is generally not applicable. Being able to assess model fit in factor analysis is useful both for testing our model construction and for evaluating its generalizability in a holdout sample.

Lastly, factor analysis not only extracts a set of factors but also rotates their axes to aid in interpretability. These rotations enforce a “simple structure” on the factor solution by adding additional optimization criteria that resolve the rotational indeterminacy of the multidimensional set of factors. In general these optimization criteria (e.g. “varimax”; 17) prioritize solutions where each factor has relatively few items with strong loadings, and each item loads strongly on few factors. This ensures a relatively simple structure along each latent axis, generally allowing the nature of each latent factor to be inferred from its top items. In theory this kind of rotation could also be applied to PCA results, but this is not conventionally part of PCA analysis and lacks the depth of previous literature evaluating the use of rotation in factor analysis.

In sum, while both factor analysis and PCA provide dimensionality reduction that captures the correlation structure observed in phenotypic data, factor analysis has the benefit of providing a testable model and adds a conventional step of factor rotation to identify simpler axes for understanding the identified structure.

Results

To demonstrate how our choice of factor analysis compares to PCA for the purpose of our analyses, we additionally perform PCA on the same partial Pearson correlation matrix that served as input to the EFA.

The first 36 principal components explained 21.6% of the variance in the input phenotypes. This is slightly higher than the variance explained by the EFA solution (18.5%), which is perhaps unsurprising given that the PCA algorithm is “greedy” and seeks to maximize explained variance in the input matrix. By comparison, the EFA algorithm optimizes for the accuracy of approximating the observed correlation between variables rather than directly optimizing the variance explained.

Comparison of the identified factors and PCs shows broad differences. Estimated correlations between factor scores and principal components reveal only sporadic overlap between the contents of the factors and components (**Extended Data Figure 3A**). Inspection of the top items (i.e., by loading for EFA, by scoring coefficient for PCA) reveals that the content of each PC is less clearly differentiated than in the factor solution. A number of components, but not factors, share the same top item; for example, PCs 29 and 30 both have “Diabetes diagnosed by doctor” as their top item. As a result fewer items are strongly represented by the PCs: 175 items (or 24.0%) fall within the top 5 item in one or more EFA factors, but in the PCA this number falls to 105 (14.4%). The top items observed for each PC are also often harder to interpret (**Supplementary Table 2**).

Factor or Component	Item Number	EFA	PCA
1	1	Recent inability to stop or control worrying	Recent feelings of depression
1	2	Recent worrying too much about different things	Recent feelings of tiredness or low energy
1	3	Recent feelings or nervousness or anxiety	Recent lack of interest or pleasure in doing things
1	4	Recent trouble relaxing	Recent feelings of inadequacy
1	5	Recent feelings of depression	General happiness with own health
2	1	Happiness	Body mass index (BMI)
2	2	Fed-up feelings	Nitrogen dioxide air pollution; 2005
2	3	Mood swings	Year ended full time education
2	4	Frequency of depressed mood in last 2 weeks	Particulate matter air pollution (pm10); 2007
2	5	Miserableness	Waist circumference

Top few rows of Supplementary Table 2

The lack of alignment between the PCs and the EFA factors is consistent with the axes of variation identified in the EFA being rotated based on the “varimax” criterion for simple structure while the PCA axes are unrotated. The majority of the variance in the factor scores can be jointly explained by the PCs (**Extended Data Figure 3B**), suggesting that the overall space extracted by the two methods is similar. The EFA however yields factors that generally have sparser loadings, with loadings from the PCA generally involving moderate contributions from a much longer list of variables (**Extended Data Figure 3C&D**).

Still, for some factors the correspondence to the PCs is sufficiently diffuse that the content of the factor would likely be overlooked by inspection of the PCs alone. For example, while 76.3% of variance in Factor 23 can be explained by the 36 PCs, no single PC explains more than 11.3% of the factor’s variance. Furthermore, the most predictive PC, PC21, include variables reflecting OTC pain medication use and household size, rather than the variables related to physical activity and health behaviors captured by Factor 23. Similarly, only two PCs individually explain >10% of variance in Factor 9, which in the FA contains items related to trauma and its sequelae: PC1 (explaining 11.0%) is composed primarily of depression-related items, while PC29 (explaining 11.7%) is led by items related to hearing.

Overall, EFA and PCA are both common dimensionality reduction methods with valid potential applications to large-scale phenotypic data, but represent different priorities. However, when the desire exists to evaluate underlying structure across a dataset in addition to simply reducing data complexity, factor analysis is better aligned with those goals.

More generally, we now include additional text in the Introduction citing prior dimensionality reduction efforts at the biobank scale and highlight key features of factor analysis vs. other methods:

Dimensionality reduction is a common task in many domains, with a recent proliferation of methods having been applied to biobank-scale data. Principal component analysis (PCA), for example, provides a lower-dimensional representation of the strongest axes of variation in a dataset. It has been leveraged to, for instance, identify dimensions of genetic ancestry in genotype data (8), extract features from individual biobank questionnaires (9), and identify sets of genetic

variants with similar patterns of association across thousands of genome-wide association studies (GWAS)(10). Other data reduction approaches prioritize identifying correlated sets of variables across data types (11), modeling latent classes (12), or creating lower-dimensional representations for visualization purposes (13,14). Deep learning methods such as autoencoders and transformers have been used to integrate 'omics data across modalities (15,16), and to extract relevant features from electronic health records (EHRs) (17,18), respectively.

Biobank analyses have devoted relatively less attention to factor analysis, an approach commonly used in the social sciences that models the observed correlation between variables as arising from one or more shared continuous latent (unobserved) factors (19). Factor analysis has the benefit of being model-based, facilitating more direct statistical inference than descriptive summaries (e.g., PCA) or "black box" algorithmic solutions, and it directly prioritizes extracting factors that have a simple relationship to the observed items, when possible (20,21). Conventionally, factor analysis is applied to sets of items within a single questionnaire to identify or confirm underlying structure and estimate scores that more accurately measure the latent constructs captured. This approach has scaled successfully to large cohorts, for example in modeling measures of cognition²² or well-being (23), or in identifying structure across disease comorbidities²⁴. Recently, factor analysis has been adapted to model the structure of genetic, rather than observed phenotypic, correlations across traits (25–27), with further extensions to other types of large-scale 'omics data proposed (28).

We have also softened language in the introduction that may have implied the objective superiority of FA to other methods (i.e., by stating it was "optimized" for our use case):

We also consider the limitations and tradeoffs involved in applying factor analysis at this scale. While our approach is one of many that may be used to distill biobank data, these analyses reemphasize the value of principled dimensionality reduction, and reveal important insights into human variation across the complex and multifaceted human phenome.

Within the Discussion, we now contextualize our study within the prior biobank-scale dimensionality reduction efforts cited in the Introduction and reiterate that FA is one of a number of potential approaches:

Our approach represents only one possible way of distilling biobank-scale data into informative measures. Though comparison to PCA provides some initial intuition on the benefits of factor analysis (see **Supplementary Text** and **Extended Data Figure 2**), this study is exploratory and not intended to recommend a single best approach. Indeed, we anticipate that comparisons across methods will prove most informative. For instance, our results modelling observed phenotypic correlations may serve as a useful comparator to those of recently developed methods for decomposing genetic correlations (10,27). Furthermore, within the medical domain, supplementing curated, clinician-driven methods for defining disease phenotypes from registry data (e.g., 106,107) with data-driven connections to survey and lab-based measures where available, as suggested by our factors, could prove useful in stratifying individuals for prognosis and treatment.

Comment 2: A related issue is that where findings are consistent with the literature, this is stated as confirmation that the approach works well, but where there are novel findings (e.g. BMI/gaming), these are not followed up. I find this irritating as a reader - if you want me to believe that this approach is going to generate insights that I wasn't able to get from reading the literature, or simple analyses like correlation, then I would like to see putative hypotheses dug into, tested and validated using external data sets. The work on Factor 23 (physical activity) perhaps comes closest to this, but I would, for example, be interested in a much more detailed analysis of this (pulling apart its causes and consequences, potentially using MR to look at causality, asking directly what the best predictor of reduced mortality is, etc.). And also a much deeper analysis of the contributions here compared to other work.

Response: As the reviewer notes, we as authors struggled with the need to balance “established” findings that support the credibility of our approach with “novel” findings that, though interesting, might not be as convincing to readers. As our work is exploratory, we felt that diving too deeply into hypothesis generation and testing was beyond the scope of the manuscript. However, we have now conducted additional analyses within the Add Health dataset to extend the findings within our three “vignettes” (Factor 9, Factor 23, and the SES Factors) to an outside dataset using polygenic scores. We believe that these analyses demonstrate the extensibility of these findings beyond UKB alone and show potential “use cases” for summary statistics derived from our analyses.

In the Results section within the SES vignette, we have added the following text:

Polygenic score analyses of the SES factors conducted in the independent Add Health dataset further solidify these key distinctions and provide evidence of generalization beyond the UKB

sample (**Supplementary Table 7; Figure 4C**). The Factor 10 PGS outperforms those of Factors 5 and 15 in its associations with years of education ($R^2=7.7\%$), college completion (pseudo- $R^2=5.5\%$), and verbal cognition ($R^2=4.8\%$). In contrast, Factor 5 scores are the best predictors of income ($R^2=5.3\%$) and high-paying vs. low-paying jobs (pseudo- $R^2=5.7\%$), while Factor 15 scores are most associated with neighborhood satisfaction ($R^2=3.9\%$). Not only do these findings demonstrate replication of our results in an independent, non-UKB sample, but the level of differentiation in prediction articulates a clear disentangling of factors related to SES.

And corresponding figure panel:

Figure 4B. Associations between polygenic scores derived from the SES factors and SES-related items in an outside cohort (AddHealth). Error bars represent 95% bootstrapped confidence intervals. EA: Educational Attainment.

In the Trauma vignette, we have now added the following text:

Within the Add Health cohort, the Factor 9 PGS outperforms those derived from prior GWAS of trauma-related constructs (75–77) in predicting adulthood psychiatric diagnoses (**Supplementary Table 8**), providing further evidence for an increase in genetic signal when aggregating across multiple facets of trauma-related experiences, as well as for the generalizability of that signal to an independent sample.

In the Physical Activity vignette, we have added the following text:

Echoing results within UKB, polygenic score analyses within the independent Add Health dataset reveal that the Factor 23 PGS outperforms polygenic scores formed from each of its top 10 items in predicting medical and health behavior outcomes traditionally thought of as relevant to physical activity (e.g., $R^2=1.4\%$ for hours of physical activity per week, $R^2=2.1\%$ for BMI, and pseudo- $R^2=3.4\%$ for cardiovascular disease; **Supplementary Table 9; Figure 6C**). The Factor 23 PGS even outperforms a multivariate regression model including all 10 item-based PGS (Factor 23 $R^2=3.4\%$ [2.2%,4.7%]; combined top item PGSs $R^2=2.6\%$ [1.4%,3.9%]), further emphasizing the power gained from combining measures across the phenome in a way that leverages underlying correlation structure.

And corresponding Figure (panel A of which, we believe, addresses the reviewer's desire for comparisons of mortality prediction):

Figure 6. Comparative performance of Factor 23 versus individual items. (A) Comparison of incremental R² for mortality prediction. The comparative baseline model for each included covariates for the first 20 genetic PCs, age, chromosomal sex, age², age-x-chromosomal sex, age²-x-chromosomal sex, dummy variables representing the assessment centers of origin, and days from baseline assessment to T₀. (B) Comparison of point estimates of heritability. Error bars represent standard error. (C) Comparison of variance explained by polygenic scores for Factor 23 vs. its top 3 component items for 5 relevant traits in the external Add Health study. Error bars represent 95% bootstrapped confidence intervals, with a lower bound of 0 for visualization purposes. See **Supplementary Table 9** for comparison to all top 10 items.

The text regarding mortality prediction of items vs. Factor 23 is reproduced below for the reviewer's reference:

Prospective analyses of all-cause mortality, for example, show weaker effects for each of Factor 23's 10 top-loading items (incremental pseudo-R²=1.27 x 10⁻³ for factor vs. 6.75 x 10⁻¹⁰ to 1.12 x 10⁻³ for individual items). An unweighted sum of z-scores across these top items is also more

weakly associated with survival (incremental pseudo- $R^2=1.00 \times 10^{-3}$) than the weighted sum used by the factor scores (**Figure 6A**).

We have detailed these additional analyses within our Methods section as follows:

National Longitudinal Study of Adolescent to Adult Health (Add Health) sample

Add Health originated as an in-school survey of a nationally representative sample of US adolescents enrolled in grades 7 through 12 during the 1994–1995 school year (44). Respondents were born between 1974 and 1983, and a subset of the original Add Health respondents has been followed up with in-home interviews, which allows researchers to assess correlates of outcomes in the transition to early adulthood. In Add Health, the mean birth year of respondents is 1979 (s.d. = 1.8) and the mean age at the time of assessment (Wave 4) is 29.0 years (s.d. = 1.8).

...

Polygenic scoring

Polygenic scores in Add Health were constructed with LDpred (135). LDpred estimates polygenic scores using SNP weights that estimate the conditional association of each SNP account for LD and the estimated genetic architecture of the trait, and has been shown to have greater prediction accuracy than conventional LD pruning followed by P value thresholding.

For the Add Health sample, we used the genotyped data from the Add Health prediction cohort to create the LD reference file. After imputing the genetic data to the Haplotype Reference Consortium (136) using the Michigan Imputation Server (137), we used only HapMap3 variants with a call rate >98% and a minor allele frequency >1% to construct the polygenic scores. We limited the analyses to European-ancestry individuals. Polygenic scores were calculated with an expected fraction of causal genetic markers set at 100%. In total, we used 1,168,025 HapMap3 variants to construct the polygenic scores in Add Health. We then used Plink (138) to multiply the genotype probability of each variant by the corresponding LDpred posterior mean over all variants. For all sets of summary statistics, rather from Factor Score GWAS summary statistics, individual item GWAS summary statistics, or outside, publicly available GWAS summary statistics, we ensure that Add Health was not included in the discovery GWAS sample.

We then determined the association of a given polygenic score and an outcome of interest in Add Health. Outcomes used in analyses were as follows: (1) years of completed education; (2) income; (3) a binary measure of having ever completed college; (4) a binary indicator of working either a high-paying or low-paying job; (5) a measure of cognition called the Peabody Picture Vocabulary Test; (6) a 1-5 Likert scale measure of neighborhood satisfaction (“If, for any reason, you had to move from here to some other neighborhood, how happy or unhappy would you be?”); self-reports of doctor-diagnosed (7) panic or anxiety disorder, (8) bipolar disorder, or (9) PTSD; (10) self-report of doctor-diagnosed cardiovascular disease; (11) self-report of doctor-diagnosed diabetes; (12) self-report of number of hours of physical activity per week; (13) measured BMI; and (14) measured systolic blood pressure.

Prediction accuracy was based on an OLS or logistic regression (depending on the outcome) of the outcome phenotype on the polygenic score and a set of standard controls, which included birth year, sex, an interaction between birth year and sex, and the first 10 genetic principal components of the variance–covariance matrix of the genetic data. Variance explained by the polygenic scores was calculated in regression analyses as either the R² change (for continuous or quasi-continuous phenotypes), or the Nagelkerke’s pseudo-R² change (for binary outcomes), that is, the R² or pseudo-R² of the model including polygenic scores and covariates minus the R² or pseudo-R² of the model including only covariates. The 95% confidence intervals around all pseudo-R² values were bootstrapped with 1,000 repetitions each.

Finally, regarding the reviewer’s desire for causal inference surrounding causes, correlates, and consequences among items included within our factors, unfortunately due to relatively low item heritabilities, we are underpowered for this type of genetic analysis (e.g., MR and LCV). However, to give some insight into this question, we did perform LCV analysis, the results of which we now reference briefly in the Results section and describe in full in the Supplementary text. Given the relatively low power of these analyses, we chose not to highlight them further in the main text.

New mention(s) within the Results section:

Furthermore, identifying patterns of shared and nonshared genetic signal across items within a factor can provide insight into how they relate. ... Direct comparisons between GWAS of the top items in a factor have the potential to be more informative by suggesting causal relationships

between observed items, such as evidence for a partially causal effect of doctor diagnosed asthma on self-reported on-the-job breathing problems within Factor 11 (LCV $gcp=0.80$, $p=3.98e-5$; **Supplementary Figure 9**; see **Supplementary Text** for further details on latent causal variable [LCV] analyses); however, power to establish these relationships is limited by the that of the GWAS for each item.

...

Among top items in Factor 23, there is nominal evidence for a partially causal effect of recent participation in strenuous sports on current self-rated health (LCV $gcp=0.37$, $p=5e-3$; **Supplementary Text**; **Supplementary Figure 9**), additionally supporting a role for motivation and socialization in linking activity to perceived health.

We include a full Supplementary Text section, including a Supplementary Figure:

Latent causal variable analysis

To evaluate whether we can distinguish which items in a factor are more likely to be causes or consequences of the identified structure we performed latent causal variable (LCV; 61) analysis of their genetic results. Briefly, using genome-wide results for a given pair of genetically correlated traits LCV tests whether there's evidence for a causal relationship between the pair based on higher-order moments of their GWAS Z scores. Each relationship is characterized by an estimate of the genetic causality proportion (gcp), with values ranging from 1 (meaning the genetics of the first trait have a causal effect on the second trait), to -1 (meaning conversely that the second trait's genetics causes the first trait), with $gcp=0$ indicating no causal relationship. Using this approach, we evaluate the relationship between the top items in factor for four selected factors of interest (6, 11, 16, and 23).

Results of the LCV analyses are shown in Supplementary Figure 9. Broadly, we observe that there is limited power to distinguish causal directions between the top items based on their UK Biobank GWAS results (i.e., the standard errors are generally large). Nevertheless, we do find evidence of partial genetic causality in two instances: an effect of diagnosed asthma and other top items in Factor 11 on reported breathing problems during a job (asthma $gcp=0.80$, $p=3.98e-5$), and an effect of ICD diagnosis of chronic ischemic heart disease on ICD diagnosis of myocardial infarction.

Nominal evidence is also found for a couple other relationships, including a partially causal effect of participation in strenuous sports on overall health rating ($gcp=0.37$, $p=5.0e-3$), but they do not survive correction for multiple testing.

Overall, these results suggest that complete causality ($gcp=1$ or -1) between the items in the current study are unlikely, but that there will likely be the potential to identify partially causal relationships at larger GWAS sample sizes. We also observed that the relationships likely to be only be partially causal, consistent with the existence of some shared genetic components across items along with other genetic risk factors that are more item-specific.

Supplementary Figure 9. LCV results for top items in factors 6, 11, 16, and 23. Genetic causal proportion (gcp) values are reported for each pair of items, along with their standard error. Values in italics with a single asterisk are nominally significant from 0 ($p<.05$), and values in bold with two asterisks are significant after Bonferroni correction for the number of the trait pairs tested. A GCP value of 1 indicates that the genetic component of the item in the row is causal for the genetics of the item in the column. A GCP value of -1 indicates the reverse, that the item in the column is

genetically causal for the item in the row. A GCP of 0 indicates that the pair of items share a latent cause. Intermediate values suggest “partial” causality, i.e. that some (but not all) elements of one item’s genetics are causal for the other item.

Comment 3: As noted above, I do think there are two findings related to genetic factors that are potentially interesting. The first is the finding of variants that affect factors - but constituent phenotypes to a lesser extent. It would be interesting to dig into whether there are differential enrichments for such variants within domain (e.g. wrt Factors 11 or 16), or whether they are just smaller effects of a similar nature? Conversely, what are the enrichments for variants most strongly associated with single phenotypes than factors? Such a differential analysis could provide insights into the genetic architecture of complex traits that are only possible through biobank-style data.

Response: While we agree with the reviewer that this could prove to be a fruitful avenue of future research, for the reasons mentioned in the response above re: LCV and causal inference, we are currently underpowered for such analyses, even at the biobank scale, due to low residual heritability within top items once accounting for genetic overlap with the factor. Nevertheless, for interested readers, we have now performed such analyses comparing differential tissue enrichments of the factors vs. top-item-specific genetic variation using genomic structural equation modelling (gSEM).

We briefly reference these analyses within the Results section in the Main Text:

Furthermore, identifying patterns of shared and nonshared genetic signal across items within a factor can provide insight into how they relate. ... Item-specific effects could be hypothesized to reflect qualitatively distinct genetic mechanisms, but attempts to partition nonshared effects between top items and their corresponding factors generally identify similar loci and tissue enrichments as the factor GWAS, implying “item-specific” signal residual on the factor may commonly reflect differential relative effects of shared loci instead (**Supplementary Text; Supplementary Figure 10**).

And report them more fully within the Supplement, as copied below for reference:

Tissue specificity of shared and nonshared genetic effects

Based on the observation that some genetic loci are significantly associated with some of the top items in a factor without being associated with the factor itself (**Figure 3C**), we consider whether the more item-specific genetic effects correspond to different tissues or cell types than factor-associated effects. Such a difference would suggest that the observed item-specific loci reflect different genetic mechanisms or processes that aren't fully shared across the factor. We assess this possibility for each of the top 5 items in Factors 11 and 16, respectively, since both have known relevant tissues and multiple loci significantly associated with the top items that aren't observed in the corresponding factor GWAS.

Methods

To evaluate this possibility, we first use Genomic Structural Equation Modeling (gSEM; 62) to estimate genetic effects for a given item controlling for the shared genetic effect present in the GWAS of the factor. The item-specific effects are estimated for each SNP based on a GWAS-by-subtraction model (63). The structural part of the model is estimated using the GWAS results for HapMap3 SNPs along with reference panel LD scores previously computed from European ancestry individuals from the 1000 Genomes Project (64). We compare genome-wide significant loci from this GWAS-by-subtraction for a given item to the GWAS of the factor to identify potentially item-specific locus associations.

The GWAS-by-subtraction results for each item are then tested for tissue and cell type enrichments using LD score regression (65). We consider a total of 694 annotations based on gene expression and chromatin marks from ROADMAP, ENCODE, GTEx, and Franke lab, each tested conditional on the baseline LD v1.2 model controlling for annotated functional categories, LD-related genomic features, and MAF (66). For each item, the LD score regression for each cell type is compared to the corresponding cell type coefficient from LD score regression of the factor GWAS after standardizing the coefficients based on the estimated total observed scale SNP-heritability for the GWAS-by-subtraction of the item and the GWAS of the factor, respectively. Pairwise comparison between the item and the factor results is done for each tissue/cell type, and then the general trend across all 694 annotations is tested using Deming regression to account for the estimated standard errors on each estimate.

Results

After GWAS-by-subtraction with genomic SEM, 3 of the top 5 items from Factor 11 and none of the top 5 items from Factor 16 had genome-wide significant loci ($p < 5e-8$) remaining. The

observed loci from Factor 11 items, however, were all loci previously observed to be strongly associated with both the item and the factor, with consistently weaker results in the conditional analysis. This could indicate either incomplete control for shared effects in the GWAS-by-subtraction model or differential effects at shared loci. In either case, these results do not identify any distinctly item-specific loci reaching significance in the current results.

Across the 10 tested items, a total of 19 tissue-item pairs have significantly enriched genetic signal after Bonferroni correction within each item ($p < 7.2e-5 = .05/694$). A majority of the significant enrichments (11) are from regions with chromatin marks associated with T cells in the GWAS-by-subtraction of self-report of asthma or hayfever/allergic rhinitis/eczema diagnoses (UKB code 6152) in Factor 11. On the other hand, 18 of the 19 tissues, including all of the T cell annotations, also have significant enrichments in LD score regression of the corresponding factor GWAS. The one exception is for an enrichment related to regions of gene expression in blood vessels (A07.231.Blood.Vessels) which is significantly enriched after GWAS-by-subtraction of myocardial infarction (ICD code I21; $p=6.3e-5$) but only nominally enriched in the GWAS of factor 16 ($p=.039$). The difference in the coefficient between these results is not significant though (standardized factor coefficient = $7.10e-8$ [SE= $4.04e-8$], item coefficient = $1.99e-7$ [SE= $5.20e-8$], difference $p=.051$).

More broadly, the tissue/cell-type coefficients from the GWAS-by-subtraction results for item-specific genetic signal are highly concordant with the tissue/cell-type results for the GWAS of the corresponding factors (e.g. **Supplementary Figure 10A**). Across all items, no tissues/cell types have significantly different enrichment coefficients between the item-specific and factor GWASs after correction for multiple testing. Furthermore, for 9 of the 10 items the observed results are consistent with perfect correlation in cell type results between the factor and GWAS-by-subtraction of the item ($p > 0.5$ for Deming regression model goodness of fit). The exception is breathing problems during a job period (UKB code 22616), whose item-specific GWAS shows much less consistency in tissue/cell-type enrichments compared to Factor 11 ($p=1.45e-4$; **Supplementary Figure 10B**). Still, no individual cell type reaches Bonferroni-corrected significance for this comparison.

Taken together, these results do not identify any clear item-specific loci or patterns of enrichment among the top items in Factors 11 and 16. To the extent that there is any genetic signal remaining after controlling for the shared signal through GWAS-by-subtraction there is some evidence of loci and tissues that are shared with the factor but may have differential effects on some items. On the other hand it is difficult to distinguish true differential effects of this sort from incomplete

control of the shared effects in the gSEM modeling, so the remaining effects should be interpreted with caution. We also note that given the items are highly correlated with the factors (by design) there is generally low remaining SNP-heritability in the GWAS-by-subtraction results after controlling for the factor (observed scale $h^2_g=0.0011-0.0693$ in top 5 Factor 11 items, $h^2_g=0.00002-0.0036$ in Factor 16 items), limiting power to identify any enrichments. Thus it remains possible that item-specific loci and mechanisms exist, both for these items or in other factors we have not evaluated here, but that we lack sufficient power to identify them in the current data.

Supplementary Figure 10. Comparison of coefficients for cell/tissue type annotations between GWAS of factor scores and GWAS-by-subtraction of top items controlling for the corresponding factor. Coefficients are from LD score regression of 694 expression and chromatin-based annotations, scaled based on the total observed SNP heritability. The dashed red reference line indicates equal coefficients in the two analyses (slope = 1). Purple dots are nominally ($p < .05$) different from the equality line, but are not significant after multiple testing correction. The solid blue line indicates the results of Deming regression on the plotted data, accounting for the reported standard errors on the coefficients. Goodness of fit statistics for the Deming regression correspond to the null hypothesis that the true coefficients are perfectly correlated with measurement error equal to the reported standard errors.

Comment 4: The other aspect of interest - though it might be more of a statistical artifact than biology - is the presence of strong genetic correlation between factors, even when they are phenotypically orthogonal (e.g. factors 5, 10 and 15 in Fig 5). I'm not really sure what this tells me about genetic architecture, but it deserves some thought.

Response: While the model for the factors is orthogonal, the resulting factor scores are not constrained to be phenotypically uncorrelated, since adding that constraint induces other biases (McDonald & Burr 1967). As such, we note this cross-factor association in the Supplement:

For modeling purposes, with an eye towards computational scale and downstream analyses, we forced all factors to be orthogonal in both the EFA and CFA. Notably, though the *latent* factors were specified as orthogonal, the *observed* factor scores (generated in our case using extensions of the Bartlett and Thomson-Thurstone methods) were not necessarily orthogonal. Nonetheless, the highest pairwise correlation between factor scores was 0.176 (i.e., between F4 and F33; mean correlation=0.001[0.044]; see **Supplementary Figure 8**).

Supplementary Figure 8. Phenotypic and genetic correlations across factors. Phenotypic correlations between factors are shown in the lower triangle, and genetic correlations are shown in the upper triangle. Color indicates the magnitude and direction of correlation.

In addition, even if the factors were phenotypically uncorrelated, that would not require them to be genetically uncorrelated. In such cases the factors would have non-genetic (e.g., environmental) correlations counter-balancing their genetic correlation. Nevertheless, we agree with the reviewer that in most cases such a pattern is unlikely, and so the observed genetic correlations are likely to involve statistical artifacts from orthogonalization rather than providing any clear biological insights.

We have now further clarified our interpretation of these results within that section of the Supplement:

Finally, as shown in **Supplementary Figure 8**, requiring the underlying latent phenotypic factors to be orthogonal (and, consequently, for the most part, for the factor scores to be uncorrelated) does not imply that they are genetically uncorrelated. For cases in which the factors are genetically correlated but phenotypically uncorrelated, they would have non-genetic (e.g., environmental) correlations counter-balancing their genetic correlation. However, we believe this conflicting pattern of association to be primarily driven by statistical artifacts of orthogonalization and thus caution against any biological interpretation.

Detailed issues

Please note that some of these are potentially quite important - but they are more detailed.

Comment 5: There is little discussion of previous attempts to use factor analysis and other dimensionality-reduction techniques when analysing biobank-style data (or at least multimorbidity data).

Response: We fully agree with the reviewer's assessment and have now added additional paragraphs to the Introduction citing prior publications using dimensionality reduction on phenotypic data or genetic summary statistics at the biobank scale (and thank the reviewer for the helpful citations in their comments above!):

Dimensionality reduction is a common task in many domains, with a recent proliferation of methods having been applied to biobank-scale data. Principal component analysis (PCA), for example, provides a lower-dimensional representation of the strongest axes of variation in a dataset. It has been leveraged to, for instance, identify dimensions of genetic ancestry in genotype data (8), extract features from individual biobank questionnaires (9), and identify sets of genetic variants with similar patterns of association across thousands of genome-wide association studies (GWAS)(10). Other data reduction approaches prioritize identifying correlated sets of variables across data types (11), modeling latent classes (12), or creating lower-dimensional representations for visualization purposes (13,14). Deep learning methods such as autoencoders and transformers have been used to integrate 'omics data across modalities (15,16), and to extract relevant features from electronic health records (EHRs) (17,18), respectively.

Biobank analyses have devoted relatively less attention to factor analysis, an approach commonly used in the social sciences that models the observed correlation between variables as arising from one or more shared continuous latent (unobserved) factors (19). Factor analysis has the benefit of being model-based, facilitating more direct statistical inference than descriptive summaries (e.g., PCA) or “black box” algorithmic solutions, and it directly prioritizes extracting factors that have a simple relationship to the observed items, when possible (20,21). Conventionally, factor analysis is applied to sets of items within a single questionnaire to identify or confirm underlying structure and estimate scores that more accurately measure the latent constructs captured. This approach has scaled successfully to large cohorts, for example in modeling measures of cognition²² or well-being (23), or in identifying structure across disease comorbidities²⁴. Recently, factor analysis has been adapted to model the structure of genetic, rather than observed phenotypic, correlations across traits (25–27), with further extensions to other types of large-scale ‘omics data proposed (28).

Comment 6: While the impact of factor loadings on mortality is assessed, I expect that many of these may vary in terms of the age and sex distribution. Given that age and sex impact mortality, I would expect to see this analysis removing the effects of such covariates.

Response: This was the case in initial analyses but thank the reviewer for pointing out that this was not clear based on the manuscript text. We have now been more explicit about covariates used in mortality analyses within the Methods section:

In addition to the standard phenotypic analysis covariates (i.e., first 20 genetic PCs, age, chromosomal sex, age², age-x-chromosomal sex, age²-x-chromosomal sex, and dummy variables representing the assessment centers of origin), a covariate was added representing days from baseline assessment to T_0 .

And have included these covariates in the legend of Figure 2:

Figure 2. Prospective mortality hazard ratios and heritability estimates across all 35 factors. (A) Mortality hazards per factor. Factors are ordered from most protective to most predictive of mortality from time of last survey completion to the last date at which death records were available for analysis. T_0 was defined as the last contact an individual had with the UK Biobank

study, within the context of the items included in the factors. **(B)** Heritability estimates for each factor. Factors are ordered by heritability point estimates. Error bars in Panel **A** reflect 95% CIs, while error bars in Panel **B** reflect standard errors. For both panels, darker blue boxes remain significant after adjustment for multiple comparisons. Covariates for mortality and heritability analyses included the first 20 genetic PCs, age, chromosomal sex, age², age-x-chromosomal sex, age²-x-chromosomal sex, and dummy variables representing the assessment centers of origin. Mortality analyses additionally included a covariate representing days from baseline assessment to *T*₀.

We have also now included a Supplementary Table (**Supplementary Table 11**) with associations between factor scores and our included phenotypic covariates:

	Age		Sex		Genetic PCs		Centers		All Covs.	
	Pearson R	P-value	Pearson R	P-value	Model R2	Model P-value	Model R2	Model P-value	Model R2	Model P-value
f1	-0.006	2.66E-04	0.001	6.29E-01	0.000	3.80E-25	0.002	8.60E-126	0.002	2.10E-140
f2	0.010	9.83E-10	0.014	2.52E-16	0.001	9.02E-33	0.002	1.43E-138	0.003	7.10E-183
f3	-0.007	1.67E-05	0.009	2.58E-08	0.001	9.24E-36	0.001	1.04E-91	0.002	2.52E-149
f4	-0.005	4.11E-03	0.021	1.06E-37	0.021	0	0.240	0	0.256	0
f5	-0.009	6.79E-08	-0.030	9.81E-74	0.001	1.76E-70	0.013	0	0.016	0
f6	0.001	3.97E-01	-0.020	1.67E-33	0.001	6.81E-83	0.004	1.11E-313	0.005	0
f7	-0.008	1.27E-06	0.008	1.03E-06	0.001	7.76E-47	0.002	1.30E-116	0.003	7.69E-185
f9	-0.051	6.59E-207	-0.019	5.37E-31	0.002	7.56E-103	0.007	0.00E+00	0.011	0
f10	0.057	1.16E-259	-0.007	9.43E-06	0.004	3.91E-285	0.038	0	0.044	0
f11	-0.005	4.26E-03	-0.001	7.41E-01	0.001	1.16E-35	0.001	3.99E-97	0.002	1.38E-116
f12	0.004	1.65E-02	0.007	3.54E-05	0.001	5.61E-55	0.002	3.89E-159	0.003	8.55E-199
f13	-0.013	9.92E-15	-0.011	1.06E-11	0.001	2.05E-95	0.005	0	0.006	0
f14	-0.041	2.17E-132	-0.003	6.51E-02	0.002	7.73E-152	0.011	0	0.014	0
f15	-0.058	2.61E-263	0.007	7.97E-06	0.002	2.01E-175	0.020	0	0.026	0
f16	0.039	6.54E-123	-0.001	4.22E-01	0.002	1.84E-125	0.005	0	0.008	0
f17	-0.018	1.13E-13	0.004	7.37E-02	0.000	3.07E-09	0.005	5.75E-165	0.006	6.48E-184
f18	0.000	8.27E-01	-0.001	6.18E-01	0.001	3.56E-45	0.005	0	0.005	0
f19	-0.009	8.29E-08	-0.016	1.17E-22	0.001	6.39E-46	0.001	8.59E-68	0.002	3.81E-121
f20	-0.039	3.62E-124	-0.025	1.12E-51	0.016	0	0.074	0	0.078	0
f21	-0.018	6.19E-28	0.051	2.22E-210	0.001	6.85E-83	0.007	0	0.011	0
f22	-0.061	2.13E-295	-0.032	5.66E-84	0.001	3.03E-35	0.002	8.66E-144	0.007	0
f23	0.007	4.96E-05	0.018	2.77E-26	0.001	2.86E-55	0.003	8.36E-233	0.004	2.60E-287
f24	0.100	0	0.121	0	0.028	0	0.023	0	0.074	0
f25	-0.017	2.09E-25	-0.004	1.57E-02	0.001	3.81E-39	0.004	6.06E-298	0.005	0
f26	0.000	9.36E-01	-0.001	3.95E-01	0.001	5.57E-41	0.005	0	0.006	0
f27	-0.166	0	-0.026	3.52E-53	0.005	0	0.103	0	0.137	0
f28	0.022	1.26E-38	0.003	1.32E-01	0.001	5.23E-30	0.002	1.54E-119	0.003	4.61E-184
f29	0.028	8.08E-62	-0.025	1.74E-51	0.001	1.57E-57	0.004	9.44E-279	0.007	0
f30	0.004	7.28E-03	-0.023	4.50E-45	0.001	3.54E-84	0.002	2.23E-137	0.003	7.40E-233
f31	0.031	9.76E-79	0.024	3.55E-48	0.001	3.45E-52	0.007	0	0.010	0
f32	0.008	1.26E-06	-0.010	6.41E-09	0.001	2.31E-34	0.004	3.24E-274	0.005	0
f33	0.008	1.77E-06	-0.002	2.15E-01	0.135	0	0.656	0.00E+00	0.705	0
f34	-0.020	1.92E-32	0.001	4.95E-01	0.003	8.69E-205	0.005	0.00E+00	0.008	0
f35	0.006	1.90E-04	0.000	8.18E-01	0.001	2.10E-32	0.001	1.75E-45	0.001	6.67E-78
f36	0.005	4.28E-03	-0.001	7.02E-01	0.000	6.18E-18	0.000	2.29E-12	0.001	2.51E-25

Comment 7: It's not clear to me that heritability estimates for factors and individual items are directly comparable given that they are measured on different scales. Please consider whether this needs to be addressed.

Response: The reviewer is correct that heritability estimates are affected by the type of measurement (i.e., continuous vs. binary items). Conventionally this is resolved by reporting all estimates on the liability scale with adjustments for dichotomizing items, etc. However, for the factor scores there is not a clear transformation to the liability scale since they are neither binary (i.e., since they are weighted sums over a large number of items) nor fully continuous (i.e., when individual binary/categorical items contribute substantial weight). For this reason, we choose to report all heritabilities on the observed scale. Although this will tend to understate the amount of underlying genetics that may exist for binary/categorical items, it will equitably indicate the amount of genetic signal captured in the item/factor as it is currently measured. We think that identifying the amount of signal captured in the existing measure of each item/factor remains quite informative despite the scale restrictions.

We have attempted to make it more clear in the text that we are considering the observed-scale heritability only, as this may have previously been confusing due to the ambiguous use of the word "observed":

...the observed-scale SNP heritability of the 35 factors is on the whole higher than for the 505 component items (mean item observed-scale $h_g^2=0.05[0.07]$; 2-sample t-test $p=0.002$; **Figure 3A**).

We also now make clear in a new Methods subsection that we are making these comparisons to items on the *observed* scale, and make our reasoning for this choice clear as well:

Comparison of factor vs. item SNP heritability

For each of the 505 items included in the final factor model, observed-scale hg^2 estimates were downloaded from the publicly available Neale Lab UKB Round 2 mega-GWAS Heritability Browser (https://nealelab.github.io/UKBB_Idsc/index.html).

It should be noted that conventionally, in order to account for differences in heritability estimates by measurement type (i.e., continuous vs. binary items), estimates are reported on the liability scale. Indeed, this is the default estimate reported in the Heritability Browser. However, for factor scores there is not a clear transformation to the liability scale, since they are neither binary (i.e., since they are weighted sums over a large number of items) nor fully continuous (i.e., when individual binary/categorical items contribute substantial weight). For this reason, we report all heritabilities—for both factors and constituent items—on the observed scale. Although this will tend to understate the amount of underlying genetics that may exist for binary items, it will equitably indicate the amount of genetic signal captured in the item or factor as it is currently measured.

Comment 8: When there is comparison of genetic correlation to phenotypes from external GWAS, I note that these often contain UK Biobank as a constituent cohort - and often the dominant one. I think these should be compared only to cohorts within UK Biobank - for example, when a factor has a strong overlap with EA, it is not surprising that it has a strong genetic correlation with EA measured in the same cohort.

Response: For the purposes of this response, we assume the reviewer meant “compared only to cohorts *outside of* UK Biobank”, not “*within* UK Biobank.”

To help readers assess the degree of potential overlap between samples used in our analyses and in prior summary statistics, we have now created a Supplementary Table (**Supplementary Table 12**) detailing relevant information for each of these prior studies, including 1) whether or not the study included UKB samples, and 2) the maximum genetic covariance intercept with any of our factors, which is expected to be proportional to correlation in sampling error between the two GWAS, as a function of the degree of overlap and the phenotypic correlation.

The first few rows of this Supplementary Table are pasted below for reference:

Category	Phenotype	PMID	Download Link	Contains UKB Samples?	Maximum Gcov Intercept
neurological	Parkinson's disease	31701892	LINK	Y	0.017

	Alzheimer's disease	34493870	LINK	Y	0.031
	stroke	29531354	LINK	N	0.017
	epilepsy	30531953	LINK	N	0.020
anthropometric	BMI	25673413	LINK	N	0.018
	height	25282103	LINK	N	0.122
	WHR	25673412	LINK	N	0.018
	waist circumference	25673412	LINK	N	0.024
	hip circumference	25673412	LINK	N	0.019
	body fat percentage	26833246	LINK	N	0.015
	lean body mass	28724990	LINK	N	0.016
	intracranial volume	27694991	LINK	N	0.013

We have also noted within the Results section this genetic covariance intercept, and the genetic correlations with UKB participants removed, when such sumstats were available:

Factor 11, which captures asthma diagnosis and related medications, comorbidities, and lab findings, has a strong genetic correlation with a prior GWAS of asthma (45) ($r_g=0.89(0.01)$; $\text{Intercept}_{\text{Gcov}}=0.59[0.02]$; UKB-excluded [46] $r_g=0.82[0.04]$)...

GWAS of Factor 16 captures known lipid biology, ... and exhibits strong genetic correlation with a prior CAD GWAS (49) ($r_g=0.87[0.02]$; $\text{Intercept}_{\text{Gcov}}=0.42[0.01]$; UKB-excluded [50] $r_g=0.81[0.04]$).

Comparison of Factor 28 to a prior GWAS of type 2 diabetes (50), however, reveals imperfect capture of the clinical definition ($r_g=0.68[0.02]$, $\text{Intercept}_{\text{Gcov}}=0.36[0.01]$; UKB-excluded $r_g=0.70[0.03]$; **Extended Data Figure 9**)...

Factor 10 shares substantial genetic overlap with prior GWAS of EA (60) ($r_g=0.93[0.01]$; $\text{Intercept}_{G_{cov}}=0.29[0.01]$; UKB-excluded $r_g=0.92[0.02]$) and its correlates, including household income (61) and cognitive performance (62).

Factor 9 shows strong genetic correlations with prior GWAS of trauma exposure (75) ($r_g=0.93[0.02]$; $\text{Intercept}_{G_{cov}}=0.53[0.01]$), childhood maltreatment (76) ($r_g=0.81[0.02]$; $\text{Intercept}_{G_{cov}}=0.46[0.01]$), and PTSD (77) ($r_g=0.75[0.07]$; $\text{Intercept}_{G_{cov}}=0.13[0.01]$). It has moderate genetic correlations with external GWAS of psychiatric (e.g., schizophrenia [78] $r_g=0.35[0.03]$; $\text{Intercept}_{G_{cov}}=0.02[0.01]$) and substance use (e.g., cannabis use disorder [79] $r_g=0.45[0.05]$; $\text{Intercept}_{G_{cov}}=0.01[0.01]$) outcomes (**Extended Data Figure 7**).

We now further note the above caveats in a new Methods subsection as well:

Genetic correlation analyses

Genetic correlations across factors were computed using LD score regression (119) with LD score estimates derived from European-ancestry individuals in the 1000 Genomes Project. We additionally ran genetic correlations between our factors and 68 traits selected from prior GWAS, spanning a number of anthropometric, medical, psychological, behavioral, and sociodemographic domains (**Supplementary Table 12**). Traits used for genetic correlation analyses were chosen before conducting the analyses, with the agreement of the coauthors. Note that given the near-ubiquity of the UK Biobank in modern genetics research, a number of these prior GWAS share overlapping samples and, at times, component phenotype definitions, with the current study. As such, we have reported genetic correlations with UKB samples removed within the text as well, when such summary statistics were available. We also report the genetic covariance intercepts, which are a proxy for the amount of sample overlap present across pairs of summary statistics within a genetic correlation analysis (119,134).

Comment 9: I don't think it's justifiable really to only focus on European ancestry individuals. I understand the desire for reducing heterogeneity, but kind of the point of the paper is to embrace the full breadth of human biology and I don't see how you can exclude people based on their race, ethnicity or genetic ancestry on that basis. It feels inappropriate to me.

Response: We fully agree with the reviewer here, and we truly wish we had the data and statistical power to explore other ancestries. Indeed, some of our colleagues are deeply involved with creating methods and data that we hope will soon be available to push back on this unfortunate trend of highly European data-driven genetics analyses, so that ideally future biobank analyses can take advantage of the full breadth of human diversity. However, given that a large portion of our analytic plan involved characterizing the factors based on orthogonal data and included genetics, which is notoriously sensitive to ancestral heterogeneity, we decided that use of the full dataset was beyond the scope of these analyses. We do view this as a key limitation to our paper and as such have explicitly noted the restricted ancestry within the abstract:

In one of the largest uses of factor analytic methods to date, we distill hundreds of diagnoses, assessments, and survey items from **unrelated individuals with predominantly estimated European genetic ancestry** in UK Biobank into 35 latent constructs.

We also devote additional text within the Discussion to ancestry restriction, as well as non-representativeness of the population in general, as a limitation:

Participants at the core of this study are, for example, unlikely to be representative of the global population. Analyses were restricted to UKB participants of predominately estimated European genetic ancestry, thus limiting generalizability of results, particularly as it relates to genetic inference (98). In addition, UK Biobank participants are known to be nonrepresentative of the UK population as a whole, with documented ascertainment and participation biases (4,99–101). Even among those participating in the study, individuals providing more complete initial responses and contributing to later optional follow-up surveys are known to be, in general, healthier and more educated (102,103). This trend is reflected in our own core, low-missingness data group being substantially more likely to report having completed college or university (45.7% core group, 30.7% non-core group; $\chi^2=3816.0$, $p<0.001$). While polygenic score analyses in Add Health suggest that genetic signal captured in the fitted factors at least partially generalizes to individuals of European ancestry in the United States, identifying which constructs elucidated in this study are most robustly maintained across different sociopolitical, cultural, and diagnostic contexts requires future work in additional cohorts.

We join the reviewer in being highly disappointed that discriminatory, societally-driven factors have created a world in which integrating individuals who are historically disadvantaged into genetics research is methodologically fraught, and risks further perpetuating incorrect assumptions of inherited differences across populations.

Comment 10: I couldn't work out what the units of Extended Data Fig. 3 are - please label.

Response: We thank the reviewer for their great attention to detail! We have updated the figure's accompanying text to more clearly denote that the box-and-whisker plots show the distributions of association test statistics (i.e., z scores) of the factor with medical phecodes within each category, while including our standard covariates (i.e., first 20 genetic PCs, age, chromosomal sex, age², age-x-chromosomal sex, age²-x-chromosomal sex, and dummy variables representing the assessment centers of origin) in each regression. Note that due to restructuring, this is now **Extended Data Figure 4**:

Extended Data Figure 4. Phcode associations by factor. Box-and-whisker plots are shown for associations with 403 derived medical phcodes grouped by category. These associations are defined as the test statistics (i.e., z scores) for the factor score in a logistic regression model including our standard covariates (i.e., first 20 genetic PCs, age, chromosomal sex, age², age-x-chromosomal sex, age²-x-chromosomal sex, and dummy variables representing the assessment

centers of origin). Boxes represent the middle quartiles of a factor’s test statistics across phecodes within a category, with whiskers extending to 1.5x the interquartile range. Median values per category are indicated by individual black lines inside the boxes. The dotted grey lines represents the critical test statistics for significance at $p < 0.05$ once correcting for comparisons across all 403 phecodes.

Given the similarity of Extended Data Figure 4 to Main Text Figure 5, we have updated the corresponding figure text there as well:

Figure 5. Factor 9 associations across top-level inpatient diagnostic phecodes. Box-and-whisker plots are shown for associations within UKB with 403 derived medical phecodes grouped by category. These associations are defined as the test statistics (i.e., z scores) for the factor score in a logistic regression model including our standard covariates (i.e., first 20 genetic PCs, age, chromosomal sex, age², age-x-chromosomal sex, age²-x-chromosomal sex, and dummy variables representing the assessment centers of origin). Boxes represent the middle quartiles of Factor 9’s test statistics across phecodes within a category, with whiskers extending to maximum and minimum observed values, excluding outliers >1.5x the interquartile range away from the middle quartiles while are plotted individually. Median values per category are indicated by individual black

lines inside the boxes. The dotted grey lines represent the critical test statistics for significance at $p < 0.05$ once correcting for comparisons across all 403 phecodes.

We finally wish to end this response by thanking the editors and reviewers for their insights. As we're sure the editors and reviewers know well, it is rare that a paper gets three sets of reviews and editors' comments that are so useful and important for improving a paper. This is truly a rare opportunity, and we hope that our edits, new analyses, and responses have not only satisfied the reviewers and editors, but have transformed our paper for the better.

Decision Letter, first revision:

16th October 2023

Dear Dr. Carey,

Thank you for your patience as we've prepared the guidelines for final submission of your Nature Human Behaviour manuscript, "Principled distillation of UK Biobank phenotype data reveals underlying structure in human variation" (NATHUMBEHAV-22102813A). Please carefully follow the step-by-step instructions provided in the attached file, and add a response in each row of the table to indicate the changes that you have made. Please also address the additional marked-up edits we have proposed within the reporting summary. Ensuring that each point is addressed will help to ensure that your revised manuscript can be swiftly handed over to our production team.

We would hope to receive your revised paper, with all of the requested files and forms within two-three weeks. Please get in contact with us if you anticipate delays.

Nature Human Behaviour offers a Transparent Peer Review option for new original research manuscripts submitted after December 1st, 2019. As part of this initiative, we encourage our authors to support increased transparency into the peer review process by agreeing to have the reviewer comments, author rebuttal letters, and editorial decision letters published as a Supplementary item. When you submit your final files please clearly state in your cover letter whether or not you would like to participate in this initiative. Please note that failure to state your preference will result in delays in accepting your manuscript for publication.

In recognition of the time and expertise our reviewers provide to Nature Human Behaviour's editorial process, we would like to formally acknowledge their contribution to the external peer review of your manuscript entitled "Principled distillation of UK Biobank phenotype data reveals underlying structure in human variation". For those reviewers who give their assent, we will be publishing their names alongside the published article.

Cover suggestions

We welcome submissions of artwork for consideration for our cover. For more information, please see our guide for cover artwork.

ORCID

Non-corresponding authors do not have to link their ORCIDs but are encouraged to do so. Please note that it will not be possible to add/modify ORCIDs at proof. Thus, please let your co-authors know that if they wish to have their ORCID added to the paper they must follow the procedure described in the following link prior to acceptance: <https://www.springernature.com/gp/researchers/orcid/orcid-for-nature-research>

Nature Human Behaviour has now transitioned to a unified Rights Collection system which will allow our Author Services team to quickly and easily collect the rights and permissions required to publish your work. Approximately 10 days after your paper is formally accepted, you will receive an email in providing you with a link to complete the grant of rights. If your paper is eligible for Open Access, our Author Services team will also be in touch regarding any additional information that may be required to arrange payment for your article.

Please note that *Nature Human Behaviour* is a Transformative Journal (TJ). Authors may publish their research with us through the traditional subscription access route or make their paper immediately open access through payment of an article-processing charge (APC). Authors will not be required to make a final decision about access to their article until it has been accepted. Find out more about Transformative Journals

For information regarding our different publishing models please see our Transformative

Journals page. If you have any questions about costs, Open Access requirements, or our legal forms, please contact ASJournals@springernature.com.

[REDACTED]

Best regards,

[REDACTED]

Editorial Assistant

Nature Human Behaviour

On behalf of

[REDACTED]

Senior Editor

Nature Human Behaviour

Reviewer #1:

Remarks to the Author:

No further comments

Reviewer #2:

Remarks to the Author:

The authors have adequately addressed the comments made by the reviewers in the revised version of the manuscript. Therefore, I have no further comments.

Reviewer #3:

Remarks to the Author:

Many thanks to the authors for such comprehensive revision. There has been a lot of additional work done and the resulting manuscript is considerably improved. I enjoyed reading it and I'm sure many others will too.

I have only two remaining comments:

1. The authors have pushed back on my recommendation to extend to all ethnic groups / genetic ancestries in UK Biobank. Whether this is or isn't acceptable is an editorial decision. FWIW, I don't think it is acceptable - imagine explaining to someone in UK Biobank that you decided to exclude them on the basis of their race? I don't think history will judge such decisions in anything other than a discriminatory light, however 'reasonable' the motivation. The authors are obviously not alone in taking this stance, which is why they feel it is defensible, but I don't think that's an excuse

personally.

2. I didn't raise it the first time round, so feel free to ignore, but I'm not sure many readers will understand what is meant by 'Principled distillation'. Why not just 'Factor analysis'? (FA is no more principled than PCA or LDA for example).

Final Decision Letter:

Dear Dr Carey,

We are pleased to inform you that your Article "Principled distillation of UK Biobank phenotype data reveals underlying structure in human variation", has now been accepted for publication in *Nature Human Behaviour*.

Please note that *Nature Human Behaviour* is a Transformative Journal (TJ). Authors may publish their research with us through the traditional subscription access route or make their paper immediately open access through payment of an article-processing charge (APC). Authors will not be required to make a final decision about access to their article until it has been accepted. Find out more about Transformative Journals

Authors may need to take specific actions to achieve compliance with funder and institutional open access mandates. If your research is supported by a funder that requires immediate open access (e.g. according to Plan S principles) then you should select the gold OA route, and we will direct you to the compliant route where possible. For authors selecting the subscription publication route, the journal's standard licensing terms will need to be accepted, including self-archiving policies. Those licensing terms will supersede any other terms that the author or any third party may assert apply to any version of the manuscript.

With best regards,
[REDACTED]